# Private Identity Testing for High-Dimensional Distributions

**Clément Canonne**
IBM Research, Almaden
ccanonne@cs.columbia.edu

**Gautam Kamath**
Cheriton School of Computer Science
University of Waterloo
g@csail.mit.edu

**Audra McMillan**
Khoury College of Computer Sciences, Northeastern University
Department of Computer Science, Boston University
audramarymcmillan@gmail.com

**Jonathan Ullman**
Khoury College of Computer Sciences
Northeastern University
jullman@ccs.neu.edu

**Lydia Zakynthinou**
Khoury College of Computer Sciences
Northeastern University
zakynthinou.l@northeastern.edu

## Abstract

We present novel differentially private identity (goodness-of-fit) testers for natural and widely studied classes of multivariate product distributions: product distributions over $\{\pm1\}^d$ and Gaussians in $\mathbb{R}^d$ with known covariance. Our testers have improved sample complexity compared to those derived from previous techniques, and are the first testers whose sample matches the order-optimal minimax sample complexity of $O(d^{1/2}/\alpha^2)$ in many parameter regimes. We construct two types of testers, exhibiting tradeoffs between sample complexity and computational complexity. Finally, we provide a two-way reduction between testing a subclass of multivariate product distributions and testing univariate distributions, thereby obtaining upper and lower bounds for testing this subclass of product distributions.

## 1 Introduction

A foundation of statistical inference is *hypothesis testing*: given two disjoint sets of probability distributions $\mathcal{H}_0$ and $\mathcal{H}_1$, we want to design an algorithm $T$ that takes a random sample $X$ from some distribution $P \in \mathcal{H}_0 \cup \mathcal{H}_1$ and, with high probability, determines whether $P$ is in $\mathcal{H}_0$ or $\mathcal{H}_1$. Hypothesis tests formalize yes-or-no questions about an underlying population given a random sample from that population, and are ubiquitous in the physical, life, and social sciences, where hypothesis tests with high confidence are the gold standard for publication in top journals.

In many of these applications—clinical trials, social network analysis, or demographic studies, to name a few—this sample contains sensitive data belonging to individuals, in which case it is crucial for the hypothesis test to respect these individuals' *privacy*. It is particularly desirable to guarantee *differential privacy* [33], which has become the *de facto* standard for the analysis of private data. Differential privacy is used as a measure of privacy for data analysis systems at Google [35], Apple [31], and the U.S. Census Bureau [26]. Differential privacy and related notions of *algorithmic stability* are also crucial for statistical validity even when confidentiality is not a direct concern, as they provide generalization guarantees in an adaptive setting [32, 8, 58].

While differentially private hypothesis testing has been extensively studied (see Section 1.3), almost all this work has focused on *low-dimensional distributions*. Our main contribution is to give novel algorithms for hypothesis testing of *high-dimensional distributions* with improved sample complexity. In particular, we give differentially private algorithms for the following fundamental problems:

1. Given samples from a product distribution $P$ over $\{\pm 1\}^d$, decide if $P$ is the uniform distribution or is $\alpha$-far from the uniform distribution in total variation distance. Or, equivalently, decide if $\mathbb{E}[P] = 0$ or if $\|\mathbb{E}[P]\|_2 \geq \alpha$.

2. Given samples from a product distribution $P$ over $\{0,1\}^d$, decide if $P$ is equal to some given extremely biased distribution $Q$ with mean $\mathbb{E}[Q] \preceq O(\frac{1}{d})$ or is $\alpha$-far from $Q$ in total variation distance. In this case our tester achieves the provably optimal sample complexity.

3. Given samples from a multivariate Gaussian $P$ in $\mathbb{R}^d$ whose covariance is known to be the identity $\mathbb{I}_{d \times d}$, decide if $P$ is $\mathcal{N}(0, \mathbb{I}_{d \times d})$ or is $\alpha$-far from $\mathcal{N}(0, \mathbb{I}_{d \times d})$ in total variation distance. Or, equivalently, decide if $\mathbb{E}[P] = 0$ or if $\|\mathbb{E}[P]\|_2 \geq \alpha$.

Although we will focus on the first contribution since it highlights the main technical contributions, we note that the third contribution, (private) hypothesis testing on the mean of a Gaussian, is one of the most fundamental statistical primitives (see, e.g., the Z-test and Student's t-test). The main challenge in solving these high-dimensional testing problems privately is that the only known non-private test statistics for these problems have high worst-case sensitivity. That is, these test statistics can potentially be highly brittle to changing even a single one of the samples. We overcome this challenge by identifying two methods for reducing the sensitivity of the test statistic without substantially changing its average-case behavior on typical datasets sampled from the distributions we consider. The first is based on a novel *private filtering* method, which gives a computationally efficient tester. The second combines the method of *Lipschitz extensions* [13, 52] with *recursive preconditioning*, which yields an exponential-time tester, but with improved sample complexity.

## 1.1    Background: Private Hypothesis Testing

We start by giving some background on private hypothesis testing. First, we say that an algorithm $A \colon \mathcal{X}^* \to \{0,1\}$ is a differentially private hypothesis tester for a pair $\mathcal{H}_0, \mathcal{H}_1$ over domain $\mathcal{X}$ if

1. $A$ is $\varepsilon$-differentially private in the *worst case*. That is, for *every* pair of samples $X, X' \in \mathcal{X}^n$ differing on one sample, $A(X)$ and $A(X')$ are $\varepsilon$-*close*. In general, $A(X)$ and $A(X')$ are $(\varepsilon, \delta)$-close if for $b \in \{0, 1\}$, $\Pr[A(X) = b] \leq e^\varepsilon \Pr[A(X') = b] + \delta$.

2. $A$ distinguishes $\mathcal{H}_0$ from $\mathcal{H}_1$ *on average*. If $X = (X^{(1)}, \ldots, X^{(n)})$ is drawn i.i.d. from some $P \in \mathcal{H}_0$ then $A(X)$ outputs 0 with high probability, and similarly if $P \in \mathcal{H}_1$. The minimum number of samples $n$ such that $A$ distinguishes $\mathcal{H}_0$ and $\mathcal{H}_1$ is the *sample complexity* of $A$.

Note that we want testers that are private in the worst case, yet accurate on average. It is important for privacy to be a worst-case notion, rather than contingent on the assumption that the data is sampled i.i.d. from some $P \in \mathcal{H}_0 \cup \mathcal{H}_1$ because we have no way of verifying this assumption (which may be a modeling simplification), and once privacy is lost it cannot be recovered. Worst-case privacy notions also enjoy strong composition and generalization properties not shared by average-case privacy.

There exists a black-box method for obtaining a differentially private tester from any non-private tester $A$ using the *sample-and-aggregate framework* [56]. Specifically, given any tester $A$ with sample complexity $n$, we can obtain an $\varepsilon$-differentially private tester with sample complexity $O(n/\varepsilon)$. When $\varepsilon$ is a constant, the reduction is within a constant factor of the optimal sample complexity; however, this overhead factor in the sample complexity blows up as $\varepsilon \to 0$.

One can often obtain stronger results using a white-box approach. For example, suppose $P$ is a Bernoulli random variable and we aim to test if $P = \mathrm{Ber}(1/2)$ or $P = \mathrm{Ber}(1/2 + \alpha)$. Non-privately, $\Theta(1/\alpha^2)$ samples are necessary and sufficient. Thus the black-box approach gives a sample complexity of $\Theta(1/\alpha^2\varepsilon)$. However, if we work directly with the test statistic $T(X) = \frac{1}{n}\sum_{j=1}^{n} X^{(j)}$, we can obtain privacy by computing $T(X) + Z$ where $Z$ is drawn from an appropriate distribution

with standard deviation $O(1/\varepsilon n)$. One can now show that this private test succeeds using

$$n = \underbrace{O\left(\frac{1}{\alpha^2}\right)}_{\text{non-private sc}} + \underbrace{O\left(\frac{1}{\alpha\varepsilon}\right)}_{\text{overhead for privacy}} . \tag{1}$$

samples, which actually *matches* the non-private sample complexity up to a factor of $1 + o(1)$ unless $\varepsilon$ is very small, namely $\varepsilon = o(\alpha)$. Our main contribution is to achieve qualitatively similar results for the high-dimensional testing problem we have described above.

## 1.2 Our Results

**Theorem 1.1** (Informal). *There is a linear-time, $\varepsilon$-differentially private tester $A$ that distinguishes the uniform distribution over $\{\pm1\}^d$ from any product distribution over $\{\pm1\}^d$ that is $\alpha$-far in total variation distance using $n = n(d, \alpha, \varepsilon)$ samples for*

$$n = \underbrace{O\left(\frac{d^{1/2}}{\alpha^2}\right)}_{\text{non-private sc}} + \underbrace{\tilde{O}\left(\frac{d^{1/2}}{\alpha\varepsilon}\right)}_{\text{overhead for privacy}} . \tag{2}$$

The sample complexity in Theorem 1.1 has an appealing form. One might even conjecture that this sample complexity is optimal by analogy with the case of privately *estimating* a product distribution over $\{\pm1\}^d$ or a Gaussian in $\mathbb{R}^d$ with known covariance, for which the sample complexity is $\tilde{\Theta}(d/\alpha^2 + d/\alpha\varepsilon)$ in both cases [51]. However, the next result shows that there is in fact an exponential-time private tester that has even lower sample complexity in some range of parameters.

**Theorem 1.2** (Informal). *There is an exponential-time, $\varepsilon$-differentially private tester $A$ that distinguishes the uniform distribution over $\{\pm1\}^d$ from any product distribution over $\{\pm1\}^d$ that is $\alpha$-far in total variation distance using $n = n(d, \alpha, \varepsilon)$ samples for*

$$n = \underbrace{O\left(\frac{d^{1/2}}{\alpha^2}\right)}_{\text{non-private sc}} + \underbrace{\tilde{O}\left(\frac{d^{1/2}}{\alpha\varepsilon^{1/2}} + \frac{d^{1/3}}{\alpha^{4/3}\varepsilon^{2/3}} + \frac{1}{\alpha\varepsilon}\right)}_{\text{overhead for privacy}} . \tag{3}$$

We remark that the sample complexity in (3) is precisely, up to logarithmic factors, the optimal sample complexity for testing uniformity of discrete distributions on the domain $\{1, \ldots, d\}$ [3], which hints that it may be optimal (especially in view of Theorem 1.3 below). The current best lower bound we can prove is $n = \Omega(\frac{d^{1/2}}{\alpha^2} + \frac{1}{\alpha\epsilon})$. Bridging the gap between this lower bound and the upper bound seems like it would require developing fundamentally different lower bound techniques. Lower bounds for multivariate distribution testing appear to be much more challenging to prove than in the univariate case, due to the necessity of maintaining independence of marginals in the construction of a coupling, which is the standard technique for proving lower bounds for private distribution testing.

The expression in (3) is rather complex and somewhat difficult to interpret and compare to (2). One way to simplify the comparison is to consider the range of the privacy parameter $\varepsilon$ where privacy comes *for free*, meaning the sample complexity is dominated by the non-private term $\Theta(d^{1/2}/\alpha^2)$. For the efficient algorithm, privacy comes for free roughly when $\varepsilon = \Omega(\alpha)$. For the computationally inefficient algorithm, however, one can show that privacy comes for free roughly when $\varepsilon = \Omega(\alpha^2 + \alpha/d^{1/4})$, which is better if both $1/\alpha$ and $d$ are superconstant.

Using a simple reduction, we show that Theorems 1.1 and 1.2 extend, with a constant-factor loss in sample complexity, to identity testing for *balanced* product distributions. That is, suppose $Q$ is a product distribution such that every coordinate of $\mathbb{E}[Q]$ is bounded away from $-1$ and $+1$. Then with a constant-factor loss in sample complexity, we can distinguish whether (i) a product distribution $P$ over $\{\pm1\}^d$ is either equal to $Q$, or (ii) whether $P$ is far from $Q$ in total variation distance.

We then focus on a specific class of Boolean product distributions, which we refer to as *extreme*. Informally, a product distribution is extreme if each of its marginals is $O(1/d)$-close to constant. For this restricted class of product distributions, we provide a two-way reduction showing that identity testing is equivalent to the identity testing in the *univariate* setting. This allows us to transfer known lower bounds on private univariate identity testing to our extreme product distribution class, which gives us the first non-trivial lower bounds for privately testing identity of product distributions.

**Theorem 1.3** (Informal). *The sample complexity of privately testing identity of univariate distributions over $[d]$ and the sample complexity of privately testing identity of* extreme *product distributions over $\{\pm 1\}^d$ are equal, up to constant factors.*

Finally, we can obtain analogous results for identity testing of multivariate Gaussians with known covariance by reduction to uniformity testing of Boolean product distributions.

**Theorem 1.4** (Informal). *There is a linear-time, $\varepsilon$-differentially private tester $A$ that distinguishes the standard normal $\mathcal{N}(0, \mathbb{I}_{d \times d})$ from any normal $\mathcal{N}(\mu, \mathbb{I}_{d \times d})$ that is $\alpha$-far in total variation distance (or, equivalently, $\|\mu\|_2 \geq \alpha$) using $n = n(d, \alpha, \varepsilon)$ samples for $n = O(\frac{d^{1/2}}{\alpha^2}) + \tilde{O}(\frac{d^{1/2}}{\alpha \varepsilon})$.*

**Theorem 1.5** (Informal). *There is an exponential-time $\varepsilon$-differentially private tester $A$ that distinguishes the standard normal $\mathcal{N}(0, \mathbb{I}_{d \times d})$ from any normal $\mathcal{N}(\mu, \mathbb{I}_{d \times d})$ that is $\alpha$-far in total variation distance using $n = n(d, \alpha, \varepsilon)$ samples for $n = O(\frac{d^{1/2}}{\alpha^2}) + \tilde{O}(\frac{d^{1/2}}{\alpha \varepsilon^{1/2}} + \frac{d^{1/3}}{\alpha^{4/3} \varepsilon^{2/3}} + \frac{1}{\alpha \varepsilon})$.*

We note that we can also obtain these results directly by extending the techniques of the Boolean case and constructing the tester. We demonstrate this for the computationally efficient test.

We highlight some tools in this paper which are useful for hypothesis testing, even without privacy constraints: (1) A reduction from testing the mean of a Gaussian, to testing uniformity of a product distribution. (2) A reduction from testing identity to a "balanced" product distribution, to testing uniformity of a product distribution.(3) An equivalence between testing identity over a domain of size $d$, and testing identity to an "extreme" product distribution in $d$ dimensions.

## 1.3 Related Work

We focus on the most relevant related work here, broader coverage appears in the supplement. Hypothesis testing with a focus on minimax rates was initiated by Ingster and coauthors in the statistics community [46, 47, 48], and the work of Batu *et al.* [10, 9] in computer science (arising as a subfield of property testing [42, 43]). Works on testing in the multivariate setting include testing of independence [9, 5, 60, 54, 2, 30, 21], and testing on graphical models [20, 28, 27, 38, 1, 11]. We note that graphical models (both Ising models and Bayesian Networks) include the product distribution case we study in this paper. Surveys and more thorough coverage include [59, 19, 41, 7, 50]. Minimax study in the private was setting initiated by Cai *et al.* [17], and we now have close to optimal algorithms and lower bounds for the univariate setting [3, 4]. Other recent works focus on testing of simple hypotheses [25, 22]. Awan and Slavkovic [6] give a universally most powerful (UMP) test when the domain size is two, however Brenner and Nissim [16] shows that UMP tests don't exist for larger domains. A line of work complementary to the minimax setting [64, 63, 65, 37, 53, 49, 18, 62, 23] designs differentially private versions of popular test statistics.

The Lipschitz-extension technique that we build upon was introduced in [13, 52], who also gave efficient Lipschitz extensions for graph statistics such as edge density and number of triangles in sparse graphs. Later work constructed efficient Lipschitz extensions for richer classes of graph statistics [57, 61]. [24] introduced a variant of the Lipschitz-extension machinery to give efficient algorithms for statistics like the median and trimmed mean. Recent results [15, 14] prove the existence of Lipschitz extensions for all differentially private algorithms, though efficiency is not a focus.

## 1.4 Techniques

To avoid certain technicalities involving continuous and unbounded data, we will describe our tester for product distributions over $\{\pm 1\}^d$, rather than for Gaussians over $\mathbb{R}^d$. The Gaussian case can be handled by using a reduction to the Boolean case, or directly, by using a nearly identical approach.

**First Attempts.** A natural starting point is to study the asymptotically-optimal non-private tester of Canonne *et al.* [20]. Let $P$ be a distribution over $\{\pm 1\}^d$ and $X = (X^{(1)}, \ldots, X^{(n)}) \in \{\pm 1\}^{n \times d}$ be $n$ i.i.d. samples from $P$. Define the test statistic

$$T(X) = \|\bar{X}\|_2^2 - nd \text{ where } \bar{X} = \sum_{j=1}^{n} X^{(j)} \tag{4}$$

The analysis of Canonne *et al.* shows that if $P$ is the uniform distribution then $\mathbb{E}[T(X)] = 0$, while if $P$ is a product distribution that is $\alpha$-far from uniform then $\mathbb{E}[T(X)] = \Omega(\alpha^2 n^2)$. Moreover,

the variance of $T(X)$ is bounded so that, for $n = O(d^{1/2}/\alpha^2)$, we can distinguish between the two. To obtain a private tester, we need to add noise to $T(X)$ with smaller magnitude than the $\alpha^2 n^2$ gap. The standard approach is to add noise to $T(X)$ calibrated to the *global sensitivity*: $\mathrm{GS}_T = \max_{X \sim X'}(T(X) - T(X'))$, where $X \sim X'$ denotes that $X$ and $X'$ are *neighboring samples* that differ on at most one sample. To ensure privacy it is then sufficient to compute a noisy statistic $T(X) + Z$ where $Z$ is chosen from an appropriate distribution (commonly, a Laplace distribution) with mean 0 and standard deviation $O(\mathrm{GS}_T/\varepsilon)$. One can easily see that the global sensitivity of $T(X)$ is $O(nd)$, so it suffices to add noise $O(nd/\varepsilon)$. The tester will require $n = \Omega(d/(\alpha^2\varepsilon))$ to succeed, giving an undesirable linear dependence in $d$. Note that this tester is dominated by the sample-and-aggregate approach.

In view of the above, one way to find a better private tester would be to identify an alternative statistic for testing uniformity of product distributions with lower global sensitivity. However, we do not know of any other such test statistic that has asymptotically optimal non-private sample complexity; a prerequisite to achieving optimal private sample complexity.

**Intuition: High-Sensitivity is Atypical.** Since we need privacy to hold in the *worst case* for every dataset, any private algorithm for computing $T$ must have error proportional to $\mathrm{GS}_T$ on *some* dataset $X$. However, the utility of the tester only applies *on average* to typical datasets drawn from product distributions. Thus, we can try to find some alternative test statistic $\hat{T}$ that is close to $T$ on these typical datasets, but has lower global sensitivity. To this end, it will be instructive to look at the sensitivity of $T$ at a particular pair of datasets $X \sim X'$, differing in samples $X^{(j)}$ and $X'^{(j)}$.

$$T(X) - T(X') = 2(\langle X^{(j)}, \bar{X} \rangle - \langle X'^{(j)}, \bar{X}' \rangle) \tag{5}$$

Notice that $T(X)$ and $T(X')$ can only differ by a large amount when one of the two datasets contains a point $X^{(j)}$ such that $\langle X^{(j)}, \bar{X} \rangle$ is large. Thus, if we could somehow restrict attention to datasets in

$$\mathcal{C}(\Delta) = \left\{ X \in \{\pm 1\}^{n \times d} : \forall j = 1, \ldots, n \ |\langle X^{(j)}, \bar{X} \rangle| \leq \Delta \right\}$$

the sensitivity would be at most $4\Delta$. As we will show, for typical datasets drawn from product distributions (except in some corner cases), the data will lie in $\mathcal{C}(\Delta)$ for $\Delta \ll nd$. For example, if we draw $X^{(1)}, \ldots, X^{(n)}$ uniformly at random in $\{\pm 1\}^d$, then we have

$$\mathbb{E}\left[ \max_{j \in [n]} |\langle X^{(j)}, \bar{X} \rangle| \right] \lesssim d + (nd)^{1/2}.$$

More generally, if $X^{(1)}, \ldots, X^{(n)}$ are drawn from any product distribution, then we show that

$$\max_{j \in [n]} |\langle X^{(j)}, \bar{X} \rangle| \lesssim \frac{1}{n}\|\bar{X}\|_2^2 + \|\bar{X}\|_2. \tag{6}$$

Note that, except in pathological cases, the right-hand side will be much smaller than $nd$. Thus, for typical datasets drawn from product distributions, the sensitivity of $T(X)$ should be much lower than its global sensitivity. We can exploit this in two different ways to obtain improved testers.

**A Sample-Efficient Tester via Lipschitz Extensions** Suppose we fix some value of $\Delta$ and we are only concerned with estimating $T(X)$ accurately on nice datasets that lie in $\mathcal{C}(\Delta)$. Even in this case, it would not suffice to add noise to $T(X)$ proportional to $\Delta/\varepsilon$, since we need privacy for *all* datasets.

However, we can compute $T(X)$ accurately on $\mathcal{C}(\Delta)$ while guaranteeing privacy for all datasets using the beautiful machinery of *privacy via Lipschitz extensions* introduced by Blocki *et al.* [13] and Kasiviswanathan *et al.* [52] in the context of node-differential-privacy for graph data. Specifically, for a function $T$ defined on domain $\mathcal{X}$, and a set $\mathcal{C} \subset \mathcal{X}$, a *Lipschitz extension of $T$ from $\mathcal{C}$* is a function $\hat{T}$ defined on all datasets such that: (1) $\hat{T}(X) = T(X)$ for every $X \in \mathcal{C}$. (2) The global sensitivity of $\hat{T}$ on all of $\mathcal{X}$ is at most the sensitivity of $T$ restricted to $\mathcal{C}$, namely $\mathrm{GS}_{\hat{T}} \leq \max_{X,X' \in \mathcal{C}} T(X) - T(X')$.

Perhaps surprisingly, such a Lipschitz extension exists for *every* real-valued function $T$ and every subset of the domain $\mathcal{C}$ [55]! Once we have the Lipschitz extension, we can achieve privacy for all datasets by adding noise to $\hat{T}$ proportional to the sensitivity of $T$ on $\mathcal{C}$. In general, the Lipschitz extension can be computed in time polynomial in $|\mathcal{X}|$, which, in our case, is $2^{nd}$.

Thus, if we knew a small value of $\Delta$ such that $X \in \mathcal{C}(\Delta)$ then we could compute a Lipschitz extension of $T$ from $\mathcal{C}(\Delta)$ and we would be done. From (6), we see that a good choice of $\Delta$ is approximately $\frac{1}{n}\|\bar{X}\|_2^2 + \|\bar{X}\|_2$. But $\|\bar{X}\|_2^2$ is precisely the test statistic we wanted to compute!

We untie this knot using a recursive approach. We start with some weak upper bound $\Delta^{(m)}$ such that we know $X \in \mathcal{C}(\Delta^{(m)})$. For example, we use the worst-case $\Delta^{(1)} = 4nd$ as a base case. Using the Lipschitz extension onto $\mathcal{C}(\Delta^{(m)})$, we can get a weak private estimate of the test statistic $T(X) = \|\bar{X}\|_2^2 - nd$ with error $\lesssim \Delta^{(m)}/\varepsilon$. At this point, we may already be able to conclude that $X$ was not sampled from the uniform distribution and reject. Otherwise, we can certify that $\|\bar{X}\|_2^2 \lesssim nd + \frac{\Delta^{(m)}}{\varepsilon}$.If $X$ was indeed sampled from a product distribution, then (6) tells us that $X \in \mathcal{C}(\Delta^{(m+1)})$ for

$$\Delta^{(m+1)} \lesssim \frac{\|\bar{X}\|_2^2}{n} + \|\bar{X}\|_2 \lesssim d + \frac{\Delta^{(m)}}{\varepsilon n} + \sqrt{nd} + \sqrt{\frac{\Delta^{(m)}}{\varepsilon}}.$$

Then, as long as $\Delta^{(m+1)}$ is significantly smaller than $\Delta^{(m)}$ we can recurse, and get a better private estimate of $\|\bar{X}\|_2^2$. Once the recurrence converges, we can stop and make a final decision whether to accept or reject. One can analyze this recurrence and show that it will converge rapidly to some $\Delta^* = O(d + \sqrt{nd} + 1/\varepsilon)$. Thus the final test will distinguish uniform from far-from-uniform provided $\Delta^*/\varepsilon \ll \alpha^2 n^2$, which occurs at the sample complexity we claim.

This recursive approach is loosely similar in spirit to methods in [51, 36], whereby we obtain more and more accurate private estimates, necessitating less noise addition at each step. However, our technique is very different from theirs, and the first to interact with the Lipschitz extension framework. We believe our work adds more evidence for the broad applicability of this perspective.

**A Computationally Efficient Tester via Private Filtering.** The natural way to make the test above computationally efficient is to explicitly construct the Lipschitz extension $\hat{T}$ onto $\mathcal{C}(\Delta)$. Although we do not know how to do so, we can start with the following *filtering* approach: To compute $\hat{T}(X)$, throw out every $X^{(j)}$ such that $\langle X^{(j)}, \bar{X} \rangle > \Delta$ and replace it with a fresh uniform sample $\hat{X}^{(j)}$. Let the resulting dataset be $Y$ and output $T(Y)$.

Obviously this will agree with $T(X)$ for $X \in \mathcal{C}(\Delta)$, but it is not clear that it has sensitivity $O(\Delta)$. The reason is that the decision to throw out $X^{(j)}$ or not depends on the global quantity $\bar{X} = \sum_j X^{(j)}$. Thus, we could have one dataset $X$ where no points are thrown out so $Y = X$, and a neighboring dataset $X'$ where, because $\bar{X}'$ is slightly different from $\bar{X}$, all points are thrown out so $Y' = \emptyset$. Then the difference between the test statistic on $Y, Y'$ could be much larger than $\Delta$.

We solve this problem by modifying the algorithm to throw out points based on $\langle X^{(j)}, \tilde{X} \rangle$ for some *private* quantity $\tilde{X} \approx \bar{X}$. Although the proof is somewhat subtle, we can use the fact that $\tilde{X}$ is private to argue that filtering based on $\tilde{X}$ and then adding noise calibrated to $\approx \Delta$ preserves privacy. In order to compensate for the fact that $\tilde{X}$ is only an approximation of $\bar{X}$, we cannot take $\Delta$ too small, which is why this tester has larger sample complexity than the non-computationally efficient one.

**Organisation.** Due to space constraints, we focus in this extended abstract on Theorem 1.2, which illustrates the main ideas and challenges in our approach, and provide a detailed outline of its proof in Section 2. All omitted details and sections can be found in the supplementary material.

## 2 Uniformity Testing for Product-of-Bernoulli Distributions

In this section, we provide the outline of our computationally inefficient algorithm for uniformity testing, Algorithm 2. We will prove the following main theorem (corresponding to Theorem 1.2).

**Theorem 2.1.** *Algorithm 2 is $\varepsilon$-differentially private. Furthermore, Algorithm 2 can distinguish between $P = \mathcal{U}_d$ and $\|P - \mathcal{U}_d\|_1 \geq \alpha$ with probability at least $2/3$ and has sample complexity $n = \tilde{O}\left(\frac{d^{1/2}}{\alpha^2} + \frac{d^{1/2}}{\alpha\varepsilon^{1/2}} + \frac{d^{1/3}}{\alpha^{4/3}\varepsilon^{2/3}} + \frac{1}{\alpha\varepsilon}\right).$*

We first focus on the privacy guarantee of Theorem 2.1, which is based on the existence of Lipschitz extensions, as established by [55]. For our dataset $X$ and any $\Delta > 0$, let us define $\mathcal{C}(\Delta) =$

$\{X \in \{\pm 1\}^{n \times d} \mid \forall j \in [n], |\langle X^{(j)}, \bar{X} \rangle| \leq \Delta\}$. The main element that we need for the privacy proof is the bound on the sensitivity of $T$ on $\mathcal{C}(\Delta^{(m)})$ for all $m \in [M]$. This would ensure that $T$ is $4\Delta^{(m)}$-Lipschitz on $\mathcal{C}(\Delta^{(m)})$, so the $4\Delta^{(m)}$-Lipschitz extensions $\hat{T}$ exist in all the rounds and line 3 in LipExtTest and line 11 in Algorithm 2 add enough noise to maintain privacy. Note that the algorithm would be private regardless of the choice of values $\Delta^{(m)}$.

Recall the definition of the test statistic of (4): $T(X) = \|\bar{X}\|_2^2 - nd$ for $\bar{X} = \sum_{j=1}^{n} X^{(j)}$. The next lemma holds by a calculation of the sensitivity which led to equation (5) and the definition of $\mathcal{C}(\Delta)$.

**Lemma 2.2** (Sensitivity of T). *For any bound $\Delta > 0$, for two neighboring datasets $X, X' \in \mathcal{C}(\Delta)$, $|T(X) - T(X')| \leq 4\Delta$.*

First, we note that $\text{LipExtTest}(X, \varepsilon', \Delta, \beta)$ is $\varepsilon'$-DP, since it only accesses $X$ via the Laplace mechanism in line 3. Since Algorithm 2 is a composition of $M - 1$ invocations of LipExtTest with DP parameter $\varepsilon' = \varepsilon/M$, and of the Laplace mechanism in line 11, its privacy guarantee is a straightforward application of DP composition and post-processing.

**Lemma 2.3** (Privacy). *Algorithm 2 is $\varepsilon$-differentially private.*

---

**Algorithm 1** LipExtTest

---

**Require:** Sample $X = (X^{(1)}, \ldots, X^{(n)})$. Parameters $\varepsilon, \Delta > 0, \beta \in (0, 1]$.
1: Define the set $\mathcal{C}(\Delta) = \{X \in \{\pm 1\}^{n \times d} \mid \forall j \in [n], |\langle X^{(j)}, \bar{X} \rangle| \leq \Delta\}$.
2: Let $\hat{T}(\cdot)$ be a $4\Delta$-Lipschitz extension of $T$ from $\mathcal{C}(\Delta)$ to all of $\{\pm 1\}^{n \times d}$.
3: Sample noise $r \sim \text{Lap}(4\Delta/\varepsilon)$ and let $z \leftarrow \hat{T}(X) + r$.
4: **if** $z > 10n\sqrt{d} + 4\Delta \ln(1/\beta)/\varepsilon$ **then return** reject.
5: **return** accept.

---

**Algorithm 2** Private Uniformity Testing via Lipschitz Extension

---

**Require:** Sample $X = (X^{(1)}, \ldots, X^{(n)}) \in \{\pm 1\}^{n \times d}$ drawn from $P^n$. Parameters $\varepsilon, \alpha, \beta > 0$.
1: Let $M \leftarrow \lceil \log n \rceil, \varepsilon' \leftarrow \varepsilon/M, \beta \leftarrow 1/(10n)$.
2: Let $\Delta^{(1)} \leftarrow nd$ and $\Delta^* \leftarrow 1000 \max(d, \sqrt{nd}, \ln(1/\beta)/\varepsilon') \cdot \ln(1/\beta)$.
3: **for** $m \leftarrow 1$ to $M - 1$ **do**
4: $\quad$ **if** $\Delta^{(m)} \leq \Delta^*$ **then**
5: $\quad\quad$ Let $\Delta^{(M)} \leftarrow \Delta^{(m)}$ and exit the loop.
6: $\quad$ **else**
7: $\quad\quad$ **if** $\text{LipExtTest}(X, \varepsilon', \Delta^{(m)}, \beta)$ returns reject **then return** reject
8: $\quad\quad$ Set $\Delta^{(m+1)} \leftarrow 11\left(d + \sqrt{nd} + \frac{\Delta^{(m)}}{n\varepsilon'} + \sqrt{\frac{\Delta^{(m)}}{\varepsilon'}}\right) \ln \frac{1}{\beta}$.
9: Define the set $\mathcal{C}(\Delta^{(M)}) = \{X \in \{\pm 1\}^{n \times d} \mid \forall j \in [n], |\langle X^{(j)}, \bar{X} \rangle| \leq \Delta^{(M)}\}$.
10: Let $\hat{T}(\cdot)$ be a $4\Delta^{(M)}$-Lipschitz extension of $T$ from $\mathcal{C}(\Delta^{(M)})$ to all of $\{\pm 1\}^{n \times d}$.
11: Sample noise $r \sim \text{Lap}(4\Delta^{(M)}/\varepsilon')$ and let $z \leftarrow \hat{T}(X) + r$.
12: **if** $z > \frac{1}{4}n(n-1)\alpha^2$ **then return** reject
13: **return** accept.

---

We now turn our attention to the utility guarantee. The next lemma will help us determine how the bound on the sensitivity of the statistic decreases in each round, as stated in (6). It is established by a concentration bound on the sum of the independent r.v.'s $Y_i = x_i \bar{X}_i \mid \bar{X}$, around its mean $\|\bar{X}\|_2^2/n$.

**Lemma 2.4.** *If $X$ is drawn i.i.d. from a product distribution, then, with probability at least $1 - 2n\beta$,*

$$\forall x \in X, \; |\langle x, \bar{X} \rangle| \leq \frac{\|\bar{X}\|_2^2}{n} + \sqrt{2}\|\bar{X}\|_2 \sqrt{\ln(1/\beta)}. \tag{7}$$

As in Algorithm 2, we let $\Delta^* = 1000 \max(d, \sqrt{nd}, M \ln(1/\beta)/\varepsilon) \cdot \ln(1/\beta)$, where $\beta = 1/(10n)$ and $M = \lceil \log(n) \rceil$. Based on the previous lemma, we now analyze the key subroutine, LipExtTest:

1. If $X$ is drawn from a product distribution and, for some $\Delta$, LipExtTest returns accept, then $X \in \mathcal{C}(\Delta')$, where $\Delta' \ll \Delta$, determined as in line 8 of Algorithm 2 (Lemma 2.5);

2. If $X$ is drawn from the uniform distribution, LipExtTest will return accept (Lemma 2.6).

**Lemma 2.5** (Sensitivity reduction). *Fix some $\Delta \geq 0$, and let $\Delta' = 11(d + \sqrt{nd} + \Delta/(n\varepsilon) + \sqrt{\Delta/\varepsilon}) \ln(1/\beta)$. If (i) $X$ is drawn from a product distribution, (ii) $X$ satisfies (7), (iii) $X \in \mathcal{C}(\Delta)$, and (iv) $\textsc{LipExtTest}(X, \varepsilon, \Delta, \beta)$ returns* **accept***, then $X \in \mathcal{C}(\Delta')$ with probability at least $1 - \beta$.*

To prove the lemma, we first observe that, by condition (iii), $\hat{T}(X) = T(X) = \|\bar{X}\|_2^2 - nd$. So condition (iv) and the Laplace mechanism guarantee that $\|\bar{X}\|_2^2 \leq 11(nd + \Delta \ln(1/\beta)/\varepsilon)$, with probability $1 - \beta$. By conditions (i) and (ii), (7) implies that $\forall x \in X \ |\langle x, \bar{X}\rangle| \leq \Delta'$ for the given $\Delta'$.

**Lemma 2.6.** *If (i) $X$ is drawn from the uniform distribution $\mathcal{U}_d$, satisfies $T(X) \leq 10n\sqrt{d}$, and (7), and (ii) $\Delta \geq \Delta^*$, then $\textsc{LipExtTest}(X, \varepsilon, \Delta, \beta)$ returns* **accept** *with probability at least $1 - \beta$.*

*Proof.* By condition (i), $\|\bar{X}\|_2^2 = T(X) + nd \leq 11nd$. Plugging this in (7), we get that $\forall x \in X$, $|\langle x, \bar{X}\rangle| \leq 11nd/n + 2\sqrt{nd\ln(1/\beta)} \leq 22 \max(d, \sqrt{nd\ln(1/\beta)}) \leq \Delta^* \leq \Delta$, where the last inequality holds by condition (ii). Thus, $X \in \mathcal{C}(\Delta)$ so $\hat{T}(X) = T(X)$ in line 2 and we can analyze line 4 by bounding $\hat{T}(X) + r = T(X) + r \leq 10n\sqrt{d} + r$. By the Laplace mechanism, $|r| \leq 4\Delta \ln(1/\beta)/\varepsilon$, with probability $1 - \beta$. Hence, $\textsc{LipExtTest}$ accepts with probability $1 - \beta$. $\square$

With the analysis of this subroutine, we turn to the utility guarantee, which, unlike the privacy, only needs to hold for datasets drawn from a product distribution. The crux of the proof is the following:

1. If $X$ is drawn i.i.d. from a product distribution and passes line 3 in round $m$, then it belongs in $\mathcal{C}(\Delta^{(m+1)})$ with high probability. Thus in every round that the dataset has not been rejected, we have that with high probability $\hat{T}(X) = T(X)$.

2. If $P = \mathcal{U}_d$, then with high probability $X$ passes all the steps in the loop at line 3.

3. The number of rounds $M$ is sufficient to guarantee that the sensitivity and thus the noise added to $\hat{T}(X)$ in line 11 is small enough that one distinguishes between the two hypotheses.

For the latter, we will need to bound $\Delta^{(M)}$, i.e., the sensitivity of $T(X)$ restricted to $\mathcal{C}(\Delta^{(M)})$. The proof follows by observing that in each round where $\Delta > \Delta^*$, $\Delta$ is reduced by $1/2$, based on line 8.

**Lemma 2.7.** *For $M = \lceil \log(n) \rceil$, $\varepsilon' = \varepsilon/M$, and $n = \Omega(\log(1/\beta)/\varepsilon')$, $\Delta^{(M)} \leq \Delta^*$.*

We can now complete our proof outline, showing that the last test will give the expected guarantees. Since our test relies on a private adaptation of the test from [20], we state a version of its guarantees.

**Lemma 2.8** (Non-private Test Guarantees). *For the test statistic $T(X) = \|\bar{X}\|_2^2 - nd$:*

- *If $P = \mathcal{U}_d$ then $\mathbb{E}[T(X)] = 0$, whereas if $\|P - \mathcal{U}_d\|_1 \geq \alpha$ then $\mathbb{E}[T(X)] > \frac{1}{2}n(n-1)\alpha^2$.*

- *$\mathrm{Var}(T(X)) \leq 2n^2 d + 4n\mathbb{E}[T(X)]$.*

*Proof of Theorem 2.1.* The privacy guarantee was established in Lemma 2.3. First, note that given $\Delta^{(1)} = nd$, we have $\mathcal{C}(\Delta^{(1)}) = \{\pm 1\}^{n \times d}$, so $X \in \mathcal{C}(\Delta^{(1)})$.

Completeness: Suppose $X$ is drawn from $\mathcal{U}_d$. By the guarantees of the non-private test and Chebyshev's inequality, with probability $49/50$, $T(X) \leq 10n\sqrt{d}$. Moreover, since $\mathcal{U}_d$ is *a fortiori* a product distribution, $X$ satisfies (7) with probability at least $1 - 2n\beta$. Assume both of these events hold. By Lemma 2.6 and a union bound over the $M - 1$ calls in line 7, we get that with probability at least $1 - M\beta$ every call returns **accept**. Assume this holds. Invoking Lemma 2.5, this implies that for all rounds $m$, $X \in \mathcal{C}(\Delta^{(m)})$ except with probability at most $M\beta$, and in particular $X \in \mathcal{C}(\Delta^{(M)})$.

In this case, we have $\hat{T}(X) = T(X)$ in line 11. It remains to show that with high probability, $z = T(X) + r \leq n(n-1)\alpha^2/4$ in line 12. Since $r \sim \mathrm{Lap}(4\Delta^{(M)}/\varepsilon')$, with probability at least $49/50$, $|r| \leq 4\Delta^{(M)}(\ln 50)/\varepsilon'$. Again, condition on this, and suppose for now that $n$ satisfies

$$|r| \leq \frac{4\Delta^{(M)} \ln 50}{\varepsilon'} \leq \frac{n(n-1)\alpha^2}{8}. \tag{8}$$

Then, by Chevyshev's and the non-private test guarantees, we can bound the probability that Algorithm 2 rejects in line 11 as $\Pr[z > n(n-1)\alpha^2/4] \leq 1/50$ for $n = \Omega(d^{1/2}/\alpha^2)$. Therefore, by an overall union bound and $\beta = 1/(10n)$, the algorithm rejects with probability at most $1/3$.

It remains to determine the constraint on $n$ for inequality (8) to hold. By Lemma 2.7, it suffice for $n^2\alpha^2 = \Omega(\Delta^*/\varepsilon') = \Omega\left(\max(d, \sqrt{nd}, \log(1/\beta)/\varepsilon') \cdot \log(1/\beta)/\varepsilon'\right)$. Recalling our choice of $\varepsilon' = \varepsilon/M$, $\beta = 1/(10n)$, and $M = \lceil \log(n) \rceil$, it suffices to choose

$$n = \tilde{\Omega}\left(\frac{d^{1/2}}{\alpha\varepsilon^{1/2}} + \frac{d^{1/3}}{\alpha^{4/3}\varepsilon^{2/3}} + \frac{1}{\alpha\varepsilon}\right). \tag{9}$$

Soundness: Suppose $X$ is drawn from a product distribution which is $\alpha$-far from uniform. By Lemma 2.4, $X$ satisfies (7) with probability at least $1 - 2n\beta$. Suppose that the algorithm does not return reject in any of the $M - 1$ calls in line 7 (otherwise, we are done). Conditioning on this, by Lemma 2.5 and an immediate induction, with probability $1 - M\beta$, $X \in \mathcal{C}(\Delta^{(m)})$ for all rounds $m$, and in particular $X \in \mathcal{C}(\Delta^{(M)})$. The rest of the proof follows exactly as the completeness, relying on (8), the guarantees of the Laplace mechanism and the non-private test, and Chebyshev's inequality. Combining equation (9) with the non-private bound $\Omega(\frac{d^{1/2}}{\alpha^2})$ yields the stated sample complexity. $\square$

## Broader Impact

This work focuses on privacy-preserving statistics: the task of performing statistical learning while maintaining the privacy of data participants. The right to privacy is a fundamental right, and works in this area allow for the facilitation of valuable social science and economic research, while preserving this fundamental right. There are many examples where releases of statistical data, even aggregate statistics, without formal privacy guarantees has resulted in blatant privacy violations. For example, Homer et al. [45] demonstrated that it is possible to identify individuals who participated in a GWAS study based on certain released test statistics.This can be especially harmful when resolving individuals who participated in trials pertaining to a socially stigmatized disease (e.g., HIV/AIDS). This finding was considered so significant that the NIH immediately revoked open access to a number of statistical results from medical trials, including $\chi^2$-statistics and $p$ values. Hypothesis tests are applied to sensitive data in a myriad of studies, e.g. on voting behavior [44] or attitude toward abortion [34], highlighting urgent need for hypothesis tests which respect the privacy of participant data.

Some specific applications of goodness-of-fit testing are [12, 39, 40]. The study of uniformity testing of product distributions is also equivalent to the well-motivated question of *mean testing* of categorical data, which has applications to high-dimensional statistical analysis. Further, uniformity testing is also deeply related to mean testing of high-dimensional normal distributions, as exemplified by our reduction from the latter to the former. Non-DP mean testing in high-dimensional settings itself has a long history in statistical hypothesis testing; see for example, [29].

Our objective is to initiate the study of high-dimensional hypothesis testing under privacy constraints. Although we do not close the chapter on identity testing, we believe our results to be an important, non-trivial, and natural first step, and a prerequisite to any further investigation (e.g., general identity testing for non-spherical Gaussians, graphical models, or sub-Gaussian distributions). Our work exposes and addresses a number of core challenges that arise in private high-dimensional data analysis. Indeed, the problems we study capture a challenge that is widespread in differential privacy: that solving high dimensional testing problems privately is difficult because the known non-private testers have high global sensitivity. We propose a novel way of dealing with this issue by reducing to the "typical sensitivity." We demonstrate the effectiveness of this technique on identity testing, but we conjecture that it will find applications to other testing problems and beyond.

## Acknowledgments and Disclosure of Funding

Part of this work was done while GK, AM, JU, and LZ were visiting the Simons Institute for the Theory of Computing, and while CC was a Motwani Postdoctoral Fellow at Stanford University. GK was supported as a Microsoft Research Fellow, as part of the Simons-Berkeley Research Fellowship program. Some of this work was completed while visiting Microsoft Research, Redmond. AM was supported by NSF grant CCF-1763786, a Sloan Foundation Research Award, and a postdoctoral fellowship from BU's Hariri Institute for Computing. JU and LZ were supported by NSF grants CCF-1718088, CCF-1750640, and CNS-1816028. Subsequently, LZ has been supported by a Facebook

fellowship. CC was supported by a Goldstine Fellowship. The authors would like to thank Jayadev Acharya and Himanshu Tyagi for discussions that contributed to the reduction of identity testing of multivariate Gaussians to uniformity testing of Boolean product distributions.

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
