[Supplementary Material]

# Private Identity Testing for High-Dimensional Distributions

June 5, 2020

**Abstract**

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

### 1.2.4 Useful Tools for Non-Private Hypothesis Testing

We highlight some tools in this paper which are useful for hypothesis testing, even without privacy constraints.

- A reduction from testing the mean of a Gaussian, to testing uniformity of a product distribution (Theorem 3.1).

- A reduction from testing identity to a "balanced" product distribution, to testing uniformity of a product distribution (Corollary 5.2).

- An equivalence between testing identity over a domain of size $d$, and testing identity to an "extreme" product distribution in $d$ dimensions (Theorem 6.2).

## 1.3 Techniques

To avoid certain technicalities involving continuous and unbounded data, we will describe our tester for product distributions over $\{\pm 1\}^d$, rather than for Gaussians over $\mathbb{R}^d$. The Gaussian case can be handled using a reduction to the Boolean case, or can be handled directly using a nearly identical approach.

**First Attempts.** A natural starting point is to study the asymptotically-optimal non-private tester of Canonne *et al.* [CDKS17]. Let $P$ be a distribution over $\{\pm 1\}^d$ and $X = (X^{(1)}, \ldots, X^{(n)}) \in \{\pm 1\}^{n \times d}$ be $n$ i.i.d. samples from $P$. Define the test statistic

$$T(X) = \|\bar{X}\|_2^2 - nd \quad \text{where} \quad \bar{X} = \sum_{j=1}^{n} X^{(j)}$$

The analysis of Canonne *et al.* shows that if $P$ is the uniform distribution then $\mathbb{E}[T(X)] = 0$, while if $P$ is a product distribution that is $\alpha$-far from uniform then $\mathbb{E}[T(X)] = \Omega(\alpha^2 n^2)$. Moreover, the variance of $T(X)$ can be bounded so that, for some $n = O(d^{1/2}/\alpha^2)$ we can distinguish between these two cases. In order to obtain a private tester, we need to somehow add noise to $T(X)$ whose magnitude is much smaller than the $\alpha^2 n^2$ gap between the two cases.

The standard approach to obtaining a private tester is to add noise to the statistic $T(X)$ calibrated to its *global sensitivity*, which is defined as

$$\mathrm{GS}_T = \max_{X \sim X'} (T(X) - T(X'))$$

where $X \sim X'$ denotes that $X$ and $X'$ are *neighboring samples* that differ on at most one sample. To ensure privacy it is then sufficient to compute a noisy statistic $T(X) + Z$ where $Z$ is chosen from an appropriate distribution (commonly, a Laplace distribution) with mean 0 and standard deviation $O(\mathrm{GS}_T/\varepsilon)$.

One can easily see that the global sensitivity of $T(X)$ is $O(nd)$, so it suffices to add noise $O(nd/\varepsilon)$. The tester will still be successful provided $n = \Omega(d/(\alpha^2 \varepsilon))$, resulting in an undesirable linear dependence in $d$. In particular, this is dominated in all parameter regimes

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

 can set $\Delta^{(1)} = 4nd$ as a base case, which is the worst-case upper bound on the sensitivity. Using this upper bound, and the Lipschitz extension onto the set $\mathcal{C}(\Delta^{(m)})$, we can then get a weak private estimate of the test statistic

$T(X) = \|\bar{X}\|_2^2 - nd$ with error $\lesssim \Delta^{(m)}/\varepsilon$. At this point, we may already have enough information to conclude that $X$ was not sampled from the uniform distribution and reject. Otherwise, we can certify that

$$\|\bar{X}\|_2^2 \lesssim nd + \frac{\Delta^{(m)}}{\varepsilon}$$

If $X$ was indeed sampled from a product distribution, then (5) tells us that $X \in \mathcal{C}(\Delta^{(m+1)})$ for

$$\Delta^{(m+1)} \lesssim \frac{\|\bar{X}\|_2^2}{n} + \|\bar{X}\|_2 \lesssim d + \frac{\Delta^{(m)}}{\varepsilon n} + \sqrt{nd} + \sqrt{\frac{\Delta^{(m)}}{\varepsilon}}$$

Then, as long as $\Delta^{(m+1)}$ is significantly smaller than $\Delta^{(m)}$ we can recurse, and get a better private estimate of $\|\bar{X}\|_2^2$. Once the recurrence converges and $\Delta^{(m+1)}$ is no longer getting smaller, we can stop and make a final decision whether to accept or reject. One can analyze this recurrence and show that it will converge rapidly to some $\Delta^* = O(d + \sqrt{nd} + 1/\varepsilon)$. Thus the final test will distinguish uniform from far-from-uniform provided $\Delta^*/\varepsilon \ll \alpha^2 n^2$, which occurs at the sample complexity we claim.

This recursive approach is loosely similar in spirit to methods in [KLSU19, FKT20], whereby we obtain more and more accurate private estimates, necessitating less noise addition at each step. However, our technique is very different from theirs, and the first to interact with the Lipschitz extension framework. We believe our work adds more evidence for the broad applicability of this perspective.

**A Computationally Efficient Tester via Private Filtering.** The natural way to make the test above computationally efficient is to try to explicitly construct the Lipschitz extension $\hat{T}$ onto $\mathcal{C}(\Delta)$ using the following *filtering* approach. Although we do not know how to do so, we can start with the following simple candidate:

> $\hat{T}(X)$ : Throw out every $X^{(j)}$ such that $\langle X^{(j)}, \bar{X} \rangle$ is larger than some $\Delta$ and let the resulting dataset be $Y$. Output $T(Y)$.

Obviously this will agree with $T(X)$ whenever $X \in \mathcal{C}(\Delta)$, however it is unclear how to argue that it has sensitivity at most $O(\Delta)$. The reason is that the decision to throw out $X^{(j)}$ or not depends on the global quantity $\bar{X} = \sum_j X^{(j)}$. Thus, potentially, we could have one dataset $X$ where no points are thrown out so $Y = X$, and a neighboring dataset $X'$ where, because $\bar{X}'$ is slightly different from $\bar{X}$, all points are thrown out and $Y' = \emptyset$. In this case, the difference between the test statistic on $Y, Y'$ could be much larger than $\Delta$.

We solve this problem by modifying the algorithm to throw out points based on $\langle X^{(j)}, \tilde{X} \rangle$ for some *private* quantity $\tilde{X} \approx \bar{X}$. Although the proof is somewhat subtle, we can use the fact that $\tilde{X}$ is private to argue that, under some appropriate coupling of the two executions, $Y$ and $Y'$ differ on at most one sample $Y^{(j)}, Y'^{(j)}$ and $\langle Y^{(j)}, \bar{Y} \rangle$ and $\langle Y'^{(j)}, \bar{Y}' \rangle$ are at most $O(\Delta)$. In order to compensate for the fact that $\tilde{X}$ is only an approximation of $\bar{X}$, we cannot take $\Delta$ too small, which ultimately is why this tester has larger sample complexity than the non-computationally efficient one.

## 1.4 Related Work

Over the last couple decades, there has been significant work on hypothesis testing with a focus on minimax rates. The starting point in the statistics community could be considered

the work of Ingster and coauthors [Ing94, Ing97, IS03]. Within theoretical computer science, study on hypothesis testing arose as a subfield of property testing [GGR96, GR00]. Work by Batu *et al.* [BFR+00, BFF+01] formalized several of the commonly studied problems, including testing of uniformity, identity, closeness, and independence. Other representative works in this line include [BKR04, Pan08, Val11, CDVV14, VV14, ADK15, BV15, DKN15, CDGR16, DK16, Gol16, BCG17, DKW18]. Some works on testing in the multivariate setting include testing of independence [BFF+01, AAK+07, RX14, LRR13, ADK15, DK16, CDKS18], and testing on graphical models [CDKS17, DP17, DDK18, GLP18, ABDK18, BBC+19]. We note that graphical models (both Ising models and Bayesian Networks) include the product distribution case we study in this paper. Surveys and more thorough coverage of related work on minimax hypothesis testing include [Rub12, Can15, Gol17, BW18, Kam18].

Early work on differentially private hypothesis testing began in the statistics community with [VS09, USF13], and there has more recently been a large body of work on this subject primarily in the computer science community. Most relevant to this paper is the line of work on minimax sample complexity of private hypothesis testing, initiated by Cai *et al.* [CDK17]. [ASZ18] and [ADR18] have given worst-case nearly optimal algorithms for goodness-of-fit and closeness testing of arbitrary discrete distributions. Other recent works in this vein focus on testing of simple hypotheses [CKM+18, CKM+19]. A paper of Awan and Slavkovic [AS18] gives a universally optimal test when the domain size is two, however Brenner and Nissim [BN14] shows that such universally optimal tests cannot exist when the domain has more than two elements. A line of work complementary to the minimax results [WLK15, GLRV16, KR17, KSF17, CBRG18, SGHG+19, CKS+19] designs differentially private versions of popular test statistics for testing goodness-of-fit, closeness, and independence, as well as private ANOVA, focusing on the performance at small sample sizes. Work by Wang *et al.* [WKLK18] focuses on generating statistical approximating distributions for differentially private statistics, which they apply to hypothesis testing problems. There has also been work on hypothesis testing in the *local model* of differential privacy, with a focus on both the minimax setting [DJW13, She18, ACFT19] and the asymptotic setting [GR18].

Looking more broadly at private algorithms in statistical settings, there has recently been significant study into estimation tasks. Some settings of interest include univariate discrete distributions [BNSV15, DHS15], Gaussians [KV18, KLSU19], and multivariate settings [KLSU19, CWZ19]. Upper and lower bounds for learning the mean of a product distribution over the hypercube in $\ell_\infty$-distance include [BDMN05, BUV14, DMNS06, SU17]. [AKSZ18] designed nearly optimal algorithms for estimating properties like support size and entropy. [Smi11] gives an algorithm which allows one to estimate asymptotically normal statistics with minimal increase in the sample complexity. In the local model, some works on distribution estimation include [DJW13, WHW+16, KBR16, ASZ19, DR18, YB18, GRS19, JKMW19].

The Lipschitz-extension technique that we build upon was introduced in [BBDS13, KNRS13], who also gave efficient Lipschitz extensions for graph statistics such as the edge density and the number of triangles in sparse graphs. Later work constructed efficient Lipschitz extensions for richer classes of graph statistics [RS16, SU19]. A related work of [CD20] introduced a variant of the Lipschitz-extension machinery and used it to give efficient algorithms for statistics such as the median and trimmed mean. Recent results [BCSZ18b, BCSZ18a] prove the existence of Lipschitz extensions for all differentially private algorithms, though efficiency is not a focus.

# 2 Preliminaries

## 2.1 Differential Privacy

Informally, differential privacy is a property that a randomized algorithm satisfies if its output distribution does not change significantly under the change of a single data point.

Let $S, S' \in \mathcal{S}^n$ be two datasets of the same size. We say that $S, S'$ are *neighbors*, denoted as $S \sim S'$, if they differ in at most one data point.

**Definition 2.1** (Differential Privacy, [DMNS06])**.** A randomized algorithm $\mathcal{A} \colon \mathcal{S}^n \to \mathcal{O}$ is $(\varepsilon, \delta)$-*differentially private* (DP) if for all neighboring datasets $S, S'$ and all measurable $O \subseteq \mathcal{O}$,

$$\Pr[\mathcal{A}(S) \in O] \le e^\varepsilon \Pr[\mathcal{A}(S') \in O] + \delta.$$

Algorithm $\mathcal{A}$ is $\varepsilon$-*differentially private* if it satisfies the definition for $\delta = 0$.

Differential privacy satisfies the following two useful properties.

**Lemma 2.2** (Post-Processing, [DMNS06])**.** *Let $\mathcal{A} \colon \mathcal{S}^n \to \mathcal{O}$ be a randomized algorithm that is $(\varepsilon, \delta)$-differentially private. For every (possibly randomized) $f : \mathcal{O} \to \mathcal{O}'$, $f \circ \mathcal{A}$ is $(\varepsilon, \delta)$-differentially private.*

**Lemma 2.3** (Composition, [DMNS06])**.** *Let $\mathcal{M}_i \colon \mathcal{S}^n \to \mathcal{O}_i$ be an $(\varepsilon_i, \delta_i)$-differentially private algorithm for $i \in [k]$. If $\mathcal{M}_{[k]} \colon \mathcal{S}^n \to \mathcal{O}_1 \times \ldots \times \mathcal{O}_k$ is defined as $\mathcal{M}_{[k]}(S) = (\mathcal{M}_1(S), \ldots, \mathcal{M}_k(S))$, then $\mathcal{M}_{[k]}$ is $(\sum_{i=1}^k \varepsilon_i, \sum_{i=1}^k \delta_i)$-differentially private.*

A common technique that differentially private mechanisms use is to add zero-mean noise of appropriate scale to the quantities computed from the dataset. The scale of the noise depends on the *sensitivity* of the function of the dataset we aim to compute. Intuitively, the sensitivity represents the maximum change that the change of a single data point can incur on the output of the function.

**Definition 2.4** ($\ell_1$- and $\ell_2$-sensitivity)**.** Let $f \colon \mathcal{S}^n \to \mathbb{R}^k$. The $\ell_1$-*sensitivity* of $f$ is

$$\Delta_1 f = \max_{\substack{S, S' \in \mathcal{S}^n \\ S \sim S'}} \|f(S) - f(S')\|_1.$$

Respectively, the $\ell_2$-*sensitivity* of $f$ is $\Delta_2 f = \max_{\substack{S, S' \in \mathcal{S}^n \\ S \sim S'}} \|f(S) - f(S')\|_2$.

Two standard differentially private mechanisms, which are used very often as building blocks, are the *Laplace* and the *Gaussian* Mechanism.

**Lemma 2.5** (Laplace Mechanism, [DMNS06])**.** *Let $f \colon \mathcal{S}^n \to \mathbb{R}$, a dataset $S \in \mathcal{S}^n$, and privacy parameter $\varepsilon$. The* Laplace Mechanism

$$\tilde{f}(S) = f(S) + \mathrm{Lap}(\Delta_1 f / \varepsilon)$$

*is $\varepsilon$-differentially private and with probability at least $1 - \gamma$, $|\tilde{f}(S) - f(S)| \le \frac{\Delta_1 f}{\varepsilon} \ln \frac{1}{\gamma}$.*

**Lemma 2.6** (Gaussian Mechanism, [DKM$^+$06])**.** *Let $f \colon \mathcal{S}^n \to \mathbb{R}^d$, a dataset $S \in \mathcal{S}^n$, and privacy parameters $(\varepsilon, \delta)$. The* Gaussian Mechanism

$$\tilde{f}(S) = f(S) + \mathcal{N}(\mathbf{0}, \sigma^2 \mathbb{I}_{d \times d}), \ \ where \ \sigma = \Delta_2 f \sqrt{2 \ln(5/4\delta)} / \varepsilon$$

*is $(\varepsilon, \delta)$-differentially private and with probability at least $1 - \gamma$, $\|\tilde{f}(S) - f(S)\|_2 \le 4\sigma \sqrt{d \ln \frac{1}{\gamma}}$.*

## 2.2 Useful Facts on Distances Between Multivariate Distributions

We here record some lemmata which will be useful to us, relating total variation distance (equivalently, $L_1$) between the multivariate distributions we consider to the $\ell_2$ distances between their mean vectors.

The first is relatively standard; for the specific constants stated below, it is a direct consequence of [DMR18, Theorem 1.2].

**Fact 2.7.** *Let $\mu, \nu \in \mathbb{R}^d$. Then, the $L_1$ distance between the two multivariate Normal distributions $\mathcal{N}(\mu, \mathbb{I}_{d \times d})$, $\mathcal{N}(\nu, \mathbb{I}_{d \times d})$ satisfies*

$$\frac{1}{100} \cdot \|\mu - \nu\|_2 \leq \|\mathcal{N}(\mu, \mathbb{I}_{d \times d}) - \mathcal{N}(\nu, \mathbb{I}_{d \times d})\|_1 \leq 9 \cdot \|\mu - \nu\|_2 \,.$$

The second relates, similarly, $L_1$ distance between product distributions over $\{\pm 1\}^d$ to the $\ell_2$ distance between their means, however with a caveat – namely, at least one of the distributions needs to be "balanced," i.e., have all its marginals "not-nearly constant."

**Lemma 2.8.** *Fix $\tau \in (0, 1]$, and let $P, Q$ be product distributions over $\{\pm 1\}^d$ with mean vectors $\mu, \nu \in [-1, 1]^d$ such that $-1 + \tau \leq \nu_i \leq 1 - \tau$ for all $i$. Then, the $L_1$ distance between $P$ and $Q$ satisfies*

$$c_\tau \cdot \|\mu - \nu\|_2 \leq \|P - Q\|_1 \leq C_\tau \cdot \|\mu - \nu\|_2 \,,$$

*where $C_\tau, c_\tau > 0$ are two constants depending only on $\tau$. Moreover, one can take $C_\tau = 1/\sqrt{\tau(1 - \frac{\tau}{2})}$.*

*Proof.* The upper bound follows from [CDKS17, Corollary 3.5] (note that their parameterization is for $P, Q$ over $\{0, 1\}^d$). As for the lower bound, it is proven analogously to [KLSU19, Lemma 6.4]; with two main differences. First, their lemma is stated for $\tau = 2/3$, whereas we allow it to be an arbitrarily small constant – the same argument carries through, although at the cost of larger constant factors in the bound. Second, their lemma requires that both $P$ and $Q$ be balanced, whereas we only require one to be balanced. This can be dealt with by noting that if one distribution is not balanced in some coordinate, the difference in means in this coordinate is sufficient to witness a large total variation distance. $\square$

## 3 From Gaussian to Product-of-Bernoulli Testing

In this short section, we provide a simple argument which enables us to transfer all our results on uniformity testing for product-of-Bernoulli distributions (described next, in Section 4) to testing identity of multivariate Normal distributions. Specifically, we analyze a simple reduction which maps a sample from an unknown multivariate normal $\mathcal{N}(\mu, \mathbb{I}_{d \times d})$ to a sample from a product distribution $P_\mu$ on $\{\pm 1\}^d$, such that the standard Normal $\mathcal{N}(\mathbf{0}, \mathbb{I}_{d \times d})$ is mapped to $P_{\mathbf{0}} = \mathcal{U}_d$, while any Normal $\alpha$-far from $\mathcal{N}(\mathbf{0}, \mathbb{I}_{d \times d})$ is mapped to some $P_\mu$ that is $\Omega(\alpha)$-far from $\mathcal{U}_d$.

**Theorem 3.1.** *There exists a function $F \colon \mathbb{R}^d \to \{\pm 1\}^d$ and an absolute constant $c > 0$ such that the following holds. For $\mu \in \mathbb{R}^d$, denote by $P_\mu$ the distribution of $F(X)$ when $X$ is drawn from $\mathcal{N}(\mu, \mathbb{I}_{d \times d})$. Then*

- *If $\mu = \mathbf{0}$, then $P_{\mathbf{0}} = \mathcal{U}_d$ is the uniform distribution on $\{\pm 1\}^d$;*

- $P_\mu$ is a product distribution over $\{\pm 1\}^d$ such that $\|P_\mu - \mathcal{U}_d\|_1 \geq c \cdot \|\mathcal{N}(\mu, \mathbb{I}_{d \times d}) - \mathcal{N}(\mathbf{0}, \mathbb{I}_{d \times d})\|_1$.

*Moreover, $F$ is computable in linear time.*

Before providing the proof of the theorem, we note that, with this reduction, Theorems 1.4 and 1.5 directly follow from their respective counterparts, Theorems 1.1 and 1.2.

*Proof.* The mapping $F \colon \mathbb{R}^d \to \{\pm 1\}^d$ is defined coordinate-wise, by setting $F(x)_i := \operatorname{sgn}(x_i)$ for all $i \in [d]$; thus trivially implying the first item, as well as the time-efficiency statement and the fact that $P_\mu$ is a product distribution. We turn to the remaining part of the second item. Fix any $\mu \in \mathbb{R}^d$, and define for convenience $\alpha := \|\mathcal{N}(\mu, \mathbb{I}_{d \times d}) - \mathcal{N}(\mathbf{0}, \mathbb{I}_{d \times d})\|_1 \in [0, 2]$; by Fact 2.7, we have $\|\mu\|_2 \geq \frac{\alpha}{9}$. For every $i \in [d]$,

$$\mathbb{E}_{X \sim \mathcal{N}(\mu, \mathbb{I}_{d \times d})}[F(X)_i] = 2\mathbb{P}[F(X)_i = 1] - 1 = 2\mathbb{P}[X_i > 0] - 1 = \operatorname{Erfc}(-\mu_i/\sqrt{2}) - 1 = -\operatorname{Erf}(-\mu_i/\sqrt{2}).$$

Therefore, the mean vector $\mu'$ of $P_\mu$ satisfies

$$\|\mu'\|_2^2 = \sum_{i=1}^d \operatorname{Erf}(-\mu_i/\sqrt{2})^2 \geq 0.84^2 \sum_{i=1}^d \min(\mu_i^2/2, 1) = 0.84^2 \min(\|\mu\|_2^2/2, d) \geq 0.84^2 \alpha^2/162 \,,$$

the first inequality by Lemma 3.2 (which we will prove momentarily) and the last by our above lower bound on $\|\mu\|_2$. This shows that $\|\mu'\|_2 > \alpha/12$. Applying finally Lemma 2.8 (with $Q = \mathcal{U}_d$, for which $\tau = 1$), we get existence of an absolute constant $c > 0$ such that $\|P_\mu - \mathcal{U}_d\|_1 \geq c\alpha$, as sought. $\qquad\square$

The above argument relied on a technical lemma about the error function Erf, which we state and prove below.

**Lemma 3.2.** *For all $t \in \mathbb{R}$, $|\operatorname{Erf}(-t)| \geq 0.84 \cdot \min\{|t|, 1\}$.*

*Proof.* Consider the functions $g(t) = (\operatorname{Erf}(-t))^2$ and $f(t) = \frac{g(t)}{t^2}$ for $t > 0$. It holds that

$$g'(t) = 2\operatorname{Erf}(-t) \cdot (\operatorname{Erf}(-t))' = -\frac{4}{\sqrt{\pi}} \exp(-t^2)\operatorname{Erf}(-t). \tag{6}$$

So $g(t)$ is increasing in $(0, \infty)$, since $g'(t) > 0$. Then, for $t > 1$, $g(t) > g(1) = (\operatorname{Erf}(-1))^2 > 0.84^2$. Thus,

$$|\operatorname{Erf}(-t)| > 0.84 \ \forall t > 1. \tag{7}$$

Now consider the function $f(t)$ in $(0, \infty)$.

$$f'(t) = \frac{g'(t)}{t^2} - \frac{2g(t)}{t^3} = -\frac{4}{\sqrt{\pi}t^2} \exp(-t^2)\operatorname{Erf}(-t) - \frac{2}{t^3}(\operatorname{Erf}(-t))^2 \qquad \text{(by (6))}$$

$$= -\frac{4\exp(-t^2)\operatorname{Erf}(-t)}{\sqrt{\pi}t^3}\left(t - \exp(t^2)\int_0^t \exp(-x^2)\,dx\right) \qquad \text{(since } -t < 0)$$

Let us denote $h(t) = t - \exp(t^2)\int_0^t \exp(-x^2)\,dx$ for $t \in [0, \infty)$. It holds that

$$h'(t) = 1 - \left(\exp(t^2)\right)'\int_0^t \exp(-x^2)\,dx - \exp(t^2)\exp(-t^2) = -2t\exp(t^2)\int_0^t \exp(-x^2)\,dx \leq 0.$$

So $h(t)$ is decreasing in $[0, \infty)$ and $h(t) < h(0) = 0$ for $t > 0$. It follows that $f'(t) < 0$, so $f(t)$ is decreasing in $(0, \infty)$. Thus, for $t \in (0, 1]$, $f(t) \geq f(1) = 0.84^2$. Equivalently,

$$|\text{Erf}(-t)| \geq 0.84|t| \ \forall t \in (0, 1]. \tag{8}$$

By inequalities (7) and (8), we conclude that $|\text{Erf}(-t)| \geq 0.84 \cdot \min\{|t|, 1\}$ for $t > 0$ and the claim follows by symmetry. $\qquad\square$

# 4  Uniformity Testing for Product-of-Bernoulli Distributions

In this section we introduce our algorithms for uniformity testing. Both algorithms use a noisy version of the test statistic for uniformity testing introduced in [CDKS17]:

$$T(X) = \sum_{i=1}^{d} (\bar{X}_i^2 - n). \tag{9}$$

The tests differ in the way that they reduce the sensitivity of the test statistic, giving tradeoffs between sample complexity and computational complexity.

## 4.1  A Computationally Inefficient Private Algorithm

In this section we focus on a computationally inefficient algorithm for uniformity testing based on Lipschitz extensions. This algorithm has two main components: a Lipschitz extension that allows us to control the amount of noise added to the test statistic, and an iterative step that rejects a successively larger class of distributions. The key idea is that the statistic $\|\bar{X}\|_2^2$ is related to both the test statistic and the sensitivity of the test statistic. Algorithm 2 proceeds in rounds. Suppose that if $X$ has survived until round $m$ then with high probability $X \in \mathcal{C}(\Delta^{(m)})$. The test statistic $T$ has sensitivity $\Delta^{(m)}$ on $\mathcal{C}(\Delta^{(m)})$. Thus, we can then compute a test statistic $\hat{T}$ that agrees with $T$ on $\mathcal{C}(\Delta^{(m)})$, but has sensitivity $\Delta^{(m)}$ on all datasets, as in the LipschitzExtensionTest (Algorithm 1). The existence of such a Lipschitz extension is guaranteed by the McShane-Whitney extension theorem [McS34]. If $X$ passes the test, then we have a new bound for $\|\bar{X}\|_2^2$, which implies a new bound $\Delta^{(m+1)}$ for the sensitivity of $T(X)$, which we use in the next round. It turns out that this bound is decreasing, so we can recurse until we either conclude that the data is not uniform, or we have a sufficiently small upper bound on the sensitivity $\Delta^*$. With this bound and if $X$ is not rejected in any of the iterations of Algorithm 2, we run one last Lipschitz test with threshold $O(n^2\alpha^2)$ which will determine if the data is uniform or $\alpha$-far from uniform.

Let $P = P_1 \times \cdots \times P_d$ be a product distribution over $\{\pm 1\}^d$, which is specified by its mean vector $p = (p_1, \cdots, p_d) \in [-1, 1]^d$. We will denote the product of $d$ independent copies of the uniform distribution by $\mathcal{U}_d$. We draw samples from distribution $P$ and aim to distinguish between the cases $P = \mathcal{U}_d$ and $\|P - \mathcal{U}_d\|_1 \geq \alpha$ with probability at least 2/3.

The main theorem of this section is the following:

**Theorem 4.1.** *Algorithm 2 is $\varepsilon$-differentially private. Furthermore, Algorithm 2 can distinguish between the cases $P = \mathcal{U}_d$ and $\|P - \mathcal{U}_d\|_1 \geq \alpha$ with probability at least 2/3 and has sample complexity*

$$n = \tilde{O}\left(\frac{d^{1/2}}{\alpha^2} + \frac{d^{1/2}}{\alpha\varepsilon^{1/2}} + \frac{d^{1/3}}{\alpha^{4/3}\varepsilon^{2/3}} + \frac{1}{\alpha\varepsilon}\right).$$

Our test (Algorithm 2) will rely on a private adaptation of the test from [CDKS17]; for this reason, we begin by recalling the guarantees of the test statistic $T$ developed in [CDKS17]. In their work, Canonne *et al.* use Poisson sampling; however the interplay between privacy (which is defined with regard to a fixed set of samples) and Poisson sampling (where the number of samples is itself randomized) is tricky at best. For this reason, we state a version of their result without Poisson sampling, whose proof can be found in Appendix A.

**Lemma 4.2** (Non-private Test Guarantees). *For the test $T$ defined in* (9), *the following hold:*

- *If $P = \mathcal{U}_d$ then $\mathbb{E}[T(X)] = 0$ and $\mathrm{Var}(T(X)) \leq 2n^2 d$.*

- *If $\|P - \mathcal{U}_d\|_1 \geq \alpha$ then $\mathbb{E}[T(X)] > \frac{1}{2}n(n-1)\alpha^2$.*

- *$\mathrm{Var}(T(X)) \leq 2n^2 d + 4n\mathbb{E}[T(X)]$.*

---

**Algorithm 1** LIPSCHITZEXTENSIONTEST

---

**Require:** Sample $X = (X^{(1)}, \ldots, X^{(n)})$. Parameters $\varepsilon, \Delta > 0, \beta \in (0, 1]$.
1: Define the set $\mathcal{C}(\Delta) = \left\{X \in \{\pm 1\}^{n \times d} \mid \forall j \in [n], |\langle X^{(j)}, \bar{X} \rangle| \leq \Delta\right\}$.
2: Let $\hat{T}(\cdot)$ be a $4\Delta$-Lipschitz extension of $T$ from $\mathcal{C}(\Delta)$ to all of $\{\pm 1\}^{n \times d}$.
3: Sample noise $r \sim \mathrm{Lap}(4\Delta/\varepsilon)$ and let $z \leftarrow \hat{T}(X) + r$.
4: **if** $z > 10n\sqrt{d} + 4\Delta \ln(1/\beta)/\varepsilon$ **then**
5:     **return** reject.
6: **return** accept.

---

**Algorithm 2** Private Uniformity Testing via Lipschitz Extension

---

**Require:** Sample $X = (X^{(1)}, \ldots, X^{(n)}) \in \{\pm 1\}^{n \times d}$ drawn from $P^n$. Parameters $\varepsilon, \alpha, \beta > 0$.
1: Let $M \leftarrow \lceil \log n \rceil$, $\varepsilon' \leftarrow \varepsilon/M$, $\beta \leftarrow 1/(10n)$.
2: Let $\Delta^{(1)} \leftarrow nd$ and $\Delta^* \leftarrow 1000 \max(d, \sqrt{nd}, \ln(1/\beta)/\varepsilon') \cdot \ln(1/\beta)$.
3: **for** $m \leftarrow 1$ to $M - 1$ **do**
4:     **if** $\Delta^{(m)} \leq \Delta^*$ **then**
5:        Let $\Delta^{(M)} \leftarrow \Delta^{(m)}$ and exit the loop.
6:     **else**
7:        **if** LIPSCHITZEXTENSIONTEST$(X, \varepsilon', \Delta^{(m)}, \beta)$ returns reject **then**
8:           **return** reject
9:        Set $\Delta^{(m+1)} \leftarrow 11\left(d + \sqrt{nd} + \frac{\Delta^{(m)}}{n\varepsilon'} + \sqrt{\frac{\Delta^{(m)}}{\varepsilon'}}\right) \ln \frac{1}{\beta}$.
10: Define the set $\mathcal{C}(\Delta^{(M)}) = \left\{X \in \{\pm 1\}^{n \times d} \mid \forall j \in [n], |\langle X^{(j)}, \bar{X} \rangle| \leq \Delta^{(M)}\right\}$.
11: Let $\hat{T}(\cdot)$ be a $4\Delta^{(M)}$-Lipschitz extension of $T$ from $\mathcal{C}(\Delta^{(M)})$ to all of $\{\pm 1\}^{n \times d}$.
12: Sample noise $r \sim \mathrm{Lap}(4\Delta^{(M)}/\varepsilon')$ and let $z \leftarrow \hat{T}(X) + r$.
13: **if** $z > \frac{1}{4}n(n-1)\alpha^2$ **then return** reject
14: **return** accept.

---

We first focus on the privacy guarantee of Theorem 4.1. The mechanism is the composition of $M$ invocations of the Laplacian mechanism in lines 3 and line 12. Our privacy proof is based on the existence of Lipschitz extensions, as established by the following theorem.

**Lemma 4.3** (McShane–Whitney extension theorem, [McS34])**.** *Let $\varphi \colon \mathcal{C} \to \mathbb{R}$ be a real-valued, $L$-Lipschitz function defined on a subset $\mathcal{C}$ of a metric space $\mathcal{M}$. Then, there exists an $L$-Lipschitz map $\hat{\varphi} \colon \mathcal{M} \to \mathbb{R}$ that extends $\varphi$, that is, $\varphi(x) = \hat{\varphi}(x) \ \forall x \in \mathcal{C}$.*

In this work, we will invoke the McShane–Whitney extension theorem with the metric space $\mathcal{M}$ being the space of databases with the metric induced by the neighboring relation.

Let us define for our dataset $X$ and any $\Delta > 0$,

$$\mathcal{C}(\Delta) = \left\{ X \in \{\pm 1\}^{n \times d} \mid \forall j \in [n], \ |\langle X^{(j)}, \bar{X} \rangle| \leq \Delta \right\}.$$

The main element that we need for the privacy proof is the bound on the sensitivity of $T$ on $\mathcal{C}(\Delta^{(m)})$ for all $m \in [M]$. This would ensure that $T$ is $4\Delta^{(m)}$-Lipschitz on $\mathcal{C}(\Delta^{(m)})$, so the $4\Delta^{(m)}$-Lipschitz extensions $\hat{T}$ exist in all the rounds and lines 3 and 12 add enough noise to maintain privacy. Note that the algorithm would be private regardless of the choice of values $\Delta^{(m)}$.

**Lemma 4.4** (Sensitivity of T)**.** *For any bound $\Delta > 0$, for two neighboring datasets $X, X' \in \mathcal{C}(\Delta)$, $|T(X) - T(X')| \leq 4\Delta$.*

*Proof.* Without loss of generality, assume that $X$ and $X'$ differ on the $n$-th sample. Then, we can write $X = (X^{(1)}, \ldots, X^{(n)})$ and $X' = (X^{(1)}, \ldots, X'^{(n)})$. We can now calculate the difference:

$$
\begin{aligned}
T(X) - T(X') &= \sum_{i=1}^{d} \left[ \left( \sum_{j=1}^{n} X_i^{(j)} \right)^2 - \left( \sum_{j=1}^{n} X_i'^{(j)} \right)^2 \right] \\
&= \sum_{i=1}^{d} \left[ \left( \sum_{j=1}^{n} X_i^{(j)} - \sum_{j=1}^{n} X_i'^{(j)} \right) \cdot \left( \sum_{j=1}^{n} X_i^{(j)} + \sum_{j=1}^{n} X_i'^{(j)} \right) \right] \\
&= \sum_{i=1}^{d} \left[ \left( X_i^{(n)} - X_i'^{(n)} \right) \cdot \left( \sum_{j=1}^{n} X_i^{(j)} + \sum_{j=1}^{n} X_i'^{(j)} \right) \right] \\
&= \sum_{i=1}^{d} \left[ X_i^{(n)} \cdot \left( 2\sum_{j=1}^{n} X_i^{(j)} + X_i'^{(n)} - X_i^{(n)} \right) - X_i'^{(n)} \cdot \left( 2\sum_{j=1}^{n} X_i'^{(j)} + X_i^{(n)} - X_i'^{(n)} \right) \right] \\
&= 2\langle X^{(n)}, \bar{X} \rangle - 2\langle X'^{(n)}, \bar{X}' \rangle + \sum_{i=1}^{d} (X_i^{(n)} + X_i'^{(n)}) \cdot (X_i'^{(n)} - X_i^{(n)})
\end{aligned}
$$

Therefore, we have

$$T(X) - T(X') = 2\langle X^{(n)}, \bar{X} \rangle - 2\langle X'^{(n)}, \bar{X}' \rangle + \|X'^{(n)}\|_2^2 - \|X^{(n)}\|_2^2. \tag{10}$$

Observe that, because $X_i^{(n)}, X_i'^{(n)}$ are in $\{\pm 1\}$, we will have $\|X'^{(n)}\|_2^2 = \|X^{(n)}\|_2^2 = d$, leading to $T(X) - T(X') \leq 2(\langle X^{(n)}, \bar{X} \rangle - \langle X'^{(n)}, \bar{X}' \rangle)$, as stated earlier, in (4). This readily implies the bound

$$|T(X) - T(X')| \leq 2|\langle X^{(n)}, \bar{X} \rangle| + 2|\langle X'^{(n)}, \bar{X}' \rangle|.$$

Since $X, X' \in \mathcal{C}(\Delta)$, we know that $\left| \langle X^{(n)}, \bar{X} \rangle \right| \leq \Delta$ and $\left| \langle X'^{(n)}, \bar{X}' \rangle \right| \leq \Delta$. It follows that $|T(X) - T(X')| \leq 4\Delta$. $\qquad\square$

**Lemma 4.5** (Privacy)**.** LipschitzExtensionTest$(X, \varepsilon, \Delta, \beta)$ *is $\varepsilon$-differentially private.*

*Proof.* LIPSCHITZEXTENSIONTEST$(X, \varepsilon, \Delta, \beta)$ only accesses the data $X$ via the Laplace Mechanism in step 3, which is $\varepsilon$-DP by Lemma 2.5. $\qquad \square$

**Lemma 4.6** (Privacy). *Algorithm 2 is $\varepsilon$-differentially private.*

*Proof.* Algorithm 2 is a composition of $M$ invocations of LIPSCHITZEXTENSIONTEST with DP parameter $\varepsilon/M$, and of the Laplace mechanism in step 12. By Lemmas 2.5 and 4.5, each of these steps is individually $\varepsilon' = (\varepsilon/M)$-differentially private. Since these are the only steps that access the data, the privacy guarantee follows from Lemmas 2.3 and 2.2. $\qquad \square$

Now we have established the privacy of Algorithm 2, we turn our attention to the utility guarantee. First, we prove a claim that will help us determine how the bound on the sensitivity of the statistic decreases in each round.

**Lemma 4.7.** *If $X$ is drawn i.i.d. from a product distribution, then, with probability at least $1 - 2n\beta$,*

$$\forall x \in X, \ |\langle x, \bar{X} \rangle| \leq \frac{\|\bar{X}\|_2^2}{n} + \sqrt{2}\|\bar{X}\|_2 \sqrt{\ln(1/\beta)}. \tag{11}$$

*Proof.* Since the $X^{(j)}$'s are i.i.d., if we condition on the sum $\bar{X} = \sum_{j=1}^{n} X^{(j)}$ then $\mathbb{E}[X^{(k)} \mid \bar{X}] = \bar{X}/n$ by symmetry. This implies that $\mathbb{E}[\langle X^{(k)}, \bar{X} \rangle \mid \bar{X}] = \langle \mathbb{E}[X^{(k)} \mid \bar{X}], \bar{X} \rangle = \frac{1}{n}\|\bar{X}\|_2^2$.

Moreover, we can rewrite the inner product as $\langle X^{(k)}, \bar{X} \rangle = \sum_{i=1}^{d} X_i^{(k)} \bar{X}_i = \sum_{i=1}^{d} Y_i^{(k)}$, where, for all $i \in [d]$ and $k \in [n]$, we have

$$Y_i^{(k)} \mid \bar{X} = X_i^{(k)} \bar{X}_i \mid \bar{X} = \begin{cases} +\bar{X}_i, & \text{w.p. } \frac{1+\bar{X}_i/n}{2} \\ -\bar{X}_i, & \text{w.p. } \frac{1-\bar{X}_i/n}{2} \end{cases}. \tag{12}$$

That is, for all $k \in [n]$, the random variables $Y_i^{(k)} \mid \bar{X}$ for $i \in [d]$ are independent with each $Y_i^{(k)} \in \{-|\bar{X}_i|, |\bar{X}_i|\}$. By Hoeffding's inequality,

$$\Pr\left[|\langle X^{(k)}, \bar{X} \rangle - \|\bar{X}\|_2^2/n| \geq t \mid \bar{X}\right] \leq 2\exp\left(-\frac{t^2}{2\|\bar{X}\|_2^2}\right) \tag{13}$$

Denote by $E_{\bar{X}}$ the event that there exists $k \in [n]$ such that $\left|\langle X^{(k)}, \bar{X} \rangle\right| \geq \|\bar{X}\|_2^2/n + \sqrt{2}\|\bar{X}\|_2 \sqrt{\ln(1/\beta)}$. Then, by union bound and inequality (13), $\Pr\left[E_{\bar{X}} \mid \bar{X}\right] \leq 2n\beta$. Taking the expectation of the latter probability and since the bound holds for any $\bar{X}$, we have that

$$\Pr[E_{\bar{X}}] = \mathbb{E}_{\bar{X}}[\mathbf{1}_{E_{\bar{X}}}] = \mathbb{E}_{\bar{X}}[\mathbb{E}[\mathbf{1}_{E_{\bar{X}}} \mid \bar{X}]] = \mathbb{E}_{\bar{X}}\left[\Pr\left[E_{\bar{X}} \mid \bar{X}\right]\right] \leq 2n\beta.$$

as claimed. $\qquad \square$

In what follows, as in Algorithm 2, we let $\Delta^* = 1000 \max(d, \sqrt{nd}, M \ln(1/\beta)/\varepsilon) \cdot \ln(1/\beta)$, where $\beta = 1/(10n)$ and $M = \lceil \log(n) \rceil$.

Based on the previous lemma, we now analyze the key subroutine, LIPSCHITZEXTENSIONTEST. We already established its privacy guarantee in Lemma 4.5. We will prove two additional properties:

1. if $X$ is drawn from a product distribution and, for some $\Delta$, LIPSCHITZEXTENSIONTEST returns accept, then $X \in \mathcal{C}(\Delta')$, where $\Delta' \ll \Delta$ (Lemma 4.8);

2. if $X$ is drawn from the uniform distribution, LIPSCHITZEXTENSIONTEST will return accept (Lemma 4.9).

**Lemma 4.8** (Sensitivity reduction). *Fix some $\Delta \geq 0$, and let $\Delta' = 11(d + \sqrt{nd} + \Delta/(n\varepsilon) + \sqrt{\Delta/\varepsilon}) \ln(1/\beta)$. If the following four conditions hold:*

*(i) $X$ is drawn from a product distribution,*

*(ii) $X$ satisfies (11),*

*(iii) $X \in \mathcal{C}(\Delta)$, and*

*(iv) LIPSCHITZEXTENSIONTEST$(X, \varepsilon, \Delta, \beta)$ returns accept,*

*then $X \in \mathcal{C}(\Delta')$ with probability at least $1 - \beta$.*

*Proof.* Let $\Delta \geq 0$, and assume $X$ is drawn from a product distribution and satisfies $X \in \mathcal{C}(\Delta)$ and (11). Since $X \in \mathcal{C}(\Delta)$, the Lipschitz extension $\hat{T}$ coincides with $T$ on line 2, and therefore $\hat{T}(X) = T(X) = \|\bar{X}\|_2^2 - nd$. Since LIPSCHITZEXTENSIONTEST returned accept, we further know that

$$T(X) + r \leq 10n\sqrt{d} + \frac{4\Delta}{\varepsilon} \ln \frac{1}{\beta}$$

where $r$ is a $\mathrm{Lap}(4\Delta/\varepsilon)$ random variable, and therefore with probability at least $1 - \beta$ has magnitude at most $4\Delta \ln(1/\beta)/\varepsilon$ by Lemma 2.5. Thus, with probability $1 - \beta$,

$$\|\bar{X}\|_2^2 \leq 10n\sqrt{d} + 8\Delta \ln(1/\beta)/\varepsilon + nd \leq 11(nd + \Delta \ln(1/\beta)/\varepsilon).$$

Substituting the bound on $\|\bar{X}\|_2^2$ in inequality (11), we get that with probability at least $1 - \beta$, for all $x \in X$

$$|\langle x, \bar{X}\rangle| \leq 11 \left( d + \sqrt{nd \ln \frac{1}{\beta}} + \frac{\Delta}{n\varepsilon} \ln \frac{1}{\beta} + \sqrt{\frac{\Delta}{\varepsilon}} \ln \frac{1}{\beta} \right) \leq 11 \left( d + \sqrt{nd} + \frac{\Delta}{n\varepsilon} + \sqrt{\frac{\Delta}{\varepsilon}} \right) \ln \frac{1}{\beta} \quad (14)$$

This concludes the proof, as the RHS corresponds to our setting of $\Delta'$. $\qquad\square$

**Lemma 4.9.** *If (i) $X$ is drawn from the uniform distribution $\mathcal{U}_d$, satisfies $T(X) \leq 10n\sqrt{d}$, and (11), and (ii) $\Delta \geq \Delta^*$, then LIPSCHITZEXTENSIONTEST$(X, \varepsilon, \Delta, \beta)$ returns accept with probability at least $1 - \beta$.*

*Proof.* Since $T(X) = \|\bar{X}\|_2^2 - nd$, our assumption implies that $\|\bar{X}\|_2^2 \leq nd + 10n\sqrt{d} \leq 11nd$. Plugging this in (11), we have that

$$|\langle x, \bar{X}\rangle| \leq \frac{11nd}{n} + 2\sqrt{nd \ln(1/\beta)} \leq 22 \max(d, \sqrt{nd \ln(1/\beta)}) \leq \Delta^* \leq \Delta$$

so that $X \in \mathcal{C}(\Delta)$. But when this happens, we have $\hat{T}(X) = T(X)$ in line 2 of LIPSCHITZEXTENSIONTEST, and thus we can analyze what happens in line 4 by bounding $T(X) + r$, which by our assumption is at most $10n\sqrt{d} + r$. By Lemma 2.5, with probability at least $1 - \beta$ we have $|r| \leq 4\Delta \ln(1/\beta)/\varepsilon$. Whenever this holds, the algorithm outputs accept. $\qquad\square$

With the analysis of this subroutine being done, we turn to the proof of Theorem 4.1. Unlike the privacy guarantee, this only needs to hold for datasets drawn from a product distribution. The crux of the proof can be summarized as follows:

1. If $X$ is drawn i.i.d. from a product distribution and passes line 3 in round $m$, then it belongs in $\mathcal{C}(\Delta^{(m+1)})$ with high probability. Thus in every round that the dataset has not been rejected, we have that with high probability $\hat{T}(X) = T(X)$, so we are just running a noisy version of the non-private test. Further with each iteration we refine our bound on the sensitivity required.

2. If $P = \mathcal{U}_d$, then with high probability $X$ passes all the steps in the loop at line 3.

3. The number of rounds $M$ is sufficient to guarantee that the sensitivity and thus the amount of noise added to $\hat{T}(X)$ in the last test in line 12 is small enough that one distinguishes between the two hypotheses with the desired sample complexity.

For our argument, we will need to show the noise added in line 12 of Algorithm 2 is small enough that we can still distinguish between the two hypotheses. The magnitude of that noise depends on the sensitivity, $\Delta^{(M)}$, of $T(X)$ restricted to the set $\mathcal{C}(\Delta^{(M)})$. The next lemma upper bounds $\Delta^{(M)}$.

**Lemma 4.10.** *For $M = \lceil \log(n) \rceil$, $\varepsilon' = \varepsilon/M$, and $n = \Omega(\log(1/\beta)/\varepsilon')$,*

$$\Delta^{(M)} \leq \Delta^* = 1000 \max(d, \sqrt{nd}, \ln(1/\beta)/\varepsilon') \cdot \ln(1/\beta).$$

*Proof.* Let us recall the update rule for the bound on the sensitivity of the statistic $T(X)$: $\Delta^{(1)} = nd$, and, for $m \geq 1$,

$$\Delta^{(m+1)} \leftarrow 11 \left( d + \sqrt{nd} + \frac{\Delta^{(m)}}{n\varepsilon'} + \sqrt{\frac{\Delta^{(m)}}{\varepsilon'}} \right) \ln \frac{1}{\beta}.$$

If at any point in the iteration $\Delta^{(m)} \leq \Delta^*$, then we set $\Delta^{(M)} = \Delta^{(m)}$ and exit the loop, which trivially proves the lemma.
Otherwise, assume that $\Delta^{(m)} > 1000 \max(d, \sqrt{nd}, \ln(1/\beta)/\varepsilon') \ln(1/\beta)$ for some $m$. Then,

$$
\begin{aligned}
\Delta^{(m+1)} &< 11 \left( \frac{\Delta^{(m)}}{1000 \ln(1/\beta)} + \frac{\Delta^{(m)}}{1000 \ln(1/\beta)} + \frac{\Delta^{(m)}}{n\varepsilon'} + \sqrt{\frac{\Delta^{(m)}}{\varepsilon'}} \right) \ln \frac{1}{\beta} \\
&\leq 11 \left( \frac{\Delta^{(m)}}{1000} + \frac{\Delta^{(m)}}{1000} + \frac{\Delta^{(m)}}{1000} + \sqrt{\frac{(\Delta^{(m)})^2}{1000 \ln^2 \frac{1}{\beta}}} \ln \frac{1}{\beta} \right) \\
&\leq \frac{385}{1000} \Delta^{(m)}
\end{aligned}
$$

which is strictly less that $\Delta^{(m)}/2$. This implies that, for $M = \lceil \log(\Delta^{(1)}/\Delta^*) \rceil$ rounds, one must have $\Delta^{(M)} \leq \Delta^*$. Since $\Delta^* \geq d$, $M = \lceil \log(n) \rceil$ rounds suffice. $\qquad \square$

We now complete our proof, showing that the last test will give the expected guarantees, based on all the previous lemmas.

*Proof of Theorem 4.1.* The privacy guarantee was established in Lemma 4.6. It remains to prove that if $P = \mathcal{U}_d$ then Algorithm 2 outputs accept with probability at least $2/3$ (*completeness*) and that if $\|P - \mathcal{U}_d\|_1 \geq \alpha$ then Algorithm 2 outputs reject with probability at least $2/3$ (*soundness*). Before starting, note that given $\Delta^{(1)} = nd$, we have $\mathcal{C}(\Delta^{(1)}) = \{\pm 1\}^{n \times d}$, so the guarantee that $X \in \mathcal{C}(\Delta^{(1)})$ is automatic.

**Completeness:** Suppose $X$ is drawn from $\mathcal{U}_d$. By Lemma 4.2, Chebyshev's inequality implies that

$$\Pr[T(X) \geq 10n\sqrt{d}] \leq \frac{2n^2 d}{100n^2 d} = \frac{1}{50}.$$

Moreover, since $\mathcal{U}_d$ is *a fortiori* a product distribution, we get that $X$ satisfies (11) with probability at least $1 - 2n\beta$. We hereafter assume both of these events hold.

By Lemma 4.9 and a union bound over the $M - 1$ calls to LipschitzExtensionTest in line 7, we get that with probability at least $1 - M\beta$ every call returns accept. Assume this is the case. Invoking now Lemma 4.8, this implies that for all $m$, $X \in \mathcal{C}(\Delta^{(m)})$ except with probability at most $M\beta$, and in particular $X \in \mathcal{C}(\Delta^{(M)})$.

In this case, we have $\hat{T}(X) = T(X)$ in line 12. It remains to show that with high probability, $z = T(X) + r \leq \frac{1}{4}n(n-1)\alpha^2$ in line 13. Since $r \sim \mathrm{Lap}(4\Delta^{(M)}/\varepsilon')$, by Lemma 2.5, it holds that with probability at least $49/50$, $|r| \leq 4\Delta^{(M)}(\ln 50)/\varepsilon'$. Again, condition on this, and suppose for now that $n$ is such that

$$|r| \leq \frac{4\Delta^{(M)} \ln 50}{\varepsilon'} \leq \frac{n(n-1)\alpha^2}{8}. \tag{15}$$

Then, we have, by Lemma 4.2 and Chebyshev's inequality, and recalling that $\hat{T}(X) = T(X)$, we can bound the probability that Algorithm 2 rejects in line 12 as

$$\Pr\left[z > \frac{n(n-1)\alpha^2}{4}\right] = \Pr\left[T(X) > \frac{n(n-1)\alpha^2}{4} - r\right] \leq \Pr\left[T(X) > \frac{n(n-1)\alpha^2}{8}\right]$$
$$\leq \frac{\mathrm{Var}(T(X))}{(n(n-1)\alpha^2/8)^2} \leq \frac{128d}{(n-1)^2\alpha^4},$$

which is at most $1/50$ for $n = \Omega(d^{1/2}/\alpha^2)$. Therefore, by an overall union bound, we get that the algorithm rejects with probability at most

$$\frac{1}{50} + 2n\beta + M\beta + M\beta + \frac{1}{50} + \frac{1}{50} \leq \frac{3}{50} + 4n\beta \leq 1/3$$

the last inequality by our setting of $\beta = 1/(10n)$.

To finish the proof of correctness in the completeness case, it remains to determine the constraint on $n$ for inequality (15) to hold. By Lemma 4.10 (and our setting of $\Delta^*$), it is enough to have,

$$n^2\alpha^2 = \Omega\left(\frac{\Delta^*}{\varepsilon'}\right) = \Omega\left(\frac{\max(d, \sqrt{nd}, \log(1/\beta)/\varepsilon') \cdot \log(1/\beta)}{\varepsilon'}\right).$$

Recalling our choice of $\varepsilon' = \varepsilon/M$, $\beta = 1/(10n)$, and $M = \lceil \log(n) \rceil$, it suffice to choose

$$n = \tilde{\Omega}\left(\frac{d^{1/2}}{\alpha\varepsilon^{1/2}} + \frac{d^{1/3}}{\alpha^{4/3}\varepsilon^{2/3}} + \frac{1}{\alpha\varepsilon}\right). \tag{16}$$

**Soundness:** Suppose $X$ is drawn from a product distribution which is $\alpha$-far from uniform. By Lemma 4.7, $X$ satisfies (11) with probability at least $1 - 2n\beta$: we hereafter condition on this event. Suppose further that the algorithm does not return reject in any of the $M - 1$ calls to LIPSCHITZEXTENSIONTEST in line 7 (otherwise, we are done): we will show that it will then output reject after line 13 with high probability. By Lemma 4.8 and an immediate induction, the above implies that, with probability at least $1 - M\beta$, $X \in \mathcal{C}(\Delta^{(m)})$ for all $m$, and in particular $X \in \mathcal{C}(\Delta^{(M)})$ – so that $\hat{T}_M(X) = T(X)$. We once more assume this event holds.

Similar to the completeness proof (and relying on (15)), with probability at least $49/50$, $|r| \leq \frac{n(n-1)\alpha^2}{8}$. Conditioning on this, we then have, by Lemma 4.2 and Chebyshev's inequality, that the algorithm outputs accept in the last threshold test of line 7 with probability at most

$$\Pr\left[z \leq \frac{n(n-1)\alpha^2}{4}\right] = \Pr\left[\hat{T}_M(X) \leq \frac{n(n-1)\alpha^2}{4} - r\right] \leq \Pr\left[T(X) \leq \frac{3n(n-1)\alpha^2}{8}\right]$$

$$\leq \Pr\left[T(X) \leq \frac{3\mathbb{E}[T(X)]}{4}\right] \leq \frac{16 \cdot \mathrm{Var}(T(X))}{\mathbb{E}[T(X)]^2}$$

$$\leq \frac{64d}{(n-1)^2\alpha^4} + \frac{128}{(n-1)\alpha^2} .$$

which less than $1/50$ for $n = \Omega\left(d^{1/2}/\alpha^2\right)$. By an overall union bound, this means the algorithm outputs accept with probability at most

$$\frac{1}{50} + 2n\beta + M\beta + \frac{1}{50} + \frac{1}{50} \leq 1/3$$

as desired.

We conclude that if

$$n = \tilde{\Omega}\left(\frac{d^{1/2}}{\alpha^2} + \frac{d^{1/2}}{\alpha\varepsilon^{1/2}} + \frac{d^{1/3}}{\alpha^{4/3}\varepsilon^{2/3}} + \frac{1}{\alpha\varepsilon}\right) \tag{17}$$

then Algorithm 2 can distinguish between the cases $P = \mathcal{U}_d$ and $\|P - \mathcal{U}_d\|_1 \geq \alpha$ with probability at least $2/3$. This concludes the proof of Theorem 4.1. $\qquad\square$

## 4.2 A Computationally Efficient Private Algorithm

We next turn our attention to the computationally efficient algorithm, Algorithm 3. The main focus in this section will be a computationally efficient analogue of a single Lipschitz extension step from Algorithm 2. In this algorithm, rather than reducing the sensitivity iteratively as in Algorithm 1, we perform a single preconditioning step to initially reduce the sensitivity.

The preconditioning first checks that the bias of each individual coordinate is not too large (line 3). Next, recall that since the sensitivity of $T(X)$ depends on the bound on $|\langle x, \bar{X}\rangle|$, we want this inner product to be small for all $x \in X$. If $P = \mathcal{U}_d$ then with probability $1 - \delta$, $\forall j \in [n]$,

$$|\langle X^{(j)}, \bar{X}\rangle| \leq \Delta_\delta, \tag{18}$$

for $\Delta_\delta$ as defined in Algorithm 3. This is the property that we want our dataset to satisfy and it resembles the condition we put on the sets $\mathcal{C}$ of the previous section. If any data point in $X$ does not satisfy the inner product condition in (18), we can reject the dataset. Denote by $\mathcal{C}' \in \{\pm 1\}^{n \times d}$ the set of datasets for which all points satisfy (18). Algorithm 3 proceeds by first

attempting to verify membership in $\mathcal{C}'$ by privately counting the number of datapoints that violate condition (18) when compared to a private version of $\bar{X}$, called $\tilde{X}$. We reject if the private test determines with high probability that there is a non-zero number of such points (lines 4-7). If $X$ survives the preconditioning step then Algorithm 3 attempts to ensure that $X \in \mathcal{C}'$ by replacing data points that do not satisfy (18) (again compared to a noisy $\bar{X}$) with draws from the uniform distribution (steps 8-12). Since $X$ was already close to $\mathcal{C}'$, not many data points are changed so the resulting dataset, $\hat{X}$, still "looks like" $X$ but now has bounded inner products with high probability. The Lipschitz extension $\hat{T}(X)$ can then be replaced with $T(\hat{X})$.

Algorithm 3 has a higher sample complexity than Algorithm 2. This comes from two places. Firstly, $|\langle X^{(j)}, \bar{X}\rangle|$ can be an additive factor of $d\log(1/\delta)/\varepsilon$ larger than $|\langle X^{(j)}, \tilde{X}\rangle|$, which is the quantity we actually test. Moreover, when counting the number of points that violate condition (18), we again incur an error of order $d\log(1/\delta)/\varepsilon$, which means that $X$ and $\hat{X}$ may differ more than we expect. This results in a $\sqrt{d\log(1/\delta)}/(\varepsilon\alpha)$ term that was not present in the sample complexity of the inefficient algorithm. Note that the recursion of Algorithm 2 decreases the bound for the corresponding $\Delta_\delta$ parameter but not the overhead of these steps, which dominate the component of the sample complexity due to privacy for Algorithm 3.

Unlike Algorithm 2, which satisfied pure differential privacy, Algorithm 3 as written will only satisfy approximate differential privacy. The $\delta$ term arises since there is a small probability that the projection $\hat{X}$ could fail to be in $\mathcal{C}'$, in which case the amount of noise added to $T(\hat{X})$ is insufficient for privacy. However, for hypothesis tests with constant error probabilities, sample complexity bounds for differential privacy are equivalent, up to constant factors, to sample complexity bounds for other notions of distributional algorithmic stability, such as $(\varepsilon, \delta)$-DP [DKM$^+$06], concentrated DP [DR16, BS16], KL- and TV-stability [WLF16, BNS$^+$16] (see,e.g., [ASZ18, Lemma 5]). The transformation of a test from $(\varepsilon, \delta)$-DP to $O(\varepsilon)$-DP given in [ASZ18] is efficient so Algorithm 3 implies the existence of an efficient pure DP algorithm with the same sample complexity up to constant factors.

**Theorem 4.11.** *Algorithm 3 is $(4\varepsilon, 14\delta)$-differentially private and distinguishes between the cases $P = \mathcal{U}_d$ versus $\|P - \mathcal{U}_d\|_1 \geq \alpha$ with probability at least $2/3$, having sample complexity*

$$n = \tilde{O}\left(\frac{d^{1/2}}{\alpha^2} + \frac{d^{1/2}}{\alpha\varepsilon}\right).$$

We will focus first on proving that Algorithm 3 is $(4\varepsilon, 14\delta)$-differentially private. The majority of the work in the proof will involve showing that with high probability $\hat{X}$ satisfies a property similar to (18). Since this relies on previous steps, the privacy proof is not simply an application of the composition theorem for differential privacy. We first show that lines 8-12, which replace elements of $X$ with draws from the uniform distribution, result in new elements that satisfy (18).

**Lemma 4.12.** *Let $X$ be a dataset that passes line 3 of Algorithm 3. Suppose $U^{(j)} \sim \mathcal{U}_d$ for $j \in [n]$. Then with probability $1 - 3\delta$, for all $j \in [n]$,*

$$\left|\langle U^{(j)}, \bar{X}\rangle\right| \leq \Delta_\delta.$$

*Proof.* If $X$ passes line 3, we have that $\max_{i\in[d]} |\bar{X}_i| \leq \sqrt{2n \ln \frac{d}{\delta}} + \frac{2}{\varepsilon}\ln\frac{1}{\delta} + |r_1|$. By Lemma 2.5, with probability $1 - \delta$,

$$\max_{i\in[d]} |\bar{X}_i| \leq \sqrt{2n \ln \frac{d}{\delta}} + \frac{4}{\varepsilon}\ln\frac{1}{\delta}.$$

**Algorithm 3** Efficient Private Uniformity Testing
***
**Require:** Sample $X = (X^{(1)}, \ldots, X^{(n)}) \in \{\pm 1\}^{n \times d}$ drawn from $P^n$. Parameters $\varepsilon, \delta, \alpha > 0$.
1: Let $\bar{X} \leftarrow \sum_{j=1}^{n} X^{(j)}$.

**Stage 1: Pre-processing**

2: Let $r_1 \sim \text{Lap}(2/\varepsilon)$ and $z_1 \leftarrow \max_{i \in [d]} |\bar{X}_i| + r_1$.

3: **if** $z_1 > \sqrt{2n \ln \frac{d}{\delta}} + \frac{2}{\varepsilon} \ln \frac{1}{\delta}$ **then return** reject.

4: Let $\tilde{X} \leftarrow \bar{X} + R$, where $R \sim \mathcal{N}(0, \sigma^2 \mathbb{I}_{d \times d})$ and $\sigma = \frac{\sqrt{8d \ln(5/4\delta)}}{\varepsilon}$.

5: Let $\Delta_\delta \leftarrow 16 \left( d \ln \frac{d}{\delta} + \frac{d}{n\varepsilon^2} \ln^2 \frac{1}{\delta} + \sqrt{nd} \sqrt{\ln \frac{d}{\delta} \cdot \ln \frac{n}{\delta}} + \frac{\sqrt{d}}{\varepsilon} \ln \frac{1}{\delta} \sqrt{\ln \frac{n}{\delta}} \right)$.

6: Let $r_2 \sim \text{Lap}(1/\varepsilon)$ and $z_2 \leftarrow |\{j \in [n] : |\langle X^{(j)}, \tilde{X} \rangle| > \Delta_\delta + \frac{4d}{\varepsilon} \sqrt{\ln \frac{5}{4\delta} \cdot \ln \frac{n}{\delta}} \}| + r_2$ .

7: **if** $z_2 > \frac{\ln(1/\delta)}{\varepsilon}$ **then return** reject.

**Stage 2: Filtering**

8: **for** $j = 1, \ldots, n$ **do**
9: $\quad$ **if** $|\langle X^{(j)}, \tilde{X} \rangle| > \Delta_\delta + \frac{4d}{\varepsilon} \sqrt{\ln \frac{5}{4\delta} \cdot \ln \frac{n}{\delta}}$ **then**
10: $\quad\quad$ $\hat{X}^{(j)} \leftarrow U^{(j)}$ where $U^{(j)} \sim \mathcal{U}_d$
11: $\quad$ **else**
12: $\quad\quad$ $\hat{X}^{(j)} \leftarrow X^{(j)}$

**Stage 3: Noise addition and thresholding**

13: Let $r_3 \sim \text{Lap} \left( \left( 4\Delta_\delta + \frac{48d}{\varepsilon} \sqrt{\ln \frac{5}{4\delta} \cdot \ln \frac{n}{\delta}} \right)/\varepsilon \right)$ and $z_3 \leftarrow T(\hat{X}) + r_3$.

14: **if** $z_3 > \frac{1}{4} n(n-1)\alpha^2$ **then return** reject
15: **return** accept.
***

Thus, if $x \sim \mathcal{U}_d$, we have $|\mathbb{E}\langle x, \bar{X} \rangle| = 0$, and, for all $i \in [d]$,

$$|x_i \cdot \bar{X}_i| \le \sqrt{2n \ln \frac{d}{\delta}} + \frac{4}{\varepsilon} \ln \frac{1}{\delta}.$$

Now, using Hoeffding's inequality and setting

$$t = \sqrt{2d \ln \left( \frac{n}{\delta} \right)} \left( \sqrt{2n \ln \frac{d}{\delta}} + \frac{4}{\varepsilon} \ln \frac{1}{\delta} \right)$$

we have that with probability $1 - \frac{2\delta}{n}$, it holds that

$$|\langle x, \bar{X} \rangle| \le |\mathbb{E}\langle x, \bar{X} \rangle| + t \le 8 \left( \sqrt{nd} \sqrt{\ln \frac{d}{\delta} \cdot \ln \frac{n}{\delta}} + \frac{\sqrt{d}}{\varepsilon} \ln \frac{1}{\delta} \sqrt{\ln \frac{n}{\delta}} \right) \le \Delta_\delta.$$

By union bound, for all $j \in [n]$, with probability $1 - 3\delta$, $|\langle U^{(j)}, \bar{X} \rangle| \le \Delta_\delta$. $\qquad \square$

**Lemma 4.13.** *Let $X$ be a dataset that passes line 3 and line 7 of Algorithm 3. Then with probability $1 - 6\delta$, it holds that, for every point $x \in \hat{X}$,*

$$|\langle x, \bar{\hat{X}} \rangle| \le \Delta_\delta + \frac{12d}{\varepsilon} \sqrt{\ln \frac{5}{4\delta} \cdot \ln \frac{n}{\delta}}. \tag{19}$$

*Proof.* Since $X$ passed step 7 and by Lemma 2.5, with probability $1 - \delta$,

$$\left| \left\{ j \in [n] : |\langle X^{(j)}, \tilde{X} \rangle| > \Delta_\delta + \frac{4d}{\varepsilon} \sqrt{\ln \frac{5}{4\delta} \cdot \ln \frac{n}{\delta}} \right\} \right| \leq \frac{2}{\varepsilon} \ln \frac{1}{\delta}.$$

Therefore, $X$ and $\hat{X}$ differ in at most $\frac{2}{\varepsilon} \ln \frac{1}{\delta}$ data points, so for every $x \in \hat{X}$,

$$|\langle x, \bar{\hat{X}} \rangle| \leq |\langle x, \bar{X} \rangle| + |\langle x, \bar{X} - \bar{\hat{X}} \rangle| \leq |\langle x, \bar{X} \rangle| + 2d \cdot \frac{2}{\varepsilon} \ln \frac{1}{\delta} = |\langle x, \bar{X} \rangle| + \frac{4d}{\varepsilon} \ln \frac{1}{\delta}. \tag{20}$$

For any $x$ that was resampled during line 10, that is, $x = U^{(j)}$ for some $j \in [n]$, by Lemma 4.12, we are done. Otherwise, if $x$ was not resampled then by assumption $|\langle x, \tilde{X} \rangle| \leq \Delta_\delta + \frac{4d}{\varepsilon} \sqrt{\ln \frac{5}{4\delta} \cdot \ln \frac{n}{\delta}}$ and

$$|\langle x, \bar{X} \rangle| \leq |\langle x, \tilde{X} \rangle| + |\langle x, \bar{X} - \tilde{X} \rangle| \leq \Delta_\delta + \frac{4d}{\varepsilon} \sqrt{\ln \frac{5}{4\delta} \cdot \ln \frac{n}{\delta}} + |\langle x, \bar{X} - \tilde{X} \rangle|. \tag{21}$$

Now, $\bar{X} - \tilde{X} = R$ where $R \sim \mathcal{N}(0, \sigma^2 \mathbb{I}_{d \times d})$ and $\sigma = \sqrt{8d \ln(5/4\delta)}/\varepsilon$. By symmetry,

$$\langle x, \bar{X} - \tilde{X} \rangle = \langle \mathbf{1}, R \rangle = \sum_{i=1}^{d} Y_i,$$

where each $Y_i \sim \mathcal{N}(0, \sigma^2)$. Thus, with probability $1 - 2\delta$, for all $x \in X$,

$$|\langle x, \bar{X} - \tilde{X} \rangle| \leq \frac{4d}{\varepsilon} \sqrt{\ln \frac{5}{4\delta} \cdot \ln \frac{n}{\delta}}. \tag{22}$$

Therefore, with probability $1 - 6\delta$, by inequalities (20) and (21), $\forall x \in \hat{X}$,

$$|\langle x, \bar{\hat{X}} \rangle| \leq \Delta_\delta + \frac{12d}{\varepsilon} \sqrt{\ln \frac{5}{4\delta} \cdot \ln \frac{n}{\delta}}.$$

This completes the proof. □

We have now shown that lines 3-7 with high probability either reject $X$ or alter it to ensure that it satisfies (19), which is close to the desired condition (18). The following lemma, which follows directly from Lemma 4.4, says condition (19) is sufficient to ensure that $T$ has sufficiently low sensitivity.

**Lemma 4.14** (Sensitivity of $T$). *Suppose $\hat{X}$, $\hat{X}'$ are two sample sets of size $n$, which differ in the $n$th data point. If both $\hat{X}$ and $\hat{X}'$ satisfy* (19) *then*

$$|T(\hat{X}) - T(\hat{X}')| \leq 4\Delta_\delta + \frac{48d}{\varepsilon} \sqrt{\ln \frac{5}{4\delta} \cdot \ln \frac{n}{\delta}}.$$

We have now established that with high probability, if $X$ survives until line 13 then this step adds enough noise to maintain privacy. The next lemma states that several of the other noise infusion steps are individually differentially private. Since these mechanisms are applications of well-known privacy mechanisms described in section 2.1, we will not include their proofs here.

**Lemma 4.15.** *In the notation of Algorithm 3,*

- *the mechanism $X \mapsto z_1$ in line 2 is $\varepsilon$-DP,*

- *the mechanism $X \mapsto \tilde{X}$ in line 4 is $(\varepsilon, \delta)$-DP, and*

- *given a fixed (data independent) $\tilde{X}$, $X \mapsto z_2$ in line 6 is $\varepsilon$-DP; therefore,*

- *the mechanism $X \mapsto z_2$ in line 6 is $(2\varepsilon, \delta)$-DP (by adaptive composition).*

We can now complete our proof of the privacy guarantees of Algorithm 3. As mentioned earlier, this is not simply an application of the composition theorem for differential privacy since the privacy guarantee of line 13 relies on the failure rates of earlier steps. Given two neighboring datasets $X$ and $X'$, the proof couples the random variables in separate runs of Algorithm 3 on $X$ and $X'$, to ensure that $\hat{X}$ and $\hat{X}'$ are neighboring datasets so we can apply Lemma 4.14.

**Lemma 4.16.** *Algorithm 3 is $(4\varepsilon, 14\delta)$-differentially private.*

*Proof.* Suppose without loss of generality that $X$ and $X'$ differ on the $n$th data point. We can think of the random process that maps $X$ to $\hat{X}$ as a series of random variables. We denote these random samples by $(r_1, R, r_2, U, u_n, r_3)$ where $r_1, R, r_2$ and $r_3$ are as in Algorithm 3, $U$ is the random samples that occur in line 10 on all data points except the $n$th data point and $u_n$ is the randomness potentially used on the $n$th data point in line 10. Note that since the algorithm can terminate before running the entire algorithm, not all runs of the algorithm will sample from all these random variables.

Let $C_{X, \tilde{X}} = |\{j \in [n] : |\langle X^{(j)}, \tilde{X} \rangle| > \Delta_\delta + \frac{4d}{\varepsilon} \sqrt{\ln \frac{5}{4\delta} \cdot \ln \frac{n}{\delta}}\}|$. If we denote the output of Algorithm 3 on dataset $X$ with randomness $(r_1, R, r_2, U, u_n, r_3)$ by $M(X \mid (r_1, R, r_2, U, u_n, r_3))$ then for any $r_1, R, r_2, U, u_n$ and $r_3$,

$$M(X \mid (r_1, R, r_2, U, u_n, r_3))$$
$$= M(X' \mid (r_1 + \|\bar{X}\|_\infty - \|\bar{X}'\|_\infty, R + \bar{X} - \bar{X}', r_2 + C_{X, \tilde{X}} - C_{X', \tilde{X}}, U, u_n, r_3 + T(\widehat{X}_{(U, u_n)}) - T(\widehat{X'}_{(U, u_n)})))$$

Let $B_1$ be the event that both $X$ and $X'$ pass line 3 and line 7 and $B_2$ be the event that $\hat{\tilde{X}}$ and $\hat{\tilde{X}}'$ satisfy (19). By Lemma 4.13, $\Pr[B_1 \cap B_2^c] \leq 12\delta$ and by Lemma 4.14 if $B_1 \cap B_2$ holds then, for all $S \subseteq \mathbb{R}$,[1]

$$\Pr[r_3 + T(\widehat{X}_{(U, u_n)}) - T(\widehat{X'}_{(U, u_n)}) \in S] \leq e^\varepsilon \Pr[r_3 \in S].$$

For $b \in \{\mathsf{accept}, \mathsf{reject}\}$, let $E_{X, b} = \{(r_1, R, r_2, U, u_n, r_3) \mid M(X \mid (r_1, R, r_2, U, u_n, r_3)) = b\}$. Now,

$$\begin{aligned}
\Pr[M(X) = b] &= \Pr[E_{X, b} \cap B_1^c] + \Pr[E_{X, b} \cap B_1] \\
&= \Pr[E_{X, b} \cap B_1^c] + \Pr[E_{X, b} \cap B_1 \cap B_2] + \Pr[E_{X, b} \cap B_1 \cap B_2^c] \\
&\leq \Pr[E_{X, b} \cap B_1^c] + \Pr[E_{X, b} \cap B_1 \cap B_2] + 12\delta.
\end{aligned}$$

If $B_1^c$ holds then $X$ is rejected in either line 3 or line 7. Since all the steps leading up to either of these lines are differentially private (i.e., $X \mapsto (z_1, z_2)$ is $(3\varepsilon, \delta)$-DP by Lemma 4.15), we have

$$\Pr[E_{X, b} \cap B_1^c] \leq e^{3\varepsilon} \Pr[E_{X', b} \cap B_1^c] + \delta.$$

Note that $E_{X, b}$ is also equal to

$$\{(r_1 + \|\bar{X}\|_\infty - \|\bar{X}'\|_\infty, R + \bar{X} - \bar{X}', r_2 + C_{X, \tilde{X}} - C_{X', \tilde{X}}, U, u_n, r_3 + T(\widehat{X}_{(U, u_n)}) - T(\widehat{X'}_{(U, u_n)}))$$
$$\mid M(X' \mid (r_1, R, r_2, U, u_n, r_3)) = b\},$$

so by the definition of $B_1$ and $B_2$ and Lemma 4.15,

$$\Pr[E_{X,b} \cap B_1 \cap B_2] \leq e^{4\varepsilon} \Pr[E_{X',b} \cap B_1 \cap B_2] + \delta.$$

Therefore,

$$
\begin{aligned}
\Pr[M(X) = b] &\leq \Pr[E_{X,b} \cap B_1^c] + \Pr[E_{X,b} \cap B_1 \cap B_2] + 12\delta \\
&\leq e^{3\varepsilon} \Pr[E_{X',b} \cap B_1^c] + e^{4\varepsilon} \Pr[E_{X',b} \cap B_1 \cap B_2] + 14\delta \\
&\leq e^{4\varepsilon} (\Pr[E_{X',b} \cap B_1^c] + \Pr[E_{X',b} \cap B_1 \cap B_2]) + 14\delta \\
&\leq e^{4\varepsilon} \Pr[E_{X',b}] + 14\delta \\
&= e^{4\varepsilon} \Pr[M(X') = b] + 14\delta \,,
\end{aligned}
$$

concluding the proof. $\qquad\square$

We now turn to proving that Algorithm 3 distinguishes between the cases $P = \mathcal{U}_d$ and $\|P - \mathcal{U}_d\|_1 \geq \alpha$ with probability $2/3$. The crux of the proof can be summarized as

1. If $P = \mathcal{U}_d$, then with high probability $X$ passes the first check at line 3.

2. We can choose $\Delta_\delta$ such that:

   (a) If $X$ passes line 3 then $X = \hat{X}$ with high probability, so $T(X) = T(\hat{X})$.

   (b) The amount of noise added to $T(\hat{X})$ is small enough that one can still distinguish between the two hypotheses.

The next two lemmas establish parts 1 and 2a of our proof outline.

**Lemma 4.17.** *With probability at least $1 - 3\delta$, if $P = \mathcal{U}_d$ then $z_1 \leq \sqrt{2n \ln \frac{d}{\delta}} + \frac{2}{\varepsilon} \ln \frac{1}{\delta}$.*

*Proof.* If $P = \mathcal{U}_d$ then $\mathbb{E}[\bar{X}_i] = 0$, $\forall i \in [d]$. Each $\bar{X}_i$ is a sum of $n$ independent random variables in $[-1, 1]$, therefore, by Hoeffding 's inequality, it holds that

$$\Pr[|\bar{X}_i| \geq t] \leq 2 \exp(-2t^2/4n).$$

For $t = \sqrt{2n \ln \frac{d}{\delta}}$,

$$\Pr\left[|\bar{X}_i| \geq \sqrt{2n \ln \frac{d}{\delta}}\right] \leq \frac{2\delta}{d}.$$

Therefore, by a union bound, with probability at least $1 - 2\delta$, $|\bar{X}_i| \leq \sqrt{2n \ln(d/\delta)}$ holds for all $i \in [d]$. By Lemma 2.5, it holds that with probability $1 - \delta$, $|r_1| \leq \frac{2}{\varepsilon} \ln \frac{1}{\delta}$. We conclude that if $P = \mathcal{U}_d$, then with probability $1 - 3\delta$,

$$z_1 = \max_{i \in [d]} |X_i| + r_1 \leq \sqrt{2n \ln \frac{d}{\delta}} + \frac{2}{\varepsilon} \ln \frac{1}{\delta}. \qquad\square$$

**Lemma 4.18.** *If $X$ passes line 3 then with probability $1 - 6\delta$, $X = \hat{X}$.*

*Proof.* If $X$ passes line 3, then by Lemma 2.5, with probability $1 - \delta$,

$$\max_{i \in [d]} |\bar{X}_i| \leq \sqrt{2n \ln \frac{d}{\delta}} + \frac{4}{\varepsilon} \ln \frac{1}{\delta}.$$

It follows that with probability $1 - \delta$,

$$\|\bar{X}\|_2^2 = \sum_{i=1}^d \bar{X}_i^2 \leq d \left( \sqrt{2n \ln \frac{d}{\delta}} + \frac{4}{\varepsilon} \ln \frac{1}{\delta} \right)^2 \leq 2d \left( 2n \ln \frac{d}{\delta} + \frac{16}{\varepsilon^2} \ln^2 \frac{1}{\delta} \right) \leq 16d \left( n \ln \frac{d}{\delta} + \frac{1}{\varepsilon^2} \ln^2 \frac{1}{\delta} \right).$$

Setting $\beta = \delta/n$ in Lemma 4.7, we have that, with probability $1 - 2\delta$, $\forall x \in X$,

$$|\langle x, \bar{X} \rangle| \leq \frac{\|\bar{X}\|_2^2}{n} + \sqrt{2} \|\bar{X}\|_2 \sqrt{\ln(n/\delta)}.$$

Substituting the bound on $\|\bar{X}\|_2^2$ and by union bound we have that with probability $1 - 3\delta$,

$$\forall x \in X \ |\langle x, \bar{X} \rangle| \leq 16 \left( d \ln \frac{d}{\delta} + \frac{d}{n\varepsilon^2} \ln^2 \frac{1}{\delta} + \sqrt{nd \ln \frac{d}{\delta} \cdot \ln \frac{n}{\delta}} + \frac{\sqrt{d}}{\varepsilon} \ln \frac{1}{\delta} \sqrt{\ln \frac{n}{\delta}} \right) = \Delta_\delta.$$

By inequality (22) and union bound, with probability $1 - 5\delta$, $\forall x \in X$,

$$|\langle x, \tilde{X} \rangle| \leq \Delta_\delta + \frac{4d}{\varepsilon} \sqrt{\ln \frac{5}{4\delta} \cdot \ln \frac{n}{\delta}}.$$

So, with probability $1 - 5\delta$, $|\{j \in [n] : |\langle X^{(j)}, \tilde{X} \rangle| > \Delta_\delta + \frac{4d}{\varepsilon} \sqrt{\ln \frac{5}{4\delta} \cdot \ln \frac{n}{\delta}}\}| = 0$. Since with probability $1 - \delta$, $|r_2| < \frac{1}{\varepsilon} \ln \frac{1}{\delta}$, it follows that $z_2 \leq \frac{1}{\varepsilon} \ln \frac{1}{\delta}$. Therefore, with probability $1 - 6\delta$, $X$ survives lines 3 and 7 and none of the points get changed in lines 8-12, that is, $X = \hat{X}$. □

We are now ready to prove the main theorem of this section.

*Proof of Theorem 4.11.* The privacy guarantee was established in Lemma 4.16, it remains to prove completeness and soundness.

**Completeness:** Suppose $P = \mathcal{U}_d$. By Lemma 4.17, with probability $1 - 3\delta$, $X$ survives line 3. Conditioned on surviving line 3, by Lemma 4.18, with probability $1 - 6\delta$, we have that $X = \hat{X}$. Thus, by union bound, with probability $1 - 9\delta$, $X = \hat{X}$ and $T(\hat{X}) = T(X)$.

As in the proof of Theorem 4.1, the remainder of the proof involves showing that for the given choice of $n$, it holds that standard deviation of the test statistic does not overwhelm the signal, that is,

$$\frac{\ln 20}{\varepsilon} \left( 4\Delta_\delta + \frac{48d}{\varepsilon} \sqrt{\ln \frac{5}{4\delta} \cdot \ln \frac{n}{\delta}} \right) \leq \frac{n(n-1)\alpha^2}{8}. \tag{23}$$

If (23) holds then with probability 0.95, $|r_3| \leq \frac{n(n-1)\alpha^2}{8}$. If in addition $n = \Omega\left(\frac{d^{1/2}}{\alpha^2}\right)$, then we can show that the final test returns accept, except with constant probability. Overall, with probability at least $0.9 - 9\delta$, as in the proof of Theorem 4.1, Algorithm 3 will return accept. For $\delta \leq 0.02$, this translates to success probability at least $2/3$.

Condition (23) is satisfied provided

$$
n = \Omega\left( \frac{d^{1/2}}{\alpha^2} + \frac{d^{1/2}}{\alpha\varepsilon^{1/2}}\left(\ln\frac{d}{\delta}\right)^{1/2} + \frac{d^{1/3}}{\alpha^{2/3}\varepsilon}\left(\ln\frac{1}{\delta}\right)^{1/3} + \frac{d^{1/3}}{\alpha^{4/3}\varepsilon^{2/3}}\left(\ln\frac{d}{\delta}\right)^{1/3}\left(\ln\frac{d}{\alpha\varepsilon\delta}\right)^{1/3} \right.
$$
$$
\left. + \frac{d^{1/4}}{\alpha\varepsilon}\left(\ln\frac{1}{\delta}\right)^{1/2}\left(\ln\frac{d}{\alpha\varepsilon\delta}\right)^{1/4} + \frac{d^{1/2}}{\alpha\varepsilon}\left(\ln\frac{1}{\delta}\right)^{1/4}\left(\ln\frac{d}{\alpha\varepsilon\delta}\right)^{1/4} \right), \tag{24}
$$

which we will prove below matches our claimed sample complexity up to logarithmic factors.

**Soundness:** Let us assume that the algorithm does not return reject in line 3 or 7, which would be the desired output in this case. By Lemma 4.18, since $X$ passed line 3, with probability $1 - 6\delta$, we have that $X = \hat{X}$. The rest follows again as in the proof of Theorem 4.1.

The final sample complexity guarantee follows by observing that (up to polylogarithmic factors) one of these terms can never dominate the asymptotic sample complexity.

**Claim 4.19.** *For any choice of parameters* $d, \alpha, \varepsilon$, $\frac{d^{1/3}}{\alpha^{4/3}\varepsilon^{2/3}} \leq \max\left\{\frac{d^{1/2}}{\alpha^2}, \frac{d^{1/4}}{\alpha\varepsilon}\right\}$.

*Proof.* Let us assume that $\frac{d^{1/3}}{\alpha^{4/3}\varepsilon^{2/3}} > \max\left\{\frac{d^{1/2}}{\alpha^2}, \frac{d^{1/4}}{\alpha\varepsilon}\right\}$. Then $\frac{d^{1/3}}{\alpha^{4/3}\varepsilon^{2/3}} > \frac{1}{3}\cdot\frac{d^{1/2}}{\alpha^2} + \frac{2}{3}\cdot\frac{d^{1/4}}{\alpha\varepsilon}$. By the AM-GM inequality, it holds that

$$
\frac{1}{3}\cdot\frac{d^{1/2}}{\alpha^2} + \frac{2}{3}\cdot\frac{d^{1/4}}{\alpha\varepsilon} \geq \frac{d^{1/6}}{\alpha^{2/3}}\cdot\frac{d^{2/12}}{\alpha^{2/3}\varepsilon^{2/3}} = \frac{d^{1/3}}{\alpha^{4/3}\varepsilon^{2/3}},
$$

which leads to a contradiction. Therefore, it must be that $\frac{d^{1/3}}{\alpha^{4/3}\varepsilon^{2/3}} \leq \max\left\{\frac{d^{1/2}}{\alpha^2}, \frac{d^{1/4}}{\alpha\varepsilon}\right\}$. $\square$

Ignoring the polylogarithmic factors and by Claim 4.19, the sample complexity stated in (24) is simplified to

$$
n = \tilde{O}\left( \frac{d^{1/2}}{\alpha^2} + \frac{d^{1/2}}{\alpha\varepsilon} \right).
$$

This concludes the proof of Theorem 4.11. $\square$

# 5 Balanced Identity Testing of Product Distributions

In previous sections, we considered only testing whether an unknown product distribution is uniform. We will provide a generic reduction which preserves $\ell_2$-distance between means. In the case where the distributions are "balanced" (i.e., the coordinate means are bounded away from being $-1$ or $1$), this will imply that our upper bounds carry over to this more general setting.

**Lemma 5.1.** *Suppose we are given a known binary product distribution $Q$ and a sample $X \sim P$, where $P$ is some unknown binary product distribution. There exists a (randomized) transformation $Y = f_Q(X)$ such that $Y \sim P'$, where $P'$ is some unknown binary product distribution such that $\|\mathbb{E}[P'] - \mathbb{E}[\mathcal{U}_d]\|_2 = \|\mathbb{E}[P']\|_2 = \frac{1}{2}\|\mathbb{E}[P] - \mathbb{E}[Q]\|_2$.*

*Proof.* The transformation is as follows: for each coordinate $i \in [d]$, sample $b_i \sim \mathsf{Ber}(1/2)$. If $b_i = 0$, then $Y_i = X_i$: in this case, $\mathbb{E}[Y_i] = \mathbb{E}[P_i]$. Otherwise, let $Y_i = 1$ with probability $\frac{1 - \mathbb{E}[Q_i]}{2}$, and $-1$ with probability $\frac{1 + \mathbb{E}[Q_i]}{2}$ – in this case, $\mathbb{E}[Y_i] = -\mathbb{E}[Q_i]$. Putting these cases together, we overall have that $\mathbb{E}[Y_i] = \frac{1}{2}(\mathbb{E}[P_i] - \mathbb{E}[Q_i])$. Since each coordinate is independent, we have that $Y \sim P'$, where $P'$ is a binary product distribution with the same mean. Overall, this gives us that $\|\mathbb{E}[P']\|_2^2 = \frac{1}{4}\|\mathbb{E}[P] - \mathbb{E}[Q]\|_2^2$, which allows us to conclude the desired statement. $\qquad\square$

**Corollary 5.2.** *Let $\tau > 0$ be some fixed constant, and $c = c(\tau)$ be some sufficiently small constant (which depends on $\tau$). Let $Q$ be some known product distribution over $\{\pm 1\}^d$, such that $-1 + \tau \leq \mathbb{E}[Q_i] \leq 1 - \tau$.*

*Suppose there exists an algorithm which takes $n$ samples from an unknown product distribution $P'$ over $\{\pm 1\}^d$ and can distinguish between the following two cases with probability at least $2/3$: (U1) $P' = \mathcal{U}_d$, (U2) $\|P' - \mathcal{U}_d\|_1 \geq c\alpha$. Then there exists an algorithm which takes $n$ samples from an unknown product distribution $P$ over $\{\pm 1\}^d$ and can distinguish between the following two cases with probability at least $2/3$: (B1) $P = Q$, (B2) $\|P - Q\|_1 \geq \alpha$.*

*Proof.* The proof will follow via Lemma 5.1: given a set of samples for the latter problem, we convert them via the method of Lemma 5.1, and run the given algorithm for the former problem. The former case is immediate, since equal product distributions will have equal mean vectors.

We thus consider the latter case, $\|P - Q\|_1 \geq \alpha$. The upper bound part of Lemma 2.8 implies that $\|\mathbb{E}[P] - \mathbb{E}[Q]\|_2 \geq \alpha/C$. Applying the conversion of Lemma 5.1 gives that $\|\mathbb{E}[P'] - \mathcal{U}_d\|_2 \geq \alpha/2C$. We then argue that $\|P' - \mathcal{U}_d\|_1 \geq c\alpha$, which follows from the lower bound of Lemma 2.8 (applied with $P$ and $Q$ equal to our $P'$ and $\mathcal{U}_d$, respectively), concluding the proof. $\qquad\square$

# 6 Extreme Product Distributions

In this section, we discuss algorithms and lower bounds for *extreme product distributions*. Roughly speaking, an extreme product distribution is a product distribution with marginal distributions which are sufficiently close to being deterministically either $-1$ or $1$. More precisely, we have the following:

**Definition 6.1.** Fix any constant $C > 0$. A *C-extreme product distribution* is a product distribution over $\{\pm 1\}^d$ with mean vector $(p_1, \ldots, p_d)$ such that, for all $i \in [d]$, $|p_i - 1| \leq C/d$ or $|p_i + 1| \leq C/d$, and $\sum_{i=1}^d (1 + p_i)/2 \leq C$. We often omit the dependence on the constant $C$ and refer to *extreme distributions*.

We will show that, up to constant factors, the sample complexity of identity testing for extreme product distributions and identity testing for univariate distributions are the same. This statement holds even without privacy constraints, which we consider to be of independent interest. We will show this via a pair of sample-complexity preserving reductions, which will then allow us to immediately port results from the univariate private testing literature.

More precisely, we will show the following theorem.

**Theorem 6.2.** *For $n = \Omega((\log d)/\varepsilon)$, there is an $O(n)$-sample $\varepsilon$-differentially private algorithm for testing identity to any distribution $Q_{\mathrm{univ}}$ over $[d]$ if and only if there is an $O(n)$-sample $\varepsilon$-differentially private algorithm for testing identity to any extreme product distributions $Q_{\mathrm{prod}}$ over $\{\pm 1\}^d$.*

This will be proven through a combination of Lemma 6.4 in Section 6.1, and Lemma 6.7 in Section 6.2. With Theorem 6.2 in place, we can conclude the following corollary, which is a consequence of the corresponding statements for private univariate testing in [ASZ18].

**Corollary 6.3.** *For every extreme product distribution $Q \in \{\pm 1\}^d$, there exists an $\varepsilon$-differentially private algorithm which takes*

$$n = O\left(\frac{d^{1/2}}{\alpha^2} + \frac{d^{1/2}}{\alpha \varepsilon^{1/2}} + \frac{d^{1/3}}{\alpha^{4/3} \varepsilon^{2/3}} + \frac{1}{\alpha \varepsilon}\right)$$

*samples from some unknown product distribution $P \in \{\pm 1\}^d$ and distinguishes between the cases where $P = Q$ and $\|P - Q\|_1 \geq \alpha$ with probability at least $2/3$. Furthermore, every algorithm which has the same guarantees requires $\Omega(n)$ samples from $P$, as long as $\varepsilon = \tilde{\Omega}(1/d)$.*

## 6.1 Reducing from Extreme Product Testing to Univariate Testing

In this section, we show the following, which shows that any univariate identity testing algorithm implies a *multivariate* identity testing algorithm for extreme product distributions.[2]

**Lemma 6.4.** *There exists a constant $\gamma \in (0, 1]$ (depending only on the parameter $C$ of extreme distributions) such that the following holds. If there is an $n$-sample $\varepsilon$-DP algorithm for testing identity to a distribution $Q_{\mathrm{univ}}$ over $[d]$ (with distance parameter $\gamma \alpha$), then there is an $n'$-sample $\varepsilon$-DP algorithm for testing identity to extreme product distributions over $\{\pm 1\}^d$ (with distance parameter $\alpha$) where $n' = O(n + (\log d)/\varepsilon)$.*

*Proof.* For convenience and ease of notation, we hereafter consider distributions over $\{0, 1\}^d$ instead of the equivalent choice $\{\pm 1\}^d$, as this allows us to map more easily mean vectors of product distributions to probability vectors of univariate distributions. Fix $C \geq 0$, and suppose there exists an algorithm $A$ for testing identity (with distance parameter $\alpha$) to distributions over $[d]$, with sample complexity $n(\alpha)$. Given a fixed $C$-extreme product distribution $Q_{\mathrm{prod}}$ (with mean vector $q \in [0, 1]^d$) over $\{\pm 1\}^d$, and $n$ samples from an unknown product distribution $P$ over $\{\pm 1\}^d$ (with unknown mean vector $p \in [0, 1]^d$), the claimed algorithm to test identity to $Q_{\mathrm{prod}}$ works as follows. First, without loss of generality (and up to flipping the corresponding coordinates in all samples from $P$), one can assume that $0 \leq q_i \leq C/d$ for all $i \in [d]$.

The first step is to apply the differentially private algorithms from Lemmas 6.5 and 6.6, with constant probability of failure $1/10$ and privacy parameter $\varepsilon/3$ (and, for the second algorithm, for $\tau$ set to $\max(1, C)$), to the samples of $P$, and reject immediately if either test rejects (This spends a total "privacy budget" of $2\varepsilon/3$, so that we still have $\varepsilon/3$ to use in the rest, when calling the univariate purported tester). These two tests are simultaneously correct with probability at least $4/5$; we then can continue assuming $\|p\|_\infty \leq 1/2$ and $\|p\|_1 \leq 8 \max(1, C)$.
For any of the $n$ samples $X^{(1)}, \ldots, X^{(n)} \sim P$:

- If $X^{(i)} = \mathbf{0}$, then output a sample with value $Y_i \leftarrow d + 1$;

- If $\|X^{(i)}\| = 1$, i.e., $X^{(i)}$ has exactly one non-zero coordinate, then output a sample $Y_i$ with the value of this coordinate;

- If $\|X^{(i)}\| > 1$, then output a sample with value $Y_i \leftarrow d + 2$.

This therefore generates $n$ i.i.d. samples from some distribution $P_{\text{univ}}$ over $[d+2]$. Moreover, since $Q_{\text{prod}}$ is fully known, the corresponding distribution $Q_{\text{univ}}$ (which one would obtain by applying this transformation to samples from $Q_{\text{prod}}$) is uniquely specified and known; the algorithm then invokes the univariate identity tester $A$ on the $n$ i.i.d. samples $Y_1, \ldots, Y_n$ to test identity to $Q_{\text{univ}}$.

- If $P = Q_{\text{prod}}$, then we have $P_{\text{univ}} = Q_{\text{univ}}$.

- If $\|P - Q_{\text{prod}}\|_1 > \alpha$, then $\|P_{\text{univ}} - Q_{\text{univ}}\|_1 > \gamma\alpha$, where $\gamma := \frac{e^{-C}}{8(1+16\max(1,C))}$. To prove this statement, we denote by $\mathbf{e}_j$ the $j$-th vector of the canonical basis of $\mathbb{R}^d$ and observe that by definition $P_{\text{univ}}(d+1) = P(\mathbf{0}) = \prod_{i=1}^d (1 - p_i)$, while, for $1 \leq j \leq d$, $P_{\text{univ}}(j) = P(\mathbf{e}_j) = p_j \prod_{i \neq j}(1 - p_i) = \frac{p_j}{1 - p_j} \cdot P(\mathbf{0})$. Hence,

$$\|P_{\text{univ}} - Q_{\text{univ}}\|_1 \geq \sum_{i=1}^{d+1} |P_{\text{univ}}(i) - Q_{\text{univ}}(i)|$$

$$= |P(\mathbf{0}) - Q_{\text{prod}}(\mathbf{0})| + \sum_{i=1}^{d} |\frac{p_i}{1 - p_i} \cdot P(\mathbf{0}) - \frac{q_i}{1 - q_i} \cdot Q_{\text{prod}}(\mathbf{0})|$$

If $|P(\mathbf{0}) - Q_{\text{prod}}(\mathbf{0})| > \gamma\alpha$, then we are good; otherwise, $|P(\mathbf{0}) - Q_{\text{prod}}(\mathbf{0})| \leq \gamma\alpha$, from which we can bound the second term as

$$\sum_{i=1}^{d} |\frac{p_i}{1 - p_i} \cdot P(\mathbf{0}) - \frac{q_i}{1 - q_i} \cdot Q_{\text{prod}}(\mathbf{0})|$$

$$\geq Q_{\text{prod}}(\mathbf{0}) \sum_{i=1}^{d} \left| \frac{p_i}{1 - p_i} - \frac{q_i}{1 - q_i} \right| - \sum_{i=1}^{d} \frac{p_i}{1 - p_i} |P(\mathbf{0}) - Q_{\text{prod}}(\mathbf{0})|$$

$$\geq Q_{\text{prod}}(\mathbf{0}) \sum_{i=1}^{d} \left| \frac{p_i}{1 - p_i} - \frac{q_i}{1 - q_i} \right| - \gamma\alpha \sum_{i=1}^{d} \frac{p_i}{1 - p_i}.$$

Observing that the function $f \colon x \in [0, 1/2] \to \frac{x}{1-x}$ is smooth with $f'(x) \in [1/4, 1]$, we get $\frac{1}{4}|x - y| \leq |f(x) - f(y)| \leq |x - y|$ for all $x, y \in [0, 1/2]$.and therefore

$$\sum_{i=1}^{d} |\frac{p_i}{1 - p_i} \cdot P(\mathbf{0}) - \frac{q_i}{1 - q_i} \cdot Q_{\text{prod}}(\mathbf{0})| \geq \frac{Q_{\text{prod}}(\mathbf{0})}{4} \sum_{i=1}^{d} |p_i - q_i| - 2\gamma\alpha \sum_{i=1}^{d} p_i$$

$$\geq \frac{e^{-C+O(1/d)}}{4} \sum_{i=1}^{d} |p_i - q_i| - 16\max(1, C)\gamma\alpha,$$

where for the last inequality we used the fact that $Q_{\text{prod}}$ is $C$-extreme to bound $Q_{\text{prod}}(\mathbf{0})$, and the fact that $\sum_{i=1}^{d} p_i \leq 8\max(1, C)$. For $d$ large enough, $\frac{e^{-C+O(1/d)}}{4} \geq \frac{e^{-C}}{8} = (1 + 16\max(1, C))\gamma$, and therefore (recalling the folklore subadditive bound on total variation

distance with regard to product distributions)

$$\sum_{i=1}^{d} |\frac{p_i}{1-p_i} \cdot P(\mathbf{0}) - \frac{q_i}{1-q_i} \cdot Q_{\text{prod}}(\mathbf{0})| \geq (1 + 16\max(1,C))\gamma \sum_{i=1}^{d} |p_i - q_i| - 16\max(1,C)\gamma\alpha$$
$$\geq (1 + 16\max(1,C))\gamma\|P - Q_{\text{prod}}\|_1 - 16\max(1,C)\gamma\alpha$$
$$> \gamma\alpha\,,$$

as claimed.

Correctness then follows from that of the purported univariate identity tester, called with privacy parameter $\varepsilon/3$ and failure probability $1/5$ (so that by a union bound, the overall correctness is $3/5$, and the whole procedure is $\varepsilon$-differentially private). $\qquad\square$

It only remains to provide the proofs of the two helper subroutines we used in the reduction:

**Lemma 6.5.** *There is an $\varepsilon$-differentially private algorithm which can distinguish between the cases that a product distribution over $\{0,1\}^d$ with mean vector $p$ has $\|p\|_\infty \leq 1/4$ versus $\|p\|_\infty \geq 1/2$ using $n$ samples, for $n = O((\log d)/\varepsilon)$.*

*Proof.* This will follow by an application of Report Noisy Max (see [DR14]). Draw $n = \Omega(\frac{\log d}{\varepsilon})$ samples from the product distribution, and consider the $d$ functions $f_1, \ldots, f_d$, where $f_i$ computes the empirical marginal distribution for coordinate $i$. Note that each $f_i$ has sensitivity $1/n$, so by the guarantees of Report Noisy Max, it is $\varepsilon$-differentially private to output $\hat{f}_{i^*} \triangleq \max_{i\in[d]} f_i + \text{Lap}(1/n\varepsilon)$. By a Chernoff bound, union bound, and tail bounds on Laplace random variables, the difference between $\hat{f}_i$ and $p_i$ will be bounded by $O(\sqrt{(\log d)/n} + (\log d)/(n\varepsilon))$, simultaneously for all $i \in [d]$, with probability at least $99/100$. By choosing $n = \Omega((\log d)/\varepsilon)$, we upper-bound this error term by $1/8$. Therefore, thresholding the value of $\hat{f}_{i^*}$ at the value $3/8$ will distinguish the two cases, as desired. $\qquad\square$

**Lemma 6.6.** *For any $\tau \geq 1$, there is an $\varepsilon$-differentially private algorithm which can distinguish between the cases that a product distribution over $\{0,1\}^d$ with mean vector $p$ has $\|p\|_1 \leq \tau$ versus $\|p\|_1 \geq 8\tau$ using $n$ samples, for $n = O(1/\varepsilon)$.*

*Proof.* The algorithm will first draw $n$ samples, and compute the fraction $f$ of these draws which have at least $8\tau$ ones. Note that this statistic has sensitivity $1/n$, so to privatize it, we can add a $\text{Lap}(1/n\varepsilon)$ random variable, giving us a statistic $\hat{f}$. If the result is at most $3/8$, then we output that $\|p\|_1 \leq \tau$, else, we output that $\|p\|_1 \geq 8\tau$.

Let $r$ be the probability that a single string has at least $4\tau$ ones. We start by showing that there exists a gap in the value of $r$ in the two cases. Let $N$ denote the number of 1's is a randomly drawn string from $p$. We have $\mathbb{E}[X] \leq \tau$ and $\mathbb{E}[X] \geq 8\tau \geq 8$ in the two cases, respectively. By Markov's inequality, this means that in the first case, $r \leq 1/4$. Moreover, a simple computation shows that $\text{Var}[X] = \mathbb{E}[X] - \sum_{i=1}^{d} p_i^2 \leq \mathbb{E}[X]$, so that, by Chebyshev's inequality, in the second case we get $1 - r \leq \mathbb{P}[|X - \mathbb{E}[X]| > \mathbb{E}[X]/2] \leq 4/\mathbb{E}[X] \leq 1/2$, and therefore $r \geq 1/2$. By a Chernoff bound and a tail bound on Laplace random variables, the difference between the $\hat{f}$ and $r$ will be bounded by $O(1/\sqrt{n} + 1/(n\varepsilon))$. If we choose $n = \Omega(1/\varepsilon)$, we upper bound this error term by $1/16$, and thresholding at $3/8$ will distinguish the two cases, as desired. $\qquad\square$

## 6.2 Reducing from Univariate Testing to Extreme Product Testing

In this section, we will prove the following lemma:

**Lemma 6.7.** *There exists an absolute constant $c > 0$ such that the following holds. If there is a $cn$-sample algorithm for testing identity to any extreme product distribution $Q_{\mathrm{prod}}$ over $\{\pm 1\}^d$, then there is an $n$-sample algorithm for testing identity to any distribution $Q_{\mathrm{univ}}$ over $[d]$ such that $\|Q_{\mathrm{univ}}\| = O(1/d)$. Moreover, if the former algorithm is $\varepsilon$-DP, then so is the latter.*

This will be established via a sequence of reductions: from univariate testing to Poissonized univariate testing (Lemma 6.8), to extreme product testing (Lemma 6.9); before one final observation letting us get rid of one assumption stemming from the last reduction (Remark 6.10). We describe these reductions in the following subsections.

### 6.2.1 Univariate to Poissonized Univariate

The first reduction we perform is from having a dataset of fixed size, to drawing a dataset of variable size. This technique is known as "Poissonization," and is folklore in the distribution testing literature (see, e.g., [Can15, Appendix D.3]). We include the argument here for completeness. Drawing a random number of samples $(\mathrm{Poi}(n))$, rather than a fixed budget $(n)$, has the advantage that the frequency of each symbol $i \in [d]$ will be distributed as $\mathrm{Poi}(n \cdot P_i)$, independently. At the same time, with high probability, $\mathrm{Poi}(n) \leq O(n)$, so one can simulate drawing $\mathrm{Poi}(n)$ samples with a fixed budget, at the cost of a constant factor overhead.

**Lemma 6.8.** *If there is an algorithm which draws $\mathrm{Poi}(n)$ samples and tests identity to a distribution $Q_{\mathrm{univ}}$ over $[d]$ with probability of failure at most $\delta$ (and distance parameter $\alpha$), then there is a $2n$-sample algorithm for testing identity to $Q_{\mathrm{univ}}$ with probability of failure at most $\delta + 1/10$ (and distance parameter $\alpha$).*

*Proof.* Consider the algorithm which draws $N \sim \mathrm{Poi}(n)$ samples. With high probability, this will draw at most $2n$ samples. More precisely (see, e.g., [Can17]), we have that $\Pr[N \geq 2n] \leq \exp\left(-\frac{n^2}{2n}\right) = \exp\left(-\frac{n}{2}\right)$. For $n$ larger than some absolute constant, this is less than an arbitrarily small constant.

With this in mind, we describe the $2n$-sample algorithm. It begins by drawing its own $N \sim \mathrm{Poi}(n)$. If $N > 2n$, it outputs arbitrarily. Otherwise, it runs the algorithm which takes $\mathrm{Poi}(n)$ samples, on a set of $N$ samples (drawn uniformly at random from its set of $2n$ samples). Correctness follows from correctness of the Poissonized tester: the only change is that the probability of failure increases by $\Pr[N > 2n]$, which as argued before, will be less than an arbitrarily small constant (e.g., $1/10$) for $n$ greater than some absolute constant. $\qquad\square$

### 6.2.2 Poissonized Univariate to Extreme Product

For convenience and ease of notation, we hereafter consider distributions over $\{0, 1\}^d$ instead of the equivalent choice $\{\pm 1\}^d$.

**Lemma 6.9.** *Fix any constant $C \geq 1$. Suppose $n \geq c \log d$, where $c > 0$ is a suitably large absolute constant. If there is an algorithm which takes $n$ samples and tests identity to a distribution $Q_{\mathrm{prod}}$ on $\{0, 1\}^d$ with probability of failure at most $\delta$ (and distance parameter $\alpha' := e^C \alpha/2$), then there is an algorithm which draws $\mathrm{Poi}(2n)$ samples and tests identity to any univariate distribution such that $\|Q_{\mathrm{univ}}\|_\infty \leq C/d$, with probability of failure at most $\delta + 1/10$ (and distance parameter $\alpha$). Moreover, if the former algorithm is $\varepsilon$-DP, then so is the latter.*

In addition, as will be clear from the proof below, the reduction then guarantees that the product distribution $Q_{\text{prod}}$ obtained from such $Q_{\text{univ}}$ will satisfy $\|\mathbb{E}[Q_{\text{prod}}]\|_\infty \leq C/d$, i.e., is a $C$-extreme product distribution. Therefore, the above reduction holds even when only requiring a testing algorithm for identity to extreme product distributions.

*Proof.* Consider the following process: taking $\text{Poi}(2n)$ samples from a univariate distribution $P = (p_1, \ldots, p_d)$ over $[d]$, one obtains independent random variables $N_1, \ldots, N_d$ such that $N_i \sim \text{Poi}(2np_i)$. Now, draw $M_1, \ldots, M_d \sim \text{Poi}(2n)$ (mutually independent, and independent of $(N_i)_{i \in [d]}$), and set $N_i' \leftarrow N_i + M_i$ for all $i \in [d]$.

Clearly, the random variables $N_1', \ldots, N_d'$ are mutually independent, and further $N_i' \mid M_i$ is distributed as $\text{Bin}(M_i, p_i/(1 + p_i))$, by properties of Poisson random variables and since $\frac{\mathbb{E}[N_i]}{\mathbb{E}[N_i] + \mathbb{E}[M_i]} = \frac{2np_i}{2np_i + 2n} = \frac{p_i}{1 + p_i}$.

Let now $M \leftarrow \min_{i \in [d]} M_i$. For each $i \in [d]$, define a random binary vector $V^{(i)} \in \{0, 1\}^{M_i}$ obtained by choosing uniformly at random a subset of $[M_i]$ of size $N_i$ and filling the corresponding coordinates with 1, setting the $M_i - N_i$ coordinates to 0; then defining $T_i$ as

$$T_i \leftarrow \sum_{j=1}^{M} V_j^{(i)}$$

that is, the number of coordinates set to 1 among the first $M$. We then have $T_i \sim \text{Bin}(M, p_i/(1 + p_i))$; further, conditioned on $M$, all the $T_i$'s are independent.

The outcome of this process is then a $(d + 1)$-tuple $(M, T_1, \ldots, T_d)$; we convert this into $M$ i.i.d. samples from the product distribution $P_{\text{prod}}$ on $\{0, 1\}^d$ such that $\mathbb{E}[(P_{\text{prod}})_i] = \frac{p_i}{1 + p_i}$ in the natural way, by building an $M$-by-$d$ binary matrix with exactly $T_i$ ones in the $i$-th column, before permuting independently and uniformly at random each column.

To conclude, we claim that as long as $n \geq c \log d$ (for some absolute constant $c > 0$), then $\mathbb{P}[M \geq n] \geq \frac{9}{10}$. This follows from concentration of Poisson r.v.'s (again, see e.g., [Can17]) and a union bound over the $d$ i.i.d. random variables $M_1, \ldots, M_d$.

The tester for $Q_{\text{univ}}$ then proceeds as follows: given $\text{Poi}(2n)$ samples from an unknown $P$ over $[d]$, it applies the above procedure and, with probability at least $9/10$, succeeds in producing $n$ i.i.d. samples from the distribution $P_{\text{prod}}$ over $\{0, 1\}^d$ such that $\mathbb{E}[(P_{\text{prod}})_i] = \frac{p_i}{1 + p_i}$; it then runs the identity tester for $Q_{\text{univ}}$ on those $n$ samples. (When the reduction procedure fails, i.e., when $M < n$, then the tester outputs arbitrarily.) Correctness then follows from the below observations:

- If $P = Q_{\text{univ}}$, then $P_{\text{prod}} = Q_{\text{prod}}$;

- If $\|P - Q_{\text{univ}}\|_1 \geq \alpha$, then $\|P_{\text{prod}} - Q_{\text{prod}}\|_1 \geq \alpha$.

The first item is obvious; we hereafter establish the second. Denoting by $\mathbf{e}_j$ the $j$-th standard

vector of the canonical basis of $\{0,1\}^d$, and by $\mathbf{0}$ the all-zero vector, we have

$$\|P_{\text{prod}} - Q_{\text{prod}}\|_1 \geq |P_{\text{prod}}(\mathbf{0}) - Q_{\text{prod}}(\mathbf{0})| + \sum_{j=1}^{d} |P_{\text{prod}}(\mathbf{e}_j) - Q_{\text{prod}}(\mathbf{e}_j)|$$

$$= \left| \prod_{i=1}^{d}(1 - p_i') - \prod_{i=1}^{d}(1 - q_i') \right| + \sum_{j=1}^{d} \left| \frac{p_j'}{1 - p_j'} \prod_{i=1}^{d}(1 - p_i') - \frac{q_j'}{1 - q_j'} \prod_{i=1}^{d}(1 - q_i') \right|$$

$$= \left| \prod_{i=1}^{d} \frac{1}{1 + p_j} - \prod_{i=1}^{d} \frac{1}{1 + q_j} \right| + \sum_{j=1}^{d} \left| p_j \prod_{i=1}^{d} \frac{1}{1 + p_j} - q_j \prod_{i=1}^{d} \frac{1}{1 + q_j} \right|$$

where we wrote $p_i' := \frac{p_i}{1+p_i}$ (and similarly for $q_i'$) for conciseness. If the first term, $|P_{\text{prod}}(\mathbf{0}) - Q_{\text{prod}}(\mathbf{0})|$, is greater than $e^{-C}\alpha/2$, then we are done. Otherwise, we have, by the triangle inequality,

$$\sum_{j=1}^{d} \left| p_j \prod_{i=1}^{d} \frac{1}{1 + p_j} - q_j \prod_{i=1}^{d} \frac{1}{1 + q_j} \right| \geq \prod_{i=1}^{d} \frac{1}{1 + q_j} \sum_{j=1}^{d} |p_j - q_j| - \left| \prod_{i=1}^{d} \frac{1}{1 + p_j} - \prod_{i=1}^{d} \frac{1}{1 + q_j} \right| \sum_{j=1}^{d} p_j$$

$$\geq \prod_{i=1}^{d} \frac{1}{1 + q_j} \|P - Q_{\text{univ}}\|_1 - e^{-C}\frac{\alpha}{2}$$

using that $\sum_{j=1}^{d} p_j = 1$. However, since we had assumed $\|Q_{\text{univ}}\|_\infty \leq C/d$, we can bound $\prod_{i=1}^{d} \frac{1}{1+q_j} \geq \frac{1}{(1+C/d)^d} \geq e^{-C}$, and thus overall $\|P_{\text{prod}} - Q_{\text{prod}}\|_1 \geq \frac{e^{-C}}{2}\alpha$ in this case too. $\qquad\square$

**Remark 6.10.** *The reader may observe that the "if and only if" statement of Theorem 6.2 does not seem to quite follow from the combination of Lemmata 6.4 and 6.7. Indeed, the latter only establishes that a private identity testing algorithm for extreme product distributions yields a private identity testing algorithm for univariate distributions with small infinity norm. However, this is not an issue, as a standard reduction due to Goldreich [Gol16] shows that any univariate uniformity testing algorithm implies a general univariate identity testing algorithm with only a constant loss in parameters; and this reduction preserves differential privacy. Theorem 6.2 thus follows from combining this last reduction with Lemma 6.7.*

# 7 Lower Bounds

In this section, we discuss information-theoretic lower bounds for differentially private testing in high dimensions. First, we restate the lower bound implied by Corollary 6.3.

**Theorem 7.1.** *Any $\varepsilon$-differentially private algorithm (where $\varepsilon = \tilde{\Omega}(1/d)$) which draws $n$ samples from an unknown product distribution $P \in \{\pm 1\}^d$ and, with probability at least $2/3$, distinguishes between the cases $P = Q$ and $\|P - Q\|_1 \geq \alpha$ where $Q$ is some given product distribution over $\{\pm 1\}^d$, requires*

$$n = \Omega\left( \frac{d^{1/2}}{\alpha^2} + \frac{d^{1/2}}{\alpha\varepsilon^{1/2}} + \frac{d^{1/3}}{\alpha^{4/3}\varepsilon^{2/3}} + \frac{1}{\alpha\varepsilon} \right),$$

Note that this matches the upper bound of Theorem 4.1 up to logarithmic factors. However, the $Q$'s considered in this lower bound construction are *extreme product distributions* (from

Section 6), and it leaves open the possibility that there may exist better algorithms for the case where $Q$ is the uniform distribution.

Focusing on this case, we state the following lower bound for uniformity testing of product distributions, which also holds for Gaussian mean testing. The first term in the lower bound is the non-private sample complexity of this problem [CDKS17, DDK18], and the second term is the sample complexity of testing uniformity of a Bernoulli distribution (see, e.g., [ASZ18]).

**Theorem 7.2.** *Any $\varepsilon$-differentially private algorithm which draws $n$ samples from an unknown product distribution $P \in \{\pm 1\}^d$ and, with probability at least $2/3$, distinguishes between the cases $P = Q$ and $\|P - Q\|_1 \geq \alpha$ where $Q$ is the uniform distribution over $\{\pm 1\}^d$, requires*

$$n = \Omega\left(\frac{d^{1/2}}{\alpha^2} + \frac{1}{\alpha\varepsilon}\right).$$

Lower bounds for multivariate distribution testing appear to be much more challenging to prove than in the univariate case, due to the necessity of maintaining independence of marginals in the construction of a coupling, which is the standard technique for proving lower bounds for private distribution testing. Indeed, our lower bound in Corollary 6.3 was proved by showing an equivalence to the univariate case, and using lower bounds from this setting. Nonetheless, we conjecture that the same lower bound in Corollary 6.3 holds for the uniform distribution. In particular, in the univariate case testing uniformity is known to be the "hardest" problem within identity testing [VV14, Gol16]. If this could be shown for testing product distributions (while preserving privacy), this would imply the desired lower bound, and match our upper bound in Theorem 4.1.

## Footnotes

[1] In what follows, we overlook the measurability issues, and implicitly restrict ourselves to measurable sets.

[2]We further note that the reduction is quite general, and can be straightforwardly adapted beyond differentially private algorithms.

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

# A   Missing Proofs of Section 4

We prove here the guarantees of the non-private test statistic $T$.

**Lemma A.1** (Non-private Test Guarantees, Lemma 4.2)**.** *For the test statistic* $T(X) = \sum_{i=1}^{d}\left(\sum_{j=1}^{n}(X_i^{(j)})^2 - n\right)$ *defined in* (9), *the following hold:*

- *If* $P = \mathcal{U}_d$ *then* $\mathbb{E}[T(X)] = 0$ *and* $\mathrm{Var}(T(X)) \le 2n^2 d$.

- *If* $\|P - \mathcal{U}_d\|_1 \ge \alpha$ *then* $\mathbb{E}[T(X)] > \frac{1}{2}n(n-1)\alpha^2$.

- $\mathrm{Var}(T(X)) \le 2n^2 d + 4n\mathbb{E}[T(X)]$.

*Proof.* Note that, for any $1 \le i \le d$, $\sum_{j=1}^{n} X_1^{(j)} = 2Y_i - n$ where $Y_i$ is Binomially distributed with parameters $n$ and $\frac{1+p_i}{2}$, and $Y_1, \ldots, Y_d$ are independent. Therefore, we have

$$T(X) = \sum_{i=1}^{d}\left((2Y_i - n)^2 - n\right) = \sum_{i=1}^{d}\left(4Y_i^2 - 4nY_i + n(n-1)\right)$$

and a simple computation yields $\mathbb{E}[T(X)] = n(n-1)\sum_{i=1}^{d} p_i^2 = n(n-1)\|p\|_2^2$. If $P = \mathcal{U}_d$, then this directly implies $\mathbb{E}[T(X)] = 0$; moreover, if $\|P - \mathcal{U}_d\|_1 \ge \alpha$, then $\|p\|_2^2 \ge \alpha^2/2$ (by Lemma 2.8). Thus, this establishes the claimed bounds on the expectation of the statistic.

Turning to the variance, assume first that $P = \mathcal{U}_d$, i.e., $\|p\|_2 = 0$. In this case,

$$\mathrm{Var}(T(X)) = \sum_{i=1}^{d}\mathrm{Var}\left((2Y_i - n)^2 - n\right) = \sum_{i=1}^{d}\mathbb{E}[((2Y_i - n)^2 - n)^2] = 2n(n-1)d,$$

expanding the square and using the expression for the first to fourth moments of a $\mathrm{Bin}(n, 1/2)$ random variable. For general $P$, one can compute explicitly this quantity, to obtain

$$\mathrm{Var}(T(X)) = 2n(n-1)\sum_{i=1}^{d}(1 + (2n-4)p_i^2 - (2n-3)p_i^4) \le 2n(n-1)d + 4n\mathbb{E}[T(X)]. \quad \square$$

# B   An Efficient Private Algorithm for Gaussian Mean Testing

We present in full our computationally efficient private algorithm for multivariate Gaussian mean testing (Algorithm 4). In this setting, $P$ is a multivariate Gaussian distribution with identity covariance matrix and unknown mean, that is, $P = \mathcal{N}(\mu, \mathbb{I}_{d \times d})$ for some unknown $\mu = (\mu_1, \ldots, \mu_d) \in \mathbb{R}^d$. By drawing samples from $P$, we aim to distinguish between the cases $P = \mathcal{N}(\mathbf{0}, \mathbb{I}_{d \times d})$ and $\|P - \mathcal{N}(\mathbf{0}, \mathbb{I}_{d \times d})\|_1 \ge \alpha$, with probability at least 2/3.

**Algorithm 4** Efficient Private Gaussian Mean Testing

**Require:** Sample $X = (X^{(1)}, \ldots, X^{(n)}) \in \mathbb{R}^{n \times d}$ drawn from $P^n$. Parameters $\varepsilon, \delta, \alpha > 0$.

**Stage 1: Pre-processing**

1: **if** $n < \max\left\{25 \ln \frac{d}{\delta}, \frac{5}{\varepsilon} \ln \frac{1}{\delta}\right\}$ **then return** reject.

2: Let $c_i(X) \leftarrow \sum_{j=1}^{n} \mathbb{1}\{X_i^{(j)} \leq 0\}$ and $m_i(X) \leftarrow \frac{c_i(X)}{n} - \frac{1}{2}$ for all $i \in [d]$.

3: Let $r_1 \sim \text{Lap}(1/\varepsilon n)$ and $z_1 \leftarrow \max_{i \in [d]} |m_i(X)| + r_1$.

4:

5: **if** $z_1 > \frac{\sqrt{\ln(d/\delta)}}{\sqrt{n}} + \frac{\ln(1/\delta)}{\varepsilon n}$ **then return** reject.

6: Let $B \leftarrow 3\sqrt{\ln \frac{nd}{\delta}}$ and truncate all samples so that $X_i^{(j)} \in [-B, B] \ \forall i \in [d], j \in [n]$.

7: Let $\bar{X} \leftarrow \sum_{j=1}^{n} X^{(j)}$.

8:

9: Let $r_2 \sim \text{Lap}(2B/\varepsilon)$ and $z_2 \leftarrow \max_{i \in [d]} |\bar{X}_i| + r_2$.

10:

11: **if** $z_2 > 3\sqrt{2n \ln \frac{nd}{\delta} \cdot \ln \frac{d}{\delta}} + \frac{6}{\varepsilon} \sqrt{\ln \frac{nd}{\delta}} \ln \frac{1}{\delta}$ **then return** reject.

12: Let $\tilde{X} \leftarrow \bar{X} + R$, where $R \sim \mathcal{N}(\mathbf{0}, \sigma^2 \mathbb{I}_{d \times d})$ and $\sigma = \frac{B\sqrt{8d \ln(5/4\delta)}}{\varepsilon}$.

13: Let $\Delta_\delta^G \leftarrow 144 \left( d \ln \frac{d}{\delta} + \frac{d}{n\varepsilon^2} \ln^2 \frac{1}{\delta} + \sqrt{nd} \sqrt{\ln \frac{d}{\delta} \cdot \ln \frac{n}{\delta}} + \frac{\sqrt{d}}{\varepsilon} \ln \frac{1}{\delta} \sqrt{\ln \frac{n}{\delta}} \right) \ln \frac{nd}{\delta}$.

14: Let $r_3 \sim \text{Lap}(1/\varepsilon)$ and $z_3 \leftarrow |\{j \in [n] : |\langle X^{(j)}, \tilde{X}\rangle| > \Delta_\delta^G + \frac{36d}{\varepsilon} \ln \frac{nd}{\delta} \sqrt{\ln \frac{n}{\delta} \cdot \ln \frac{5}{4\delta}}\}| + r_3$.

15: **if** $z_3 > \frac{\ln(1/\delta)}{\varepsilon}$ **then return** reject.

**Stage 2: Filtering**

16: **for** $j = 1, \ldots, n$ **do**

17: $\quad$ **if** $|\langle X^{(j)}, \tilde{X}\rangle| > \Delta_\delta^G + \frac{36d}{\varepsilon} \ln \frac{nd}{\delta} \sqrt{\ln \frac{n}{\delta} \cdot \ln \frac{5}{4\delta}}$ **then**

18: $\quad\quad$ $\hat{X}^{(j)} \leftarrow N^{(j)}$, where $N^{(j)} \sim \mathcal{N}(\mathbf{0}, \mathbb{I}_{d \times d})$

19: $\quad$ **else**

20: $\quad\quad$ $\hat{X}^{(j)} \leftarrow X^{(j)}$

**Stage 3: Noise addition and thresholding**

21: Define the function $T(\hat{X}) = \sum_{i=1}^{d} (\bar{\hat{X}}_i^2 - n)$.

22: Let $r_4 \sim \text{Lap}\left(\left(5\Delta_\delta^G + \frac{432d}{\varepsilon} \ln \frac{nd}{\delta} \sqrt{\ln \frac{n}{\delta} \cdot \ln \frac{5}{4\delta}}\right)/\varepsilon\right)$ and $z_4 \leftarrow T(\hat{X}) + r_4$.

23: **if** $z_4 > \frac{n^2 \alpha^2}{324}$ **then return** reject

24: **return** accept.

---

Algorithm 4 uses a noisy version of the same statistic that is used in the non-private folklore test and in our uniformity testing algorithms from Section 4:

$$T(X) = \sum_{i=1}^{d} \left(\bar{X}_i^2 - n\right) \tag{25}$$

For any two neighboring datasets $X \sim X'$ differing on the $n$-th sample, the sensitivity of $T$ is bounded by

$$|T(X) - T(X')| \leq 2|\langle X^{(n)}, \bar{X}\rangle| + 2|\langle X'^{(n)}, \bar{X}'\rangle| + \|X'^{(n)}\|_2^2. \tag{26}$$

This follows from inequality (10) in the proof of Lemma 4.4. By this bound, the desired condition that datasets must satisfy in order for $T$ to have low sensitivity – a condition similar to (18) – is

$$\forall j \in [n] \ |\langle X^{(j)}, \bar{X} \rangle| \leq \Delta_\delta^G \text{ and } \|X^{(j)}\|_2^2 \leq \Delta_\delta^G. \tag{27}$$

If indeed $P = \mathcal{N}(\mathbf{0}, \mathbb{I}_{d \times d})$, then, with high probability, condition (27) is satisfied. Following the same thinking as with uniformity testing, Algorithm 4 performs the same type of tests and modifications to make sure the dataset satisfies a condition similar to (27) before and if it reaches the final test, which involves the statistic $T$.

The additional challenge of multivariate Gaussian mean testing is that the samples are not bounded. While truncating the samples is necessary, we also need to ensure that any Gaussian dataset will likely remain unchanged through the execution of the algorithm, so that the final test, which involves the statistic $T$, is accurate. To achieve this, we estimate the low sensitivity quantity $|m_i(X)| = \frac{1}{n}|\sum_{j=1}^n \mathbb{1}\{X_i^{(j)} \leq 0\} - \frac{n}{2}|$, which, for Gaussian datasets, acts as a proxy for the mean of the $i$-th coordinate. Due to the good concentration properties of Gaussian random variables, the datasets that pass the test of line 5 are guaranteed to lie in a small range $[-B, B]$ with high probability, so that the truncation, which follows in line 6, will not change any of the samples. The new bound $B$ comes in the new condition (27), as we will define $\Delta_\delta^G = B^2 \Delta_\delta$.

The main theorem of this section is the following:

**Theorem B.1.** *Algorithm 4 is $(5\varepsilon, 17\delta)$-differentially private and for $P = \mathcal{N}(\mu, \mathbb{I}_{d \times d})$ it distinguishes between the cases $P = \mathcal{N}(\mathbf{0}, \mathbb{I}_{d \times d})$ and $\|P - \mathcal{N}(\mathbf{0}, \mathbb{I}_{d \times d})\|_1 \geq \alpha$ with probability at least $2/3$, having sample complexity*

$$n = \tilde{O}\left(\frac{d^{1/2}}{\alpha^2} + \frac{d^{1/2}}{\alpha\varepsilon}\right).$$

First, we prove the guarantees of the non-private test, on which our algorithm is based.

**Lemma B.2** (Non-private Test Guarantees). *For $T$ defined as in (25), the following hold:*

- $\mathbb{E}[T(X)] = n^2\|\mu\|_2^2.$

- $\mathrm{Var}(T(X)) = 2n^2d + 4n^3\|\mu\|_2^2.$

*Proof.* We have, recalling that $P$ has identity covariance matrix, that it is a product distribution with $i$-th marginal distributed as $\mathcal{N}(\mu_i, 1)$,

$$\mathbb{E}[T(X)] = \sum_{i=1}^d \left( \sum_{1 \leq j_1, j_2 \leq n} \mathbb{E}[X_i^{(j_1)} X_i^{(j_2)}] - n \right) = \sum_{i=1}^d \left( \sum_{j=1}^n \mathbb{E}[(X_i^{(j)})^2] + \sum_{j_1 \neq j_2} \mathbb{E}[X_i^{(j_1)}]\mathbb{E}[X_i^{(j_2)}] \right) - nd$$

$$= \sum_{i=1}^d \left( \sum_{j=1}^n \mathbb{E}[(X_i^{(j)})^2] + \left( \sum_{j=1}^n \mathbb{E}[X_i^{(j)}] \right)^2 - \sum_{j=1}^n \mathbb{E}[X_i^{(j)}]^2 \right) - nd$$

$$= \sum_{i=1}^d \left( \sum_{j=1}^n \mathrm{Var}(X_i^{(j)}) + \left( \sum_{j=1}^n \mathbb{E}[X_i^{(j)}] \right)^2 \right) - nd = \sum_{i=1}^d (n + n^2\mu_i^2) - nd = n^2\|\mu\|_2^2,$$

as claimed. Now, for the variance. Using independence among coordinates, we have

$$
\mathrm{Var}(T(X)) = \sum_{i=1}^{d} \mathrm{Var}\left(\sum_{1 \leq j_1, j_2 \leq n} X_i^{(j_1)} X_i^{(j_2)} - n\right) = \sum_{i=1}^{d} \mathrm{Var}\left(\sum_{j_1, j_2} X_i^{(j_1)} X_i^{(j_2)}\right)
$$

$$
= \sum_{i=1}^{d} \left( \mathbb{E}\left[\left(\sum_{j_1, j_2} X_i^{(j_1)} X_i^{(j_2)}\right)^2\right] - \mathbb{E}\left[\sum_{j_1, j_2} X_i^{(j_1)} X_i^{(j_2)}\right]^2 \right).
$$

We have already, that $\sum_{j_1, j_2} \mathbb{E}\left[X_i^{(j_1)} X_i^{(j_2)}\right] = n^2 \mu_i^2 + n$. This takes care of the second term; as for the first, we expand

$$
\mathbb{E}\left[\left(\sum_{j_1, j_2} X_i^{(j_1)} X_i^{(j_2)}\right)^2\right] = \sum_{1 \leq j_1, j_2, j_3, j_4 \leq n} \mathbb{E}\left[X_i^{(j_1)} X_i^{(j_2)} X_i^{(j_3)} X_i^{(j_4)}\right]
$$

$$
= n(\mu_i^4 + 6\mu_i^2 + 3) + 24\binom{n}{4}\mu_i^4 + 8\binom{n}{2}\mu_i^2(\mu_i^2 + 3) + 6\binom{n}{2}(\mu_i^2 + 1)^2 + 36\binom{n}{3}\mu_i^2(\mu_i^2 + 1)
$$

$$
= 3n^2 + 6n^3\mu_i^2 + n^4\mu_i^4 .
$$

Combining the two, we get

$$
\mathrm{Var}(T(X)) = \sum_{i=1}^{d}\left(3n^2 + 6n^3\mu_i^2 + n^4\mu_i^4 - (n^2\mu_i^2 + n)^2\right) = \sum_{i=1}^{d}\left(2n^2 + 4n^3\mu_i^2\right) = 2n^2 d + 4n^3\|\mu\|_2^2,
$$

as stated. $\qquad\square$

To prove the privacy guarantee of our main Theorem B.1, we need to show that with high probability $\hat{X}$ satisfies a property similar to equation (27), which can not be derived by a simple application of the composition theorem. We first show that the new replacement samples potentially drawn in lines 16-20 satisfy (27).

**Lemma B.3.** *Let $X$ be a dataset that passes line 11 of Algorithm 4. Suppose $N^{(j)} \sim \mathcal{N}(\mathbf{0}, \mathbb{I}_{d \times d})$ for $j \in [n]$. Then with probability $1 - 5\delta$, $\forall j \in [n]$,*

- $N_i^{(j)} \in [-B, B]$, $\forall i \in [d]$, *where* $B = 3\sqrt{\ln \frac{nd}{\delta}}$ *as defined in Algorithm 4.*

- $\left|\langle N^{(j)}, \bar{X}\rangle\right| \leq \Delta_\delta^G$.

*Proof.* We have that for all $j \in [n]$ and $i \in [d]$, $N_i^{(j)} \sim \mathcal{N}(0, 1)$. By the Gaussian tail bound, we get that for all $i \in [d]$ and $j \in [n]$, with probability $1 - 2\delta$, $|N_i^{(j)}| \leq \sqrt{2\ln \frac{nd}{\delta}} \leq B$. Therefore, the first point holds, with probability $1 - 2\delta$. Conditioned on that and since $X$ has passed line 11, by following the same steps as the proof of Lemma 4.12, we get that with probability $1 - 3\delta - 2\delta = 1 - 5\delta$, for all $x \in X$, $|\langle x, \bar{X}\rangle| \leq \Delta_\delta B^2 = \Delta_\delta^G$, which is the stated bound. $\qquad\square$

**Lemma B.4.** *Let $X$ be a dataset that passes line 15 of Algorithm 4. For every point $x \in \hat{X}$ it holds that, with probability $1 - 10\delta$,*

$$
\|x\|_2^2 \leq \Delta_\delta^G \text{ and } |\langle x, \bar{\hat{X}}\rangle| \leq \Delta_\delta^G + \frac{108d}{\varepsilon}\ln\frac{nd}{\delta}\sqrt{\ln\frac{n}{\delta} \cdot \ln\frac{5}{4\delta}}.
$$

*Proof.* Since $X$ passed line 15, it has also passed the truncation step in line 6. By Lemma B.3, all the new data points are also bounded in $[-B, B]$, with probability $1 - 5\delta$. It follows that for all $x \in \hat{X}$, $\|x\|_2^2 \leq dB^2 \leq 9d \ln \frac{nd}{\delta} \leq \Delta_\delta^G$. It remains to prove the second inequality.

Since $X$ passed line 15, $z_3 \leq \frac{\ln(1/\delta)}{\varepsilon}$. By Lemma 2.5, with probability $1 - \delta$, $|r_3| \leq \frac{\ln(1/\delta)}{\varepsilon}$, so

$$|\{j \in [n] : |\langle X^{(j)}, \tilde{X}\rangle| > \Delta_\delta^G + \frac{36d}{\varepsilon} \ln \frac{nd}{\delta} \sqrt{\ln \frac{n}{\delta} \cdot \ln \frac{5}{4\delta}}\}| \leq \frac{2}{\varepsilon} \ln \frac{1}{\delta}.$$

Therefore, $X$ and $\hat{X}$ differ in at most $\frac{2}{\varepsilon} \ln \frac{1}{\delta}$ data points. Thus, with probability $1 - 6\delta$, $\forall x \in \hat{X}$,

$$|\langle x, \bar{\hat{X}}\rangle| \leq |\langle x, \bar{X}\rangle| + |\langle x, \bar{X} - \bar{\hat{X}}\rangle| \leq |\langle x, \bar{X}\rangle| + \frac{4dB^2}{\varepsilon} \ln \frac{1}{\delta} = |\langle x, \bar{X}\rangle| + \frac{36d}{\varepsilon} \ln \frac{nd}{\delta} \cdot \ln \frac{1}{\delta}. \quad (28)$$

If $x$ was resampled in line 18 then by Lemma B.3, we are done. Otherwise, if $x$ was not resampled then by assumption $|\langle x, \tilde{X}\rangle| \leq \Delta_\delta^G + \frac{36d}{\varepsilon} \ln \frac{nd}{\delta} \sqrt{\ln \frac{n}{\delta} \cdot \ln \frac{5}{4\delta}}$. It holds that

$$|\langle x, \bar{X}\rangle| \leq |\langle x, \tilde{X}\rangle| + |\langle x, \bar{X} - \tilde{X}\rangle| \leq \Delta_\delta^G + \frac{36d}{\varepsilon} \ln \frac{nd}{\delta} \sqrt{\ln \frac{n}{\delta} \cdot \ln \frac{5}{4\delta}} + |\langle x, \bar{X} - \tilde{X}\rangle|. \quad (29)$$

Now, $\bar{X} - \tilde{X} = R$ where $R \sim \mathcal{N}(\mathbf{0}, \sigma^2 \mathbb{I}_{d \times d})$ and $\sigma = B\sqrt{8d \ln(5/4\delta)}/\varepsilon$, as in line 12 of Algorithm 4. By symmetry, $|\langle x, R\rangle| \leq B \cdot \left|\sum_{i=1}^d Y_i\right|$, where each $Y_i \sim \mathcal{N}(0, \sigma^2)$. It follows that with probability $1 - 2\delta$, $\left|\sum_{i=1}^d Y_i\right| \leq \sigma\sqrt{2d \ln \frac{n}{\delta}} \leq \frac{4dB}{\varepsilon}\sqrt{\ln \frac{n}{\delta} \cdot \ln \frac{5}{4\delta}}$. So $\forall x \in X$,

$$|\langle x, \bar{X} - \tilde{X}\rangle| \leq \frac{4dB^2}{\varepsilon}\sqrt{\ln \frac{n}{\delta} \cdot \ln \frac{5}{4\delta}} \leq \frac{36d}{\varepsilon} \ln \frac{nd}{\delta} \sqrt{\ln \frac{n}{\delta} \cdot \ln \frac{5}{4\delta}}. \quad (30)$$

Therefore, by union bound and inequalities (28) and (29), with probability $1 - 8\delta$, for all $x \in \hat{X}$,

$$\|x\|_2^2 \leq \Delta_\delta^G \text{ and } |\langle x, \bar{\hat{X}}\rangle| \leq \Delta_\delta^G + \frac{108d}{\varepsilon} \ln \frac{nd}{\delta} \sqrt{\ln \frac{n}{\delta} \cdot \ln \frac{5}{4\delta}}.$$

This concludes the proof of the lemma. $\qquad \square$

**Lemma B.5.** *Algorithm 4 is $(5\varepsilon, 17\delta)$-differentially private.*

*Proof.* By Lemma B.4 and inequality (26), with probability $1 - 16\delta$, for any two neighboring datasets $X \sim X'$ that reach line 22, $\hat{X}$ and $\hat{X}'$ satisfy condition (27). Therefore, for $\hat{X} \sim \hat{X}'$,

$$|T(\hat{X}) - T(\hat{X}')| \leq 5\Delta_\delta^G + \frac{432d}{\varepsilon} \ln \frac{nd}{\delta} \sqrt{\ln \frac{n}{\delta} \cdot \ln \frac{5}{4\delta}}.$$

This ensures that with probability $1 - 16\delta$, the noise added in the last test in line 22 is sufficient to ensure privacy. As in Section 4, the first observation towards proving the privacy guarantee is that the applications of the Laplace mechanism in lines 5, 11, and 15 are $\varepsilon$-DP and the application of the Gaussian mechanism in line 12 is $(\varepsilon, \delta)$-DP. Now, by coupling the random variables in separate runs of Algorithm 4 on $X$ and $X'$, to ensure that $\hat{X}$ and $\hat{X}'$ are neighboring databases exactly as in the proof of Lemma 4.16, we get that our algorithm is $(5\varepsilon, 17\delta)$-DP. $\qquad \square$

We now turn to the utility guarantee of our main theorem. As in the previous section, the crux of the proof is as follows:

1. If $P = \mathcal{N}(\mathbf{0}, \mathbb{I}_{d \times d})$, then with high probability $X$ passes the first two checks at line 5 and 11.

2. If $P = \mathcal{N}(\mu, \mathbb{I}_{d \times d})$, then:

   (a) If $X$ passes the first two checks at line 5 and 11, then $X = \hat{X}$ with high probability, so $T(X) = T(\hat{X})$.

   (b) The amount of noise added to $T(\hat{X})$ is small enough that one can still distinguish between the two hypotheses.

We will now prove some important properties of the first test of line 5 of Algorithm 4, which guarantee that for multivariate Gaussian distributions, the estimate $|m_i(X)|$ is close to the absolute mean $|\mu_i|$ of each coordinate.

**Lemma B.6.** *Suppose $P = \mathcal{N}(\mu, \mathbb{I}_{d \times d})$ and let $\{m_i(X)\}_{i=1}^{d}$ as defined in line 2 of Algorithm 4.*

- *If $\mu_i = 0$, then $\mathbb{E}[m_i(X)] = 0$.*

- $|\mathbb{E}[m_i(X)]| \geq 0.84 \cdot \min\left\{ \frac{|\mu_i|}{\sqrt{2}}, 1 \right\}$.

- *With probability $1 - 2\delta$, $|m_i(X) - \mathbb{E}[m_i(X)]| \leq \frac{\sqrt{\ln(d/\delta)}}{\sqrt{n}}$ for all $i \in [d]$.*

*Proof.* Let $Y_i^{(j)} = \mathbb{1}\{X_i^{(j)} \leq 0\}$ for $i \in [d], j \in [n]$. Let us calculate $\mathbb{E}[m_i(X)]$.

$$\mathbb{E}\left[ \frac{c_i(X)}{n} - \frac{1}{2} \right] = \frac{1}{n} \sum_{j=1}^{n} \mathbb{E}[Y_i^{(j)}] - \frac{1}{2} = \frac{1}{n} \sum_{j=1}^{n} \Pr[X_i^{(j)} \leq 0] - \frac{1}{2} = \frac{1}{n} \sum_{j=1}^{n} \Phi(-\mu_i) - \frac{1}{2} = \mathrm{Erf}\left( -\frac{\mu_i}{\sqrt{2}} \right),$$

where $\Phi(x)$ is the CDF of $\mathcal{N}(0,1)$ and $\mathrm{Erf}(x) = \frac{1}{\sqrt{\pi}} \int_{-x}^{x} \exp(-t^2) \, dx$ the error function. The first point follows, since $\mathrm{Erf}(0) = 0$. The second point follows by Lemma 3.2. It remains to prove the third point. By Hoeffding's inequality, for all $i \in [d]$,

$$\Pr\left[ \left| \sum_{j=1}^{n} Y_i^{(j)} - \mathbb{E}\left[ \sum_{j=1}^{n} Y_i^{(j)} \right] \right| > t \right] \leq 2 \exp(-2t^2/n).$$

Setting $t = \sqrt{n \ln \frac{d}{\delta}}$ and by union bound, we get that with probability $1 - 2\delta$, for all $i \in [d]$,
$|m_i(X) - \mathbb{E}[m_i(x)]| = \frac{1}{n} |c_i(X) - \mathbb{E}[c_i(X)]| \leq \frac{\sqrt{\ln(d/\delta)}}{\sqrt{n}}$. □

We can now prove that if $P = \mathcal{N}(\mathbf{0}, \mathbb{I}_{d \times d})$, then $X$ will pass the first two tests with high probability.

**Lemma B.7.** *With probability $1 - 6\delta$, if $P = \mathcal{N}(\mathbf{0}, \mathbb{I}_{d \times d})$ then Algorithm 4 does not reject neither in line 5, nor in line 11.*

*Proof.* Suppose $P = \mathcal{N}(\mathbf{0}, \mathbb{I}_{d \times d})$, that is, $\mu_i = 0$ for all $i \in [d]$. By Lemma B.6, $\mathbb{E}[m_i(X)] = 0$ and $|m_i(X)| \leq \frac{\sqrt{\ln(d/\delta)}}{\sqrt{n}}$ for all $i \in [d]$, with probability at least $1 - 2\delta$. It follows that, with probability $1 - 2\delta$,

$$\max_{i \in [d]} |m_i(X)| \leq \frac{\sqrt{\ln(d/\delta)}}{\sqrt{n}}. \tag{31}$$

Let $r_1 \sim \text{Lap}(1/\varepsilon n)$ as in Algorithm 4. By Lemma 2.5, with probability $1 - \delta$, $|r_1| \leq \frac{\ln(1/\delta)}{\varepsilon n}$. Combined with (31), we get that, with probability $1 - 3\delta$,

$$z_1 = \max_{i \in [d]} |m_i(X)| + r_1 \leq \frac{\sqrt{\ln(d/\delta)}}{\sqrt{n}} + \frac{\ln(1/\delta)}{\varepsilon n},$$

showing that the dataset $X$ will pass line 5.

For the test of line 11, we first observe that, since $P = \mathcal{N}(\mathbf{0}, \mathbb{I}_{d \times d})$, $\mathbb{E}[\bar{X}_i] = 0$ for all $i \in [d]$ and that, due to the truncation in line 6, for all the samples it holds that $|X_i^{(j)}| \leq B$, where $B = 3\sqrt{\ln \frac{nd}{\delta}}$. Following the same steps as Lemma 4.17, we get that with probability $1 - 3\delta$,

$$z_2 \leq 3\sqrt{2n \ln \frac{nd}{\delta} \cdot \ln \frac{d}{\delta}} + \frac{6}{\varepsilon}\sqrt{\ln \frac{nd}{\delta} \ln \frac{1}{\delta}},$$

showing that the dataset $X$ will pass line 11.

Therefore, $X$ will pass both the test in line 5 and the test in line 11, with probability $1 - 6\delta$. $\qquad\square$

Using Lemma B.6, we can also prove that all Gaussian datasets that pass the first two tests remain unchanged for the rest of the algorithm. The following lemma states that the dataset will not be modified during the truncation phase in line 6.

**Lemma B.8.** *Suppose* $P = \mathcal{N}(\mu, \mathbb{I}_{d \times d})$. *If dataset $X$ passes line 5 of Algorithm 4, then, with probability $1 - 5\delta$, no truncation occurs in line 6.*

*Proof.* Since $X$ passed line 5, it holds that $z_1 = \max_{i \in [d]} |m_i(X)| + r_1 \leq \frac{\sqrt{\ln(d/\delta)}}{\sqrt{n}} + \frac{\ln(1/\delta)}{\varepsilon n}$. By Lemma 2.5, with probability $1 - \delta$, $|r_1| \leq \frac{\ln(1/\delta)}{\varepsilon n}$. So with probability $1 - \delta$, for all $i \in [d]$,

$$|m_i(X)| \leq \frac{\sqrt{\ln(d/\delta)}}{\sqrt{n}} + \frac{2\ln(1/\delta)}{\varepsilon n}. \tag{32}$$

By Lemma B.6, with probability $1 - 2\delta$, $|m_i(X) - \mathbb{E}[m_i(X)]| \leq \frac{\sqrt{\ln(d/\delta)}}{\sqrt{n}}$ for all $i \in [d]$. By union bound and inequality (32), with probability $1 - 3\delta$, for all $i \in [d]$, $|\mathbb{E}[m_i(X)]| \leq \frac{2\sqrt{\ln(d/\delta)}}{\sqrt{n}} + \frac{2\ln(1/\delta)}{\varepsilon n}$. Again by Lemma B.6, the last inequality implies that with probability $1 - 3\delta$, for all $i \in [d]$,

$$0.84 \cdot \min\left\{\frac{|\mu_i|}{\sqrt{2}}, 1\right\} \leq \frac{2\sqrt{\ln(d/\delta)}}{\sqrt{n}} + \frac{2\ln(1/\delta)}{\varepsilon n}. \tag{33}$$

Since the algorithm has passed line 1, $n \geq \max\left\{25 \ln \frac{d}{\delta}, \frac{5}{\varepsilon} \ln \frac{1}{\delta}\right\}$, and we get $\frac{2\sqrt{\ln(d/\delta)}}{\sqrt{n}} + \frac{2\ln(1/\delta)}{\varepsilon n} \leq 0.8 < 0.84$. So it must be that $|\mu_i| < \sqrt{2}$, since inequality (33) can not be satisfied otherwise. Then it holds that

$$\frac{0.84|\mu_i|}{\sqrt{2}} \leq \frac{2\sqrt{\ln(d/\delta)}}{\sqrt{n}} + \frac{2\ln(1/\delta)}{\varepsilon n} \Rightarrow |\mu_i| \leq \frac{3.4\sqrt{\ln(d/\delta)}}{\sqrt{n}} + \frac{3.4\ln(1/\delta)}{\varepsilon n}. \tag{34}$$

Since all $X_i^{(j)} \sim \mathcal{N}(\mu_i, 1)$, we have the tail bound

$$\Pr\left[\left|X_i^{(j)} - \mu_i\right| > t\right] \leq 2\exp(-t^2/2).$$

Setting $t = \sqrt{2 \ln \frac{nd}{\delta}}$, we get that with probability $1-2\delta$, for all $i \in [d]$ and $j \in [n]$, $|X_i^{(j)}| \le |\mu_i| + t$. Replacing the chosen $t$ and by inequality (34),

$$|X_i^{(j)}| \le \frac{3.4\sqrt{\ln(d/\delta)}}{\sqrt{n}} + \frac{3.4\ln(1/\delta)}{\varepsilon n} + \sqrt{2 \ln \frac{nd}{\delta}} < 3\sqrt{\ln \frac{nd}{\delta}},$$

where the last inequality follows from our condition on $n$.

We have proven that, if $X$ passes line 5, then with probability $1 - 5\delta$, all samples already fall within the desired range $[-B, B]$, so the truncation does not affect the dataset. □

The following lemma states that all datasets that survive line 11 satisfy the desired conditions (27), which is sufficient to establish that with high probability, the dataset will not be modified during the resampling phase in lines 16-20.

**Lemma B.9.** *Let $X$ be a dataset that passes line 11 of Algorithm 4. With probability $1 - 3\delta$, for all $x \in X$,*

$$|\langle x, \bar{X} \rangle| \le 144 \left( d \ln \frac{d}{\delta} + \frac{d}{n\varepsilon^2} \ln^2 \frac{1}{\delta} + \sqrt{nd}\sqrt{\ln \frac{d}{\delta} \cdot \ln \frac{n}{\delta}} + \frac{\sqrt{d}}{\varepsilon} \ln \frac{1}{\delta} \sqrt{\ln \frac{n}{\delta}} \right) \ln \frac{nd}{\delta}.$$

*Proof.* Observe that, due to the truncation in line 6, $\forall k \in [n]$ and $\forall i \in [d]$, $|X_i^{(k)}| \le B$. Following the same steps as the proof of Lemma 4.18, we get that with probability $1 - 3\delta$, for all $x \in X$, $|\langle x, \bar{X} \rangle| \le \Delta_\delta B^2$, which is the stated bound. □

We will now prove the main theorem of this section.

*Proof of Theorem B.1.* The privacy guarantee was established in Lemma B.5. Now, for the utility guarantee.

*Completeness:* Suppose $P = \mathcal{N}(\mathbf{0}, \mathbb{I}_{d \times d})$. By Lemma B.7, with probability $1 - 6\delta$, $X$ passes line 5 and line 11. Also, by Lemma B.8, with probability $1 - 5\delta$, it does not get truncated in line 6 in the meantime. So with probability $1 - 11\delta$, $X$ has passed line 11 and reached line 15, unchanged.

By Lemma B.9 and union bound, with probability $1 - 14\delta$, for the chosen $\Delta_\delta^G$, $|\langle x, \bar{X} \rangle| \le \Delta_\delta^G$ holds for all $x \in X$.

As we have showed, by inequality (30), with probability $1 - 2\delta$, for all $x \in X$,

$$|\langle x, \bar{X} - \tilde{X} \rangle| \le \frac{36d}{\varepsilon} \ln \frac{nd}{\delta} \sqrt{\ln \frac{n}{\delta} \cdot \ln \frac{5}{4\delta}}.$$

It follows that with probability $1 - 16\delta$, $\forall x \in X$, $|\langle x, \tilde{X} \rangle| \le \Delta_\delta^G + \frac{36d}{\varepsilon} \ln \frac{nd}{\delta} \sqrt{\ln \frac{n}{\delta} \cdot \ln \frac{5}{4\delta}}$.

By Lemma 2.5, with probability $1 - \delta$, $|r_3| \le \frac{\ln(1/\delta)}{\varepsilon}$, so with probability $1 - 17\delta$, $z_3 \le \frac{\ln(1/\delta)}{\varepsilon}$. Thus, the dataset survives line 15 as well and no points are modified in lines 16-20. Since, with probability $1 - 17\delta$, the dataset reaches the last test in line 22 unchanged, it only remains to prove that the last test is accurate with high probability.

By Lemma 2.5, with probability $0.95$, $|r_4| \le \frac{\ln(20)}{\varepsilon} \left( 5\Delta_\delta^G + \frac{432d}{\varepsilon} \ln \frac{nd}{\delta} \sqrt{\ln \frac{n}{\delta} \cdot \ln \frac{5}{4\delta}} \right)$. We want the following to hold:

$$\frac{\ln(20)}{\varepsilon} \left( 5\Delta_\delta^G + \frac{432d}{\varepsilon} \ln \frac{nd}{\delta} \sqrt{\ln \frac{n}{\delta} \cdot \ln \frac{5}{4\delta}} \right) \le \frac{n^2 \alpha^2}{8 \cdot 81} \tag{35}$$

If this is true, we have $|r_4| \leq \frac{n^2\alpha^2}{8\cdot 81}$. Then, with probability $0.95 - 17\delta$,

$$
\begin{aligned}
\Pr\left[z_4 > \frac{n^2\alpha^2}{4\cdot 81}\right] &= \Pr\left[T(X) > \frac{n^2\alpha^2}{4\cdot 81} - r_3\right] && (T(\hat{X}) = T(X))\\
&\leq \Pr\left[T(X) > \frac{n^2\alpha^2}{8\cdot 81}\right] && (|r_4| \leq \frac{n^2\alpha^2}{8\cdot 81})\\
&= \Pr\left[T(X) > \frac{n^2\alpha^2}{8\cdot 81} + \mathbb{E}[T(X)]\right] && (\text{Lemma B.2})\\
&\leq \frac{8^2\cdot 81^2 \operatorname{Var}(T(X))}{n^4\alpha^4} && (\text{Chebyshev's inequality})\\
&\leq \frac{8^2\cdot 81^2\cdot 2d}{n^2\alpha^4} && (\text{Lemma B.2})
\end{aligned}
$$

For $n = \Omega\left(\frac{d^{1/2}}{\alpha^2}\right)$, it holds that $\Pr\left[z_4 > \frac{n^2\alpha^2}{4\cdot 81}\right] \leq 0.05$. So with probability $0.9 - 17\delta$, $z_4 \leq \frac{n^2\alpha^2}{324}$. Condition (35) and $n = \Omega\left(\frac{d^{1/2}}{\alpha^2}\right)$ is satisfied for

$$
\begin{aligned}
n = \Omega\bigg( &\frac{d^{1/2}}{\alpha^2} + \frac{d^{1/2}}{\alpha\varepsilon^{1/2}}\left(\ln\frac{1}{\delta}\right)^{1/2}\left(\ln\frac{d}{\alpha\varepsilon\delta}\right)^{1/2}\\
&+ \frac{d^{1/3}}{\alpha^{2/3}\varepsilon}\left(\ln\frac{1}{\delta}\right)^{2/3}\left(\ln\frac{d}{\alpha\varepsilon\delta}\right)^{1/3} + \frac{d^{1/3}}{\alpha^{4/3}\varepsilon^{2/3}}\left(\ln\frac{d}{\delta}\right)^{1/3}\left(\ln\frac{d}{\alpha\varepsilon\delta}\right)\\
&+ \frac{d^{1/4}}{\alpha\varepsilon}\left(\ln\frac{1}{\delta}\right)^{1/2}\left(\ln\frac{d}{\alpha\varepsilon\delta}\right)^{3/4} + \frac{d^{1/2}}{\alpha\varepsilon}\left(\ln\frac{1}{\delta}\right)^{1/4}\left(\ln\frac{d}{\alpha\varepsilon\delta}\right)^{3/4}\bigg),
\end{aligned}
$$

Notice that for this value of $n$, Algorithm 4 does not reject in line 1 either. Thus, for this value of $n$ and $\delta \leq 0.01$, Algorithm 4 returns accept, with probability $2/3$.

*Soundness:* Suppose $P = \mathcal{N}(\mu, \mathbb{I}_{d\times d})$ and $\|P - \mathcal{N}(\mathbf{0}, \mathbb{I}_{d\times d})\|_1 \geq \alpha$. Let us assume that the algorithm does not return REJECT in line 1, 5, or 11, which is the desired output in this case. Since $X$ has passed line 11, by Lemma B.8 and Lemma B.9, with probability $1 - 8\delta$, $X$ does not get truncated in line 6 and for all $x \in X$, $|\langle x, \bar{X}\rangle| \leq \Delta_\delta^G$. Similar to the completeness proof, with probability $0.95 - 11\delta$, the dataset reaches the last test in line 22 unchanged and for the chosen $n$, $|r_4| \leq \frac{n^2\alpha^2}{8\cdot 81}$. Before proving the accuracy of the last test in line 22, note that since $\|P - \mathcal{N}(\mathbf{0}, \mathbb{I}_{d\times d})\|_1 \geq \alpha$, by Lemma 2.7, we have that $\|\mu\|_2^2 \geq \alpha^2/81$. So, by Lemma B.2,

$$
\mathbb{E}[T(X)] \geq \frac{n^2\alpha^2}{81}. \tag{36}
$$

With probability $0.95 - 13\delta$,

$$
\begin{aligned}
\Pr\left[z_4 \le \frac{n^2\alpha^2}{4\cdot 81}\right] &= \Pr\left[T(X) \le \frac{n^2\alpha^2}{4\cdot 81} - r_4\right] && (T(\hat{X}) = T(X)) \\
&\le \Pr\left[T(X) \le \frac{3n^2\alpha^2}{8\cdot 81}\right] && (|r_4| \le \tfrac{n^2\alpha^2}{8\cdot 81}) \\
&\le \Pr\left[T(X) \le \frac{3\mathbb{E}[T(X)]}{8}\right] && (\text{by } (36)) \\
&= \Pr\left[\mathbb{E}[T(X)] - T(X) \ge \frac{5\mathbb{E}[T(X)]}{8}\right] \\
&\le \frac{8^2\,\mathrm{Var}(T(X))}{5^2(\mathbb{E}[T(X)])^2} && (\text{Chevyshev's inequality}) \\
&\le \frac{8^2\cdot 81^2\cdot 2d}{5^2 n^2\alpha^4} + \frac{8^2\cdot 4\cdot 81}{5^2 n\alpha^2} && (\text{Lemma B.2 and } (36))
\end{aligned}
$$

For $n = \Omega\left(\frac{d^{1/2}}{\alpha^2}\right)$, it holds that $\Pr\left[z_4 \le \frac{n^2\alpha^2}{4\cdot 81}\right] \le 0.05$. So with probability $0.9 - 11\delta$, $z_4 > \frac{n^2\alpha^2}{324}$. Thus, for $\delta \le 0.01$ and for the stated value of $n$, we conclude that with probability $2/3$, Algorithm 4 returns reject.

Ignoring the logarithmic factors, the sample complexity is simplified using Claim 4.19. $\qquad\square$