[Reviews · NeurIPS 2020]

Review 1

Summary and Contributions: Review Update: I agree that reducing dependence on privacy parameters, even marginally in some regimes, can have great practical importance and impact. I'm comfortable sticking with my score however. This paper studies the problem of private distribution testing, giving algorithms and bounds that improve the dependence of the error on the dimensionality, and hence scale better to high dimensions. Informally, they give results for (1) distinguishing the uniform distribution over (0,1)^d from anything alpha close in TV (uniformity testing over the cube) (2) They can reduce testing the mean of two Gaussians to uniformity testing over the cube (3) They can reduce testing of specific product distributions called "extreme" to univariate testing, and thus use lower bounds from univariate testing to provide the first known lower bounds for private product testing. (1) is the key result - and they give two algorithms, one efficient, and one not. Both rely on the idea of privately computing a test statistic that has high global sensitivity and low average sensitivity by recursively taking Lipschitz extensions. In general computing the extension is inefficent, but it can be done approximately giving an efficient algorithm with worse sample complexity. Their bounds give an additive cost for privacy that only scales like sqrt(dimension) rather than linear. For epsilon = Omega(alpha) their bound shows that you can get privacy for free.

Strengths: - new results for privately computing a fundamental statistical primitive - interesting use of Lipschitz extensions to reduce global sensitivity to average sensitivity while still maintaining privacy, although this is similar to prior work - novel lower bounds for this problem

Weaknesses: * unclear how practically important improving the bound from (1/epsilon) multiplicatively to an additive term really is - in many applications epsilon could be O(1)

Correctness: - yes

Clarity: - yes

Relation to Prior Work: - yes

Reproducibility: Yes

Additional Feedback: This is thorough work that represents a definite contribution to private hypothesis testing. What prevents it from receiving a higher score is that I'm not sure how significant the results are, and of how much interest they are to the Neurips community.


Review 2

Summary and Contributions: The paper studies the problem of privately testing whether a set of data samples is drawn from the uniform distribution, or \alpha far from it. The paper also provides solution to generalizations of this problem, e.g., biased product distributions and Gaussian distribution with diagonal covariance. The paper provides improved sample complexity bounds that has better dependence on dimensions, and show that without computational assumptions the sample complexity is almost optimal in certain parameter regimes.

Strengths: 1. The paper is very well-written, and the results are very easy to follow. 2. The paper uses (by now standard techniques) like using proxy statistics which are stable on typical data sets, and techniques of Lipschitz extension in a fairly non-trivial way. 3. May be I am new to this line of research on property testing with differential privacy, but I found the technique of privacy filtering to be quite novel.

Weaknesses: I am not too much familiar with the area to criticize the paper much, beyond requesting the authors to demonstrate some motivation for practical importance of these problems in real-world settings of design of privacy preserving algorithms.

Correctness: The paper seems to be correct intuitively.

Clarity: Yes.

Relation to Prior Work: Yes.

Reproducibility: Yes

Additional Feedback:


Review 3

Summary and Contributions: The paper gives algorithms and techniques for differentially private hypothesis testing for relatively simple distributions in high dimensions. The main test case is to distinguish a sample from a uniform distribution on the d-dimensional hypercube to a product distribution over {-1,1}^d that is "far" from uniform in total variation distance

Strengths: The paper gives two interesting techniques to perform these tests. 1) Lipschitz extensions. This gives good sample complexity bounds but inefficient 2) Private filtering

Weaknesses: Some of the better bounds are inefficient, some of the bounds are suboptimal

Correctness: What i looked at seems ok, did not verify all

Clarity: yes

Relation to Prior Work: I am not too familiar with prior work on private hypothesis testing

Reproducibility: Yes

Additional Feedback: I have read the author's response


Review 4

Summary and Contributions: Differential privacy is an important area of continued research with increased applicability in industry. This paper derives an optimal hypothesis test with differential privacy. The results are theoretic in nature but clearly quantify the cost of privacy in terms of the tests themselves. The approach itself is clearly derived and focusses on the average case in order to manage challenges associated with metrics of extreme values (the dependence of the algorithm on single observations which in term conflict with the definitions of differential privacy).

Strengths: The paper was clear, well-motivated and offer concrete theoretical contributions. The authors described the motivations, algorithmic design, theoretical guarantee and broader impact of the approach. While there was no empirical evidence to support the approach, the argument via optimal guarantees, correctness and associated theorems left the paper complete. I believe it will be of interest to the NeurIPS community.

Weaknesses: Unfortunately, the paper was not the most readable. The ordering of the sections jumped around a lot (intro, background, results, techniques, related work, core section, impact). I would have suggested that background and related works are tied together and results and impact due to their natural relations. Nevertheless, this is merely nitpicking and there are not many overwhelming concerns with the work. One question that the reader is left with however is the applicability of the approach in practice and whether the additional cost of privacy (despite the obvious benefits) leads to poor hypothesis testing in the wild.

Correctness: There is no obvious issue with the papers theory itself as far as I am aware.

Clarity: Please see weaknesses.

Relation to Prior Work: The paper links to related research such as hypothesis testing and its applicability in the ML community and the Lipschitz-extension technique. Differential privacy literature and related topics are assumed to be well understood by the reader.

Reproducibility: Yes

Additional Feedback:

[Author Response · NeurIPS 2020]

We thank all the reviewers for their careful reading and thoughtful comments. We also thank the reviewers for general comments on the presentation, which we will address while preparing our final manuscript.

**Reviewer 4 - Importance of removing $1/\varepsilon$ factor**: We note that for algorithms with binary output (say YES or NO), larger values of $\varepsilon$ provide rather vacuous guarantees. For example, $\varepsilon = 1.1$ would permit an algorithm to go from outputting YES with probability $1/4$ to outputting YES with probability $3/4$ by changing just a single point, which we do not consider private. Additionally, data analysis pipelines (e.g., model selection) in practice typically contain many private analyses, therefore, in order to achieve a reasonable overall privacy guarantee, each individual private algorithm must have a small $\varepsilon$ as the privacy budget is split among the queries. For both these reasons, a minimal dependence on $\varepsilon$ is preferred. We note that the $1/\varepsilon$ multiplicative baseline can be achieved with a simple application of the subsample and aggregate framework, a general purpose method for producing differentially private testing algorithms from non-private testing algorithms. The transition from a multiplicative $\varepsilon$ factor to an additive $\varepsilon$ factor has both theoretical and practical significance. In some regimes our private sample complexity is dominated by the non-private sample complexity (which never happens with a multiplicative dependence on $\varepsilon$). This implies much lower sample complexities in many important regimes, even for moderate sized $\varepsilon$: for example with $\varepsilon = 0.1$, subsample and aggregate requires 10x as much data as the non-private algorithm, while for some settings of $d$ and $\alpha$, our algorithm requires less than 2x as much data.

**Reviewer 4 - Relevance to the community**: We note that these problems are of core interest to the community, and most papers in this particular line on private hypothesis testing have appeared in either NeurIPS or ICML (see, e.g., Cai et al. ICML'17, Cummings et al. NeurIPS'18, Acharya et al. NeurIPS'18, Aliakbarpour et al. ICML'18, Aliakbarpour et al. NeurIPS'19).

**Reviewer 6 - Practical importance**: Our paper falls into the category of goodness-of-fit testing, which is ubiquitous in scientific research including studies that typically use sensitive information such as voting behaviour or clinical trials (e.g. [1, 4, 5]). The specific problems we study use the assumption that the analyst knows the family of distributions that the data come from (product or multivariate normal distributions). These types of parametric tests are often more powerful than non-parametric ones in the sense that they require fewer samples, and are thus often used in medical research [2]. In particular, testing the mean of a normal distribution is one of the most fundamental statistical primitives, most often achieved via a Z-test or T-test. (Two-tailed) Z-tests are mean tests for normal distributions with known covariance, which is exactly one of the problems we study in this paper, and they are standard tests used in studies of treatment effects [3, 6]. Since drug trials are paradigmatic of studies where the data contain highly sensitive information, this demonstrates the need for sample-efficient differentially private alternatives for this task. Finally, we would like to note that, based on our proof techniques, we generally expect that our algorithms would perform well in practice, even if the distribution is not exactly Gaussian, but rather "well-behaved" around the origin.

**Reviewer 8 - Applicability of the approach**: We believe that determining the optimal sample complexity is an important first step to the implementation of practical differentially private algorithms for these problems. Therefore, we consider our near-optimal with respect to sample complexity but computationally inefficient algorithm to be an important, non-trivial, first step towards this goal. Moreover, our computationally efficient algorithm is relatively simple to implement (as it consists of a truncation step, a filtering step, and then applying a variant of the popular chi-squared statistic). We predict that the performance of the algorithms in the wild will be faithful to their theoretical guarantees and close to their non-private counterparts. Empirical evaluation of private identity tests with similar dependence on $\varepsilon$ in their theoretical guarantees shows that the cost of privacy is low in various settings (Cai et al. ICML'17).

# References

[1] Aaron Blair, P Decoufle, and Dan Grauman. Causes of death among laundry and dry cleaning workers. *American journal of public health*, 69:508–11, 06 1979.

[2] Richard Chin and Bruce Y. Lee. Chapter 15 - Analysis of Data. In *Principles and Practice of Clinical Trial Medicine*, pages 325 – 359. Academic Press, New York, 2008.

[3] S.-C. Chow, J. Shao, H. Wang, and Y. Lokhnygina. Sample size calculations in clinical research. In *Statistical Theory and Related Fields*, Chapman & Hall/CRC Biostatistics Series. Taylor & Francis, 2018.

[4] J Gill, J Endres-Brooks, P Bauer, W.J. Marks, and Robert Montgomery. The effect of ABO blood group on the diagnosis of von willebrand disease. *Blood*, 69:1691–5, 07 1987.

[5] William A. Glaser. The family and voting turnout. *The Public Opinion Quarterly*, 23(4):563–570, 1959.

[6] K. K. Gordon Lan, Yuhwen Soo, Cynthia Siu, and Mey Wang. The use of weighted Z-tests in medical research. *Journal of Biopharmaceutical Statistics*, 15(4):625–639, 2005. PMID: 16022168.


[Meta-Review · NeurIPS 2020]

This paper studies the problem of privately answering if a dataset comes from the uniform distribution (or "far away"). The paper provides improved sample complexity and show tightness of the sample complexity (information theoeretically). Overall the result is novel and interesting. Please note reviewer's comments on clarity and improve the presentation.